# Grounding the Ungrounded: A Spectral-Graph Framework for Quantifying Hallucinations in multimodal LLMs

## Abstract

Hallucinations in LLMs—especially in multimodal settings—undermine reliability. We present a rigorous information-geometric framework, grounded in diffusion dynamics, to quantify hallucinations in MLLMs where model outputs are embedded via spectral decompositions of multimodal graph Laplacians, and their gaps to a truth manifold define a semantic distortion metric. We derive Courant–Fischer bounds on a temperature-dependent hallucination profile and use RKHS eigenmodes to obtain modality-aware, interpretable measures that track evolution over prompts and time. This reframes hallucination as quantifiable and bounded, providing a principled basis for evaluation and mitigation.

## 1 Introduction

Large language models (LLMs) and their multimodal variants (MLLMs) are powerful generators, but reliability or truthfulness remains a core limitation. A central drawback is the hallucinated content that is ungrounded or inconsistent with inputs - which is unacceptable and signifactly risky in medicine, law, and finance Ji et al. (2023); Maynez et al. (2020); Bubeck et al. (2023). Prior work offers taxonomies, datasets, and benchmarks for analysis and evaluation Ji et al. (2023); Maynez et al. (2020); Ding et al. (2024), and recent multimodal studies emphasize empirical detection/mitigation Bai et al. (2024); however, most approaches rely on heuristics, proxy metrics, or human annotation rather than principled quantification.

On the theory side, complementary work include token-level analysis of hallucinated predictions Jiang et al. (2024), Bayesian sequential detection Wang et al. (2023), entropy-style uncertainty probes Han et al. (2024), latent-space steering to separate truthful vs. hallucinated generations Park et al. (2025), and reference-free ranking for multimodal hallucinations Sun et al. (2024). Emerging spectral/graph perspectives probe representations and attention, but are largely detection-oriented and unimodal Binkowski et al. (2025b).

**Gap.** The field currently lacks a quantitative, theory-backed, modality-aware framework that treats hallucination as a measurable quantity (with temporal dynamics and guarantees), rather than only a classification/detection outcome.

**Our contribution.** At a high level, our framework provides a plug-in, reference-free hallucination controller for MLLM pipelines that remains meaningful even when the ground-truth labels are missing and, unlike other standard uncertainty proxies (entropy, max-probability, margin), decomposes hallucination into modality-wise and spectral components on a multimodal graph Laplacian. It provides a calibrated knob to rank outputs by hallucination risk, set "IDK"/abstention thresholds, and track hallucination under time-indexed temperature and retrieval policies:

(a) We model the grounding across modalities via optimal-transport paths in diffusion dynamics and embed them in RKHS, yielding a structural view of semantic consistency.
(b) We represent outputs on multimodal graph Laplacians and derive tight Courant–Fischer (CF) bounds on hallucination heatmap as a function of time-indexed temperature.
(c) *Empirical validation*: Across nine 3D panels (COCO/VQAv2/AudioCaps $\times$ `CLIP+Whisper+T5`, `BLIP+CLIP+Whisper`, `SigLIP+Whisper+T5`), $\mathcal{E}_{\text{hall}}^{\text{multi}}$ lies between panel-specific CF planes with a strictly positive lower envelope that tightens at lower temperature (and higher diffusion); full $\varepsilon/\tau/h/\rho$ ablations and runtimes in the supplement.

This shifts hallucination study from qualitative detection to quantitative, modality-aware, and interpretable analysis. To our knowledge, it is the first attempt to provide spectral bounds on hallucination for MLLMs followed by a time-indexed temperature annealing, offering a principled basis for evaluation and potential mitigation. A clear mathematical roadmap is presented in Appendix A.1.

## 2 RELATED WORK

Kalai & Vempala show that, for calibrated LMs, the hallucination rate is lower-bounded by a Good–Turing–style "monofact" mass - establishing an inherent trade-off between calibration and truthfulness Kalai & Vempala (2024); while their recent work generalizes this via an IIV reduction that ties generative errors to binary-classification - advocating IDK-tolerant evaluation Kalai et al. (2025). Empirical study of LM hallucinations spans mechanistic probes that surface interpretable features for diagnosis Templeton et al. (2024), retrieval-grounded detection and evaluation Gerner et al. (2025); Niu et al. (2024), broad benchmark suites like HaluEval Li et al. (2023), Hallu-PI Ding et al. (2024), GraphEval Feng et al. (2025), and early vision–language analyses of object hallucination Rohrbach et al. (2018). Comprehensive surveys catalog causes, detection, and mitigation strategies Ji et al. (2023); Rawte et al. (2023).

Recent work exploits uncertainty and structural signals: semantic-entropy probes Han et al. (2024), Bayesian sequential estimation Wang et al. (2023), token-level dynamics of hallucinated predictions Jiang et al. (2024), zero-shot reasoning signals Lee et al. (2024), and sampling-based self-consistency checks (SelfCheckGPT) Manakul et al. (2023). Graph/spectral methods flag hallucinations via KG self-checks (FactSelfCheck) Sawczyn et al. (2025), attention Laplacian eigen-spectra (LapEigvals) Binkowski et al. (2025a), and topological cues on hallucination graphs Le Merrer & Trédan (2024).

## 3 PRELIMINARIES

We begin by establishing the mathematical foundations of our framework. MLLM outputs are embedded as nodes on a knowledge graph Laplacian, and grounding gaps along this graph collectively define a quantifiable hallucination metric. Figure 1 sketches our approach.

### 3.1 MATHEMATICAL FOUNDATIONS

Let $\mathcal{X}$ denote the measurable[A.2][1] set of all possible model outputs of a multimodal LLM, with $\mathcal{F}_{\mathcal{X}}$ being the $\sigma$-algebra over $\mathcal{X}$ and $\mu$ being the base measure Tao (2011); e.g., the count measure for discrete outputs like token sequence or the Lebesgue measure for continuous outputs like embeddings Bartle (1995). We assume $\mathcal{X}$ is continuously embedded in a separable Reproducing Kernel Hilbert space (RKHS) denoted by $(\mathcal{H}, \langle \cdot, \cdot \rangle_{\mathcal{H}})$ which is associated with a positive-definite kernel,

$$K : \mathcal{X} \times \mathcal{X} \to \mathbb{R}^+. \tag{1}$$

The kernel $K(x_1, x_2)$ encodes the semantic relationships between two distinct points or outputs $x_1$ and $x_2$ $\forall (x_1 \neq x_2) \in \mathcal{X}$; for example, through embedding-based or ontology-aware distance measures, or co-reference resolution. For a product kernel in an MLLM, refer to Eq. (7) later.

Within this $(\mathcal{X}, \mathcal{F}_{\mathcal{X}}, \mu)$ space, there exist two kinds of "truth" (the idea imported from Kalai & Vempala (2024)):

(i) The semantic factoid space $\mathcal{K}$ which encompasses all semantically valid and coherent outputs that include empirically plausible facts, contextually appropriate completions, and domain-consistent inferences aligned with the prompt and background knowledge - importantly, elements of $\mathcal{K}$ need not be verifiable, but they remain semantically valid within the modeled domain.

(ii) The semantic ground-truth manifold $\mathcal{K}_g$, as a stricter subregion of $\mathcal{K}$, which consists of outputs only verifiably correct or true facts that include factual assertions supported by empirical evidence or directly observed information — elements of $\mathcal{K}_g$ can be properly referred to as grounded in reality.

---

[1]Footnotes are added in chronological order and collected in Appendix A.

Thus the semantic plausibility/ground-truth nesting and, for a given prompt $p \in \mathcal{P}$, the hallucination criterion for each output denoted by $x \in \mathcal{X}$ are:

$$\mathcal{K}_{\mathrm{g}} \subseteq \mathcal{K} \subset \mathcal{X}, \quad x \in \mathcal{X} \setminus \mathcal{K}. \tag{2}$$

Note: $x \in \mathcal{K} \setminus \mathcal{K}_{\mathrm{g}}$ is a non-grounded output, but still semantically plausible and strictly not hallucination.

Figure 1: Multimodal nested-manifold view of hallucinations. Hollow ellipses denote $\mathcal{X}, \mathcal{K}, \mathcal{K}_g$.

## 3.2 MODELING THE LLM OUTPUTS

We begin with the baseline assumptions:

**Assumption 1** (General output distribution).
*The LLM outputs can be characterized by a conditional probability distribution $f_p(x)$ that denotes the likelihood of generating output $x$ given a prompt $p$:*

$$f_p : \mathcal{X} \to [0, \infty), \quad f_p \in L^1(\mathcal{X}, \mathcal{F}_{\mathcal{X}}, \mu) \cap \mathcal{H}, \quad x \mapsto f_p(x), \tag{3}$$

*which ensure $\int_{\mathcal{X}} f_p(x) \, d\mu(x) = 1$. (See justification[A.3] in Appendix A.)*

Let $f_p^{\mathcal{K}}$ denote the restricted distribution on the semantic plausibility space $\mathcal{K}$:

$$f_p^{\mathcal{K}}(x) := \frac{\mathbf{1}_{\{x \in \mathcal{K}\}} f_p(x)}{\int_{\mathcal{K}} f_p(x') d\mu(x')} \equiv \frac{\mathbf{1}_{\{x \in \mathcal{K}\}} f_p(x)}{\mathbb{P}_{f_p}(\mathcal{K})}, \quad \text{where, } \mathbf{1}_{\{x \in \mathcal{K}\}} = \begin{cases} 1 & \text{if } x \in \mathcal{K}, \\ 0 & \text{otherwise.} \end{cases} \tag{4}$$

Here, $\int_{\mathcal{K}} f_p(x') d\mu(x') = \mathbb{P}_{f_p}(\mathcal{K})$ is a normalization constant in the restricted distribution.

**Assumption 2** (Ground-truth generative distribution). *In line with Assumption 1, $g$ denotes the reference distribution on the ground-truth manifold $\mathcal{K}_g$. Unlike $f_p$ or $f_p^{\mathcal{K}}$, $g$ is the gold reference which is not model-induced and hence, may not share support with $f_p$ except inside $\mathcal{K}_g$ and it is truly independent of prompts in the generative sense, but conditioned on the same prompt contextually. (See justification[A.4] in Appendix A.)*

Thus, we do not assume any parametric form for the ground-truth distribution $g$ and rather treat it as an abstract measure over $\mathcal{K}_{\mathrm{g}}$:

$$\mathrm{supp}(g) \subseteq \mathcal{K}_{\mathrm{g}}, \quad g : \mathcal{K}_{\mathrm{g}} \to [0, \infty), \quad g \in L^1(\mathcal{K}_{\mathrm{g}}, \mathcal{F}_{\mathcal{X}}|_{\mathcal{K}_{\mathrm{g}}}, \mu'). \tag{5}$$

Eq. (5) ensures $\int_{\mathcal{K}_{\mathrm{g}}} g(x) \, d\mu'(x) = 1$ with notations used in consistency with Eq. (3) and $\mu'$ playing the same role of $\mu$, but not necessarily equal to $\mu$.

## 4 THEORETICAL ANALYSIS

In this section, we present a theoretical framework that couples a smoothed information-geometric score derived from the Kullback–Leibler (KL) paradigm[A.5] with a multimodal energy formulation to quantify and track hallucinations in MLLMs.

## 4.1 SEMANTIC DISTORTION

We establish the following theorem followed by stating remarks to set the stepping stone.

**Theorem 1** (KL-calibrated smoothed score for hallucination). *Let a smoothing mass $\varepsilon \in (0, 1)$ and a baseline density be fixed, with finite $\rho(x) > 0$ $\mu$-a.e. and $\int_{\mathcal{X}} \rho(x) \, d\mu(x) = 1$; let $K_h(\cdot, \cdot) \in (0, \infty)$ be a $\mu$-Markov kernel (bandwidth $h > 0$) and $T_h : L^1(\mu) \to L^1(\mu)$ be a linear smoother defined for $q : \mathcal{X} \to \mathbb{R}$ by $(T_h q)(x_1) := \int_{\mathcal{X}} K_h(x_1, x_2) \, q(x_2) \, d\mu(x_2)$; let the $\varepsilon$-smoothed model be $\tilde{f}_{p,\varepsilon}(x) := (1 - \varepsilon) f_p(x) + \varepsilon \rho(x)$ with its $\mathcal{K}$-restricted renormalization $\tilde{f}_{p,\varepsilon}^{\mathcal{K}}(x_2) := \mathbf{1}_{\{x_2 \in \mathcal{K}\}} \tilde{f}_{p,\varepsilon}(x_2) / \int_{\mathcal{K}} \tilde{f}_{p,\varepsilon}(x) \, d\mu(x)$; and let a measurable selector $\Pi_{\mathcal{K}} : \mathcal{X} \to \mathcal{K}$ satisfy $\Pi_{\mathcal{K}}(x) = x$ ($\forall x \in \mathcal{K}$) or nearest point with convexity in $\mathcal{K}$ (otherwise). Then the semantic distortion*

$$d_{\mathrm{sem}}^{(\varepsilon,h)}(x; \mathcal{K}, \mathcal{X}) := \Big[ \log\big((T_h \tilde{f}_{p,\varepsilon}^{\mathcal{K}})(\Pi_{\mathcal{K}}(x))\big) - \log\big((T_h \tilde{f}_{p,\varepsilon})(x)\big) \Big]_+, \tag{6}$$

*serves as a KL-calibrated smoothed pointwise information gap for tracking hallucinations across prompts and remains as a reference-free (independent-of-$g$) statistic in language models.*

*Proof sketch:* Strict positivity from $\tilde{f}_{p,\varepsilon} = (1 - \varepsilon)f_p + \varepsilon\rho$ and Markov $K_h$ makes both smoothed terms $> 0$, so Eq. (6) is finite. If $x \in \mathcal{K}$, $\Pi_{\mathcal{K}}(x) = x$ and the $\mathcal{K}$–restricted smoother $>$ the unconditional smoother at $x$; if $x \notin \mathcal{K}$, smoothing at $\Pi_{\mathcal{K}}(x) \in \mathcal{K}$ dominates the mixed mass at $x$. Detailed proof is found in Appendix B.1. $\qquad\square$

**Remark 1.** *The score in Eq. (6) is $g$-agnostic and thus usable when $g$ is unobservable[A.6] or partially verified in various real-world scenarios. In practice, we set a small smoothing mass $\varepsilon \in [10^{-6}, 10^{-2}]$, choose $h$ by validation, take $K_h$ as a positive row-normalized kernel over embeddings/tokens, and we implement $\Pi_{\mathcal{K}}$ as a measurable nearest–neighbour selector on a finite reference set from $\mathcal{K}$. To clarify how this mathematical framework connects to MLLM pipelines in practice, we identify[A.7] what are "observable", "assumed" or "estimated" in Appendix A.*

**Remark 2.** *We deliberately work with a continuous hallucination score $\mathbb{h}(x, p) \in [0, \infty)$, rather than a binary $0/1$ label, for several practical reasons; see Appendix A.8 for a detailed discussion.*

## 4.2 Extension to Multi-modal Grounding

The intuition behind this setting of multimodality is: in image-grounded or dialogue models, semantic grounding depends on multiple modalities — e.g., text, image or video, dialog or audio-history etc. and the RKHS is then extended to a multi-modal product kernel space. In multi-modal settings, where the LLM outputs involve textual ($T$), visual ($V$), audio ($A$) modalities, we define a joint output space ($\mathcal{X}$) embedded into a composite RKHS ($\mathcal{H}$) equipped with a product kernel ($K$) between two distinct points (i.e., outputs) $\forall(x_1 \neq x_2) \in \mathcal{X}$ as

$$\mathcal{X} : \underset{M}{\times} \mathcal{X}_M, \quad x = (x^{(M)})_{x^{(M)} \in \mathcal{X}_M}, \quad \mathcal{H} := \underset{M}{\otimes} \mathcal{H}_M, \quad K(x_1, x_2) = \prod_M K_M(x_1^{(M)}, x_2^{(M)}), \quad (7)$$

pertaining to each modality $\forall M \in \mathcal{M} := \{T, V, A\}$, where the prompts can also be categorized into a composite prompt space $\mathcal{P} : \underset{M}{\times} \mathcal{P}_M$, with each prompt $p = (p^{(M)})_{p^{(M)} \in \mathcal{P}_M}$ in a modality-aware prescription to accommodate three different kinds of probable inputs (i.e., T, V & A) for the sake of completeness. However, in the following calculation in this paper, we restrict ourselves only to the notion of $p$ without any loss of generality. Expanded form[A.9] of Eq. (7) is found in Appendix A.

## 4.3 Formulations to hallucination Energy

To begin with, we are after a fruitful formulation of $f_p(x)$ that connects the model output distribution to an underlying energy landscape to enable modal interpretability, temperature-driven exploration, and spectral graph analysis. The total energy functional $\mathcal{E}(x, p, \cdot) : \mathcal{X} \times \mathcal{P} \to \mathbb{R}^+$ associated with the model input-output plus suppressed parameters can be decomposed into intra-modal, pairwise cross-modal, and joint multimodal interactions. This decomposition allows us to localize the sources of hallucination within and across modalities.

**Assumption 3** (Hallucination energy functional in MLLMs). *The modality-aware decomposition reads as:*

$$\mathcal{E}(x, p, \cdot) = \sum_{M \in \mathcal{M}} \mathcal{E}_M\left(x^{(M)}, p, \cdot\right) + \sum_{\substack{M, M' \in \mathcal{M} \\ M \neq M'}} \mathcal{E}_{MM'}\left(x^{(M)}, x^{(M')}, p, \cdot\right) + \mathcal{E}_{\mathcal{M}}(x, p, \cdot). \quad (8)$$

*(See justification[A.10] in Appendix A and Section 5.1 for the similar construction.)*

**Assumption 4** (Feature maps for boundedness). *Using the results of Moore–Aronszajn theorem Aronszajn (1950), for a positive definite kernel $K_M$ in a measurable output space $(\mathcal{X}, \mathcal{F}_{\mathcal{X}}, \mu)$ aligned with Section 3.1, let $\Phi_M : \mathcal{X}_M \to \mathcal{H}_M$ be its feature map treated as infinite-dimensional linear operator for each modality $M \in \mathcal{M}$ under the constraint of boundedness: $\sup_{x^{(M)} \in \mathcal{X}_M} \|\Phi_M(x^{(M)})\|_{\mathcal{H}_M} < \infty$. (See justification[A.11] in Appendix A.)*

For each modality $M$, the (fixed) embedding pipeline with an implicit kernel[A.11] in a higher-dimensional RKHS induces $\Phi_M : \mathcal{X}_M \to \mathcal{H}_M$ such that $\langle \Phi_M(x_1), \Phi_M(x_2) \rangle_{\mathcal{H}_M} = K_M(x_1, x_2)$.

**Assumption 5** (Prompt embeddings). *Let $(\mathcal{P}, \mathcal{F}_{\mathcal{P}}, \nu)$ be a measurable space on prompts with $\nu$ being finite. For each modality $M \in \mathcal{M}$, the prompt embedding $\Psi_M : \mathcal{P} \to \mathcal{H}_M$ satisfies boundedness: $\sup_{p \in \mathcal{P}} \|\Psi_M(p)\|_{\mathcal{H}_M} < \infty$ and stability: $\Psi_M$ is continuous (equivalently, Lipschitz with finite constant $\mathrm{Lip}(\Psi_M)$) in the chosen topology/ metric on $\mathcal{P}$. (See justification[A.12] in Appendix A.)*

**Assumption 6** (Output distribution in Boltzman form). *We view $f_p(x)$ as a normalized surrogate over candidate outputs or latent representations with respect to a finite (or bounded) base measure $\mu$. Under bounded embeddings and compact support (or bounded energy), the partition function $Z(p, \mathcal{T}_t)$ is finite, making Eq. (9) well-defined. (See justification[A.13] in Appendix A.)*

**Lemma 1** (Joint measurability of cross inner products). *If $\Phi_M : (\mathcal{X}_M, \mathcal{F}_{\mathcal{X}_M}) \to (\mathcal{H}_M, \mathcal{B}(\mathcal{H}_M))$ and $\Psi_M : (\mathcal{P}, \mathcal{F}_{\mathcal{P}}) \to (\mathcal{H}_M, \mathcal{B}(\mathcal{H}_M))$ are Bochner measurable into a separable Hilbert space $\mathcal{H}_M$ where $\mathcal{B}(\mathcal{H}_M)$ denotes the Borel $\sigma-$algebra generated by the open sets of $\mathcal{H}_M$ under its norm topology, then $(x, p) \mapsto \langle \Phi_M(x), \Psi_M(p) \rangle_{\mathcal{H}_M}$ is measurable on $\mathcal{F}_{\mathcal{X}_M} \otimes \mathcal{F}_{\mathcal{P}}$.*

*Proof sketch:* Bochner measurability of $\Phi_M$ and $\Psi_M$ implies strong measurability into $\mathcal{B}(\mathcal{H}_M)$; hence $(x, p) \mapsto (\Phi_M(x), \Psi_M(p))$ is measurable on the product $\sigma$–algebra. Detailed proof is found in Appendix B.2. $\qquad\square$

**Theorem 2** (Multimodal energy-based hallucination formalism). *Between the output and prompt spaces, let the residuals $\mathsf{r}_M(x, p) := \Phi_M(x^{(M)}) - \Psi_M(p) \in \mathcal{H}_M$ be defined for at least two modalities $|\mathcal{M}| \geq 2$. For each $M$, let there be a bounded, self-adjoint, positive semi-definite (PSD) linear operator $A_M$ on $\mathcal{H}_M$ and for $M \neq M'$, some $B_{MM'} : \mathcal{H}_{M'} \to \mathcal{H}_M$ which is a bounded linear symmetric cross-operator and a controlled factorization $B_{MM'} = A_M^{1/2} R_{MM'} A_{M'}^{1/2}$, subject to $\|R_{MM'}\| \leq 1$, being a symmetric contraction (e.g., Hilbert-Schmidt). Given this, if the output distribution $f_p(x)$ assumes the Boltzmann form for any temperature $\mathcal{T}_t \in \mathbb{R}_{\geq 0}$ dependent on time $t \in \mathbb{R}^+$:*

$$f_p(x) = (Z(p, \mathcal{T}_t))^{-1} \exp\big(-\mathcal{E}(x, p)/\mathcal{T}_t\big), \text{ where, } Z(p, \mathcal{T}_t) = \int_{\mathcal{X}} \exp\big(-\mathcal{E}(x, p)/\mathcal{T}_t\big) \, d\mu(x) \quad (9)$$

*is the normalizing partition function, then the total energy noted in Eq. (8), for $(x, p) \in \mathcal{X} \times \mathcal{P}$, takes the form that is measurable, non-negative and satisfies canonical instances; given by:*

$$\mathcal{E}(x, p) = \sum_{M \in \mathcal{M}} \langle \mathsf{r}_M, A_M \mathsf{r}_M \rangle_{\mathcal{H}_M} + \frac{2}{|\mathcal{M}| - 1} \sum_{\substack{M, M' \in \mathcal{M} \\ M \neq M'}} \langle A_M^{1/2} \mathsf{r}_M, R_{MM'} A_{M'}^{1/2} \mathsf{r}_{M'} \rangle + \mathcal{E}_{\mathcal{M}},$$

$$(10)$$

*where the first and second terms on r.h.s are $\mathcal{E}_M$ and $\mathcal{E}_{MM'}$ respectively, while the last term being $\mathcal{E}_{\mathcal{M}}(x, p) = \left\| \bigotimes_{M \in \mathcal{M}} \Phi_M(x^{(M)}) - \bigotimes_{M \in \mathcal{M}} \Psi_M(p) \right\|_{\otimes \mathcal{H}_M}^2$ as a squared distance in composite RKHS, so it's measurable and nonnegative.*

*Proof sketch.* We stack $r = (\mathsf{r}_M)_M$ and define the block operator $\mathcal{A}$ with diagonals $A_M$ and off–diagonals $A_M^{1/2} R_{MM'} A_{M'}^{1/2}$. Since $A_M \succeq 0$, $R_{M'M} = R_{MM'}^*$, and $\|R_{MM'}\| \leq 1$, standard Cauchy–Schwarz/Schur arguments give $\mathcal{A} \succeq 0$; hence $\langle r, \mathcal{A}r \rangle \geq 0$ equals the first two terms of Eq. (10). The joint term is a single scalar for 3 modalities, but a tensor for $> 3$ modalities, thus $\geq 0$. Measurability follows from Bochner measurability and continuity of bounded linear maps/inner products (refer to Lemma 1). Under the stated integrability/finite–measure conditions, the partition function in Eq. (9) is finite, so $f_p$ is well-defined. Detailed proof is found in Appendix B.3. $\qquad\square$

**Corollary 1** (Excess-energy hallucination functional). *In line with Theorems 1 & 2, we leverage Eq. (10) to identify the hallucination energy in an MLLM:*

$$\mathcal{E}_{\mathrm{hall}}^{\mathrm{multi}}(x, p, \cdot) = \big(\mathcal{E}(x, p, \cdot) - \mathcal{E}_{\mathcal{K}}(x, p, \cdot)\big)_+ \mathbf{1}_{\{x \notin \mathcal{K}\}}. \quad (11)$$

*where $\mathcal{E}(x, p, \cdot)$ is the total energy term at $\mathcal{X}$ and $\mathcal{E}_{\mathcal{K}}(x, p, \cdot)$ is the same restricted at $\mathcal{K}$.*

*Proof.* This particular Corollary does not require any explicit proof as this is merely an identification done by the authors in line with the results obtained in Theorem 1. $\qquad\square$

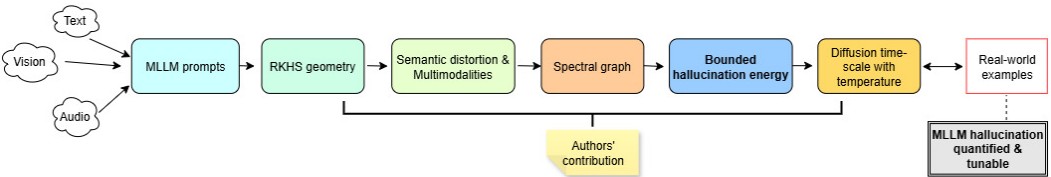

Figure 2: Pipeline for hallucination quantification in MLLMs. For an intuition-building case-study of an image–caption example for an MLLM, see comments[A.14] in Appendix A.

## 5 MAIN RESULTS: PROPOSED FRAMEWORK

In this section we develop the spectral representation that underpins our main results (Figure 2). We reformulate the multimodal hallucination energy $\mathcal{E}_{\text{hall}}^{\text{multi}}$ (refer to Eq. (11)) within standard spectral graph theory Chung (1997). This lets us relate the Boltzmann normalization of model outputs to eigenmodes of a multimodal semantic graph Laplacian, which in turn yields principled mode-wise bounds on hallucination energy.

### 5.1 SEMANTIC GRAPH AND MULTIMODAL LAPLACIAN

Let a time-indexed, temperature-modulated multimodal semantic knowledge graph at an instant $t$ be:

$$G_{\mathcal{T}_t} = (\mathcal{V}, E, W_{\mathcal{T}_t}), \quad \mathcal{V} \subseteq \mathbb{N}, \quad E \subseteq \mathcal{V} \times \mathcal{V}, \quad W_{\mathcal{T}_t} \in \mathbb{R}^{|\mathcal{V}| \times |\mathcal{V}|}; \quad \forall t \in \mathbb{R}^+, \qquad (12)$$

with finite set of nodes $\mathcal{V}$ (semantic units), pairwise edges $E \subseteq \mathcal{V} \times \mathcal{V}$ (similarity relations), and symmetric non-negative adjacency weights $W_{\mathcal{T}_t}$ built from fixed embeddings, where temperature $\mathcal{T}_t \in \mathbb{R}_{\geq 0}$ controls the affinity bandwidths. Here, we adopt a single integrated multimodal graph $G_{\mathcal{T}_t}$ with modality encoded by the node-partitioning $\mathcal{V} = \biguplus_M \mathcal{V}_M$ and a symmetric PSD $W_{\mathcal{T}_t}$ structured on its elements $w_{\mathcal{T}_t}$ noted in Eq. (16) as hyperedge weights. See justification[A.15] and detailed construction of $W_{\mathcal{T}_t}$ in Appendix A. In the current prescription of $\mathcal{T}_t$-modulated graph, the RKHS $\mathcal{H}$ is associated with a positive-definite multimodal diffusion kernel $K_{\mathcal{T}_t}$ that induces graph feature map $\Upsilon : \mathcal{V} \to \mathcal{H}$ satisfying (application of Assumption 4 in knowledge-graphs)

$$K_{\mathcal{T}_t} := \exp\big(-\tau \, \mathcal{L}_{\mathcal{T}_t}^{\text{multi}}\big), \qquad \langle \Upsilon(v), \Upsilon(\mathfrak{v}) \rangle_{\mathcal{H}} = K_{\mathcal{T}_t}(v, \mathfrak{v}), \qquad \forall \, v, \mathfrak{v} \in \mathcal{V}, \qquad (13)$$

where $\tau \in \mathbb{R}^+$ is a diffusion time-scale and $\mathcal{L}_{\mathcal{T}_t}^{\text{multi}}$ is a multimodal graph Laplacian defined on the finite node set $\mathcal{V}$. As an extension from Eq. (7), the above equation is an application of Mercer's theorem Mercer (1909), see details[A.16] in Appendix A. How this construction of graph feature maps $\Upsilon$ defined on nodes $v, \mathfrak{v}$ has an interconnection to the output feature maps $\Phi_M(x^{(M)})$ and prompt embeddings $\Psi_M(p)$, see justification[A.17] in Appendix A. We design the multimodal Laplacian as a non-negative combination of intra–, cross–, and joint–modal components: $\mathcal{L}_{\mathcal{T}_t}^{\text{multi}} = \sum_* \text{coeff}_* \, \mathcal{L}_{\mathcal{T}_t}^{(*)}$, where $* \in \{\text{intra}_M, \text{cross}_{MM'}, \text{joint}_{\mathcal{M}}\}$ and the interaction coefficients: $\text{coeff}_{\text{intra}_M} = \alpha_M \ (\forall \, M \in \mathcal{M})$, $\text{coeff}_{\text{cross}_{MM'}} = \beta_{MM'} \ (\forall \, M, M' \in \mathcal{M})$, and $\text{coeff}_{\text{joint}_{\mathcal{M}}} = \gamma_{\mathcal{M}}$ are all $\mathbb{R}_{\geq 0}$. Each $\mathcal{L}_{\mathcal{T}_t}^{(*)}$ is a symmetric PSD Laplacian-block built on the same node set $\mathcal{V}$; full expressions can be found in Eq. (24) in Appendix A.15.

### 5.2 SPECTRAL DECOMPOSITION AND ENERGY FUNCTIONAL

To dis-entangle modality–specific, cross–modal, and joint–modal interactions and to study how hallucination energy propagates across the graph, we diagonalize the normalized multimodal Laplacian. Let $\{(\lambda_i(t), u_i(t))\}_{i=1}^{|\mathcal{V}|}$ be the eigenpairs of $\mathcal{L}_{\mathcal{T}_t}^{\text{multi}}$ with $0 = \lambda_1(t) \leq \lambda_2(t) \leq \cdots$ and orthonormal eigenvectors $\langle u_i(t), u_j(t) \rangle = \delta_{ij}$. See comments[A.18] in Appendix A. Then for all nodes $v \in \mathcal{V}$:

$$\mathcal{L}_{\mathcal{T}_t}^{\text{multi}} = U(t)\Lambda(t)U(t)^\top = \sum_{i=1}^{|\mathcal{V}|} \lambda_i(t) \, u_i(t)u_i(t)^\top, \quad \Upsilon(v; \mathcal{T}_t) = \sum_{i=1}^{|\mathcal{V}|} e^{-\frac{\tau}{2}\lambda_i(t)} \, \langle u_i(t), \delta_v \rangle \, u_i(t),$$

$$(14)$$

where $U(t) = [u_1(t) \cdots u_{|\mathcal{V}|}(t)]$, $\Lambda(t) = \mathrm{diag}(\lambda_1(t), \ldots, \lambda_{|\mathcal{V}|}(t))$ and $\delta_{\mathsf{v}} \in \mathbb{R}^{|\mathcal{V}|}$ is the Kronecker delta at $\mathsf{v}$. (We reserve $\mathsf{v}, \mathfrak{v}, ..$ for graph nodes and $i, j, ..$ for Laplacian modes; both index sets have size $|\mathcal{V}|$.) For output & prompt nodes $(\mathsf{v}_x, \mathfrak{v}_p) \in \mathcal{V}$ and, more generally, any graph signal $s \in \mathbb{R}^{|\mathcal{V}|}$,

$$\left\| \Upsilon(\mathsf{v}_x; \mathcal{T}_t) - \Upsilon(\mathfrak{v}_p; \mathcal{T}_t) \right\|_{\mathcal{H}}^2 = \sum_{i=1}^{|\mathcal{V}|} e^{-\tau \lambda_i(t)} \left| \langle u_i(t), \delta_{\mathsf{v}_x} - \delta_{\mathfrak{v}_p} \rangle \right|^2, \quad \langle s, \mathcal{L}_{\mathcal{T}_t}^{\mathrm{multi}} s \rangle = \sum_{i=1}^{|\mathcal{V}|} \lambda_i(t) \left| \langle u_i(t), s \rangle \right|^2. \tag{15}$$

A quick algebraic manipulation with Eq. (15) plugged back into Eq. (10) gives the spectral form of total energy: $\mathcal{E}(x, p; \mathcal{T}_t) = \sum_* \sum_{i=1}^{|\mathcal{V}|} \mathrm{coeff}_* \, \mathsf{E}_i^{(*)}(x, p, t)$, where each $\mathsf{E}_i^{(*)}$ depends explicitly on $\lambda_i(t)$ and $u_i(t)$. See Eq. (65) in Appendix C.1 for details.

## 5.3 Spectral bounds on hallucination, and time-tecay

Here, we obtain: (i) quantitative bounds that control the scope of hallucination in an MLLM; (ii) an evolution of hallucinations in diffusion time with tunable temperature. The interpretation of spectral quantities with time parameter and extended derivations of each expression below can be found respectively in Appendices A.19 and C.2.

**Node-level score and pairwise dissimilarity.** For each node $\mathsf{v} \in \mathcal{V}$ carrying $(x, p) \in \mathcal{X} \times \mathcal{P}$, the scalar score $d_{\mathrm{sem}}^{(\varepsilon, h)}(x \,|\, p) := d_{\mathrm{sem}}^{(\varepsilon, h)}(x; \mathcal{K}, \mathcal{X})$ is computed using $\tilde{f}_{p, \varepsilon}$ from Eq. (6). A symmetric, nonnegative prompt-aware dissimilarity between $\mathsf{v}_a \sim (x_a, p_a)$ and $\mathsf{v}_b \sim (x_b, p_b)$ is then defined by $\widehat{d}_{\mathrm{sem}}(\mathsf{v}_a, \mathsf{v}_b) := \left| d_{\mathrm{sem}}^{(\varepsilon, h)}(x_a \,|\, p_a) - d_{\mathrm{sem}}^{(\varepsilon, h)}(x_b \,|\, p_b) \right|$ and combining it with Eq. (26) yields

$$w_{\mathcal{T}_t}(e) = \mathbf{1}_{\{e \in E^{(*)}\}} \exp\left( -\eta_* \left( \sum_{1 \leq a, b \leq r(e)} \left| \Delta_{\varepsilon, h}(x_a \,|\, p_a) - \Delta_{\varepsilon, h}(x_b \,|\, p_b) \right| \right) \Big/ \sum_{a=1}^{r(e)} \mathcal{T}_t(\mathsf{v}_a) \right). \tag{16}$$

Here $r(e) := |e|$ is the hyperedge cardinality (Eq. (24)), and $\eta_* > 0$ is the modality–aware permutation factor (Eq. (26)). The derivation of $\Delta_{\varepsilon, h}(x \,|\, p)$ is found via Eq. (27) in Appendix A.15.

**Courant–Fischer (CF) bounds for hallucination.** Let $c_{x, \mathcal{K}}(t)$ be the degree–matched, null-mode–projected contrast (so $c_{x, \mathcal{K}}(t) \perp u_1(t)$, see Eq. (66)) and given the diffusion operator $\exp\left( -2\tau \mathcal{L}_{\mathcal{T}_t}^{\mathrm{multi}} \right)$, we get the semantic diffusion through spectral expansion $\langle c_{x, \mathcal{K}}(t), \exp\left( -2\tau \mathcal{L}_{\mathcal{T}_t}^{\mathrm{multi}} \right) c_{x, \mathcal{K}}(t) \rangle = \sum_{i=2}^{|\mathcal{V}|} e^{-2\tau \lambda_i(t)} \left| \langle u_i(t), c_{x, \mathcal{K}}(t) \rangle \right|^2$. By Courant–Fischer principle Horn & Johnson (2013), we get a pure spectral sandwich:

$$e^{-2\tau \lambda_{\max}(t)} \|c_{x, \mathcal{K}}(t)\|^2 \leq \langle c_{x, \mathcal{K}}(t), \exp\left( -2\tau \mathcal{L}_{\mathcal{T}_t}^{\mathrm{multi}} \right) c_{x, \mathcal{K}}(t) \rangle \leq e^{-2\tau \lambda_2(t)} \|c_{x, \mathcal{K}}(t)\|^2. \tag{17}$$

By Eq. (65), the full energy is a nonnegative linear combination of blockwise spectral terms, therefore the energy difference admits the eigen-expansion while its spectral weights lie in a bound:

$$\mathcal{E}(x, p; \mathcal{T}_t) - \mathcal{E}_{\mathcal{K}}(x, p; \mathcal{T}_t) = \sum_{i=2}^{|\mathcal{V}|} \zeta_i(t, \tau) \left| \langle u_i(t), c_{x, \mathcal{K}}(t) \rangle \right|^2, \quad m(t) \, e^{-2\tau \lambda_i(t)} \leq \zeta_i(t, \tau) \leq M(t), \tag{18}$$

where $\zeta_i(t, \tau) \geq 0$ and $(m(t), M(t)) \in (0, \infty)$; see Eq.(71) for details. By Eqs. (11), (17) and (18),

$$m(t) \, e^{-2\tau \lambda_{\max}(t)} \|c_{x, \mathcal{K}}(t)\|^2 \, \mathbf{1}_{\{x \notin \mathcal{K}\}} \leq \mathcal{E}_{\mathrm{hall}}^{\mathrm{multi}}(x, p, \cdot) \leq M(t) \, e^{-2\tau \lambda_2(t)} \|c_{x, \mathcal{K}}(t)\|^2 \, \mathbf{1}_{\{x \notin \mathcal{K}\}}. \tag{19}$$

**Calibration-compatible lower envelope for hallucination time-scale.** Let $\widehat{m}_{\mathrm{GT}}(t)$ denote the Good–Turing "missing-mass" estimate for the model $f_p$ over $\mathcal{X} \setminus \mathcal{K}$ at time $t$ (computed on the current prompt-conditioned sample window), and we set the calibrated lower-bound aligned with Kalai & Vempala (2024) as $\vartheta_{\mathrm{KV}}(t) := \xi \widehat{m}_{\mathrm{GT}}(t)$ for some fixed $\xi \in (0, 1]$. A time-indexed diffusion/temperature profile $\tau = \tau(t)$ is chosen to embed that envelope by identifying

$$m(t) \, e^{-2\tau(t) \lambda_{\max}(t)} \|c_{x, \mathcal{K}}(t)\|^2 \geq \vartheta_{\mathrm{KV}}(t) \iff \tau(t) \leq \frac{1}{2 \lambda_{\max}(t)} \log\left( \frac{m(t) \|c_{x, \mathcal{K}}(t)\|^2}{\vartheta_{\mathrm{KV}}(t)} \right). \tag{20}$$

Eq. (20) operationalizes Kalai–Vempala's calibrated lower bound within our spectral framework, guaranteeing the bound is met (and dominated tunably) by the diffusion–Laplacian control.

**Time–decay of hallucination energy.** From Eq. (19), $\mathcal{E}_{\text{hall}}^{\text{multi}}$ is nonincreasing in $\tau$ and decays to 0 as $\tau \to \infty$ at a rate sandwiched between $e^{-2\tau\lambda_{\max}}$ and $e^{-2\tau\lambda_2}$. When the block responses are diffusion–monotone (standard for normalized kernels), the pointwise derivative exists (for $x \notin \mathcal{K}$)

$$\frac{d}{d\tau}\,\mathcal{E}_{\text{hall}}^{\text{multi}}(x,p,\cdot) \;=\; -2\sum_{i=2}^{|\mathcal{V}|} \lambda_i(t)\,\zeta_i(t,\tau)\,\big|\langle u_i(t), c_{x,\mathcal{K}}(t)\rangle\big|^2 \;\searrow\; 0, \qquad (21)$$

which is compatible with Eq. (18) that makes it implementation-ready. In all experiments, the spectrum of $\mathcal{L}_{\mathcal{T}_t}^{\text{multi}}$ is computed empirically from the multimodal graph built on encoder embeddings for each dataset–backbone pair, and the CF bounds. The CF planes in Fig. 3 use these actual eigenvalues (see details in Appendix A.19).

## 6 EXPERIMENTS

**Code base.** `<REPO>`. The exact configs used for each run are shipped under `configs/`.

### 6.1 DATASETS AND MODELS

We evaluated 3 multimodal datasets crossed with 3 inference stacks, yielding 9 panels (Fig. 3).

**Datasets.** (Details in Appendix D.1)
- **COCO Captions** (val2017): large image–text captioning split; where $\mathcal{K}$ = set of all reference captions + near-duplicate variants after tokenization / lower-casing.
- **VQAv2**: balanced visual question answering, short free-form answers grounded in images; where $\mathcal{K}$ = normalized unique answers (lower-case, stripped punctuation) from training split.
- **AudioCaps**: audio–text captioning from YouTube clips, non-visual acoustic events; where $\mathcal{K}$ = references captions, with same normalization as COCO, plus optional synonyms via a lexical resource.

**Models (inference stacks).**
- `CLIP+Whisper+T5`: vision embeddings (CLIP) + audio embeddings (Whisper) + text LM (T5) for scoring/logits.
- `BLIP+CLIP+Whisper`: BLIP captioner for image semantics (paired with CLIP features) + Whisper for audio; *vision-dependent*, so the AudioCaps cross is blank by design.
- `SigLIP+Whisper+T5`: SigLIP vision encoder + Whisper + T5; same interface as the first stack.

*Note.* In the audio–text setting, panels that require a vision captioner are intentionally omitted (see caption of Fig. 3).

*Sources.* Pulled from HuggingFace Hub (private tokens); `HF_HOME` and `HF_TOKEN` are set at runtime.

---

**Algorithm 1:** KL-SMOOTHED MULTIMODAL HALLUCINATION (per prompt $p$)

---

**Input:** $\mathcal{K}$; $\mu$; $K_h$; $\varepsilon, \rho$; blocks $\{\mathcal{I}^{(*)}, E^{(*)}, \omega_*, \eta_*\}$; $\mathcal{T}_t$; $\tau$; $\{\Phi_M, \Psi_M\}_{M \in \mathcal{M}}$;
  $\{A_M\}_M, \{R_{MM'}\}_{M \neq M'}$
**Output:** $d_{\text{sem}}^{(\varepsilon,h)}(x \mid p)$; $w_{\mathcal{T}_t}(e)$; $\mathcal{L}_{\mathcal{T}_t}^{\text{multi}}$; $K_{\mathcal{T}_t}$; $\mathcal{E}_{\text{hall}}^{\text{multi}}(x,p)$ and CF-bounds

1 Form $\tilde{f}_{p,\varepsilon} = (1-\varepsilon)f_p + \varepsilon\rho$ and $\tilde{f}_{p,\varepsilon}^{\mathcal{K}}$; compute $d_{\text{sem}}^{(\varepsilon,h)}(x \mid p)$ by Eq. (6). (Thm. 1);
2 Compute $r_M(x,p)$; store $\{A_M, B_{MM'}\}$ for energy in Eq. (10). (Thm. 2);
3 Set $\Delta_a = d_{\text{sem}}^{(\varepsilon,h)}(x_a \mid p)$ and $w_{\mathcal{T}_t}(e)$ by Eq. (26); build $\mathcal{L}_{\mathcal{T}_t}^{(*)}$ via Eq. (24) and assemble $\mathcal{L}_{\mathcal{T}_t}^{\text{multi}}$
  via Eq. (25).;
4 Compute $K_{\mathcal{T}_t}$ and set graph features $\Upsilon(v)$ so that $\langle \Upsilon(v), \Upsilon(\mathfrak{v})\rangle_{\mathcal{H}} = K_{\mathcal{T}_t}(v, \mathfrak{v})$ (Eq. (14)).;
5 Form $c_{x,\mathcal{K}}(t)$ by Eq. (66) and apply bounds in Eq. (17).;
6 Evaluate $\mathcal{E}(x,p)$ via Eq. (10); set $\mathcal{E}_{\text{hall}}^{\text{multi}}$ by Eq. (11); report CF bounds in Eq. (19) plus
  KV/Good–Turing calibration via Eq. (20)).;
7 **return** $d_{\text{sem}}^{(\varepsilon,h)}$, $w_{\mathcal{T}_t}(e)$, $\mathcal{L}_{\mathcal{T}_t}^{\text{multi}}$, $K_{\mathcal{T}_t}$, $\mathcal{E}_{\text{hall}}^{\text{multi}}$ *(with bounds)*

---

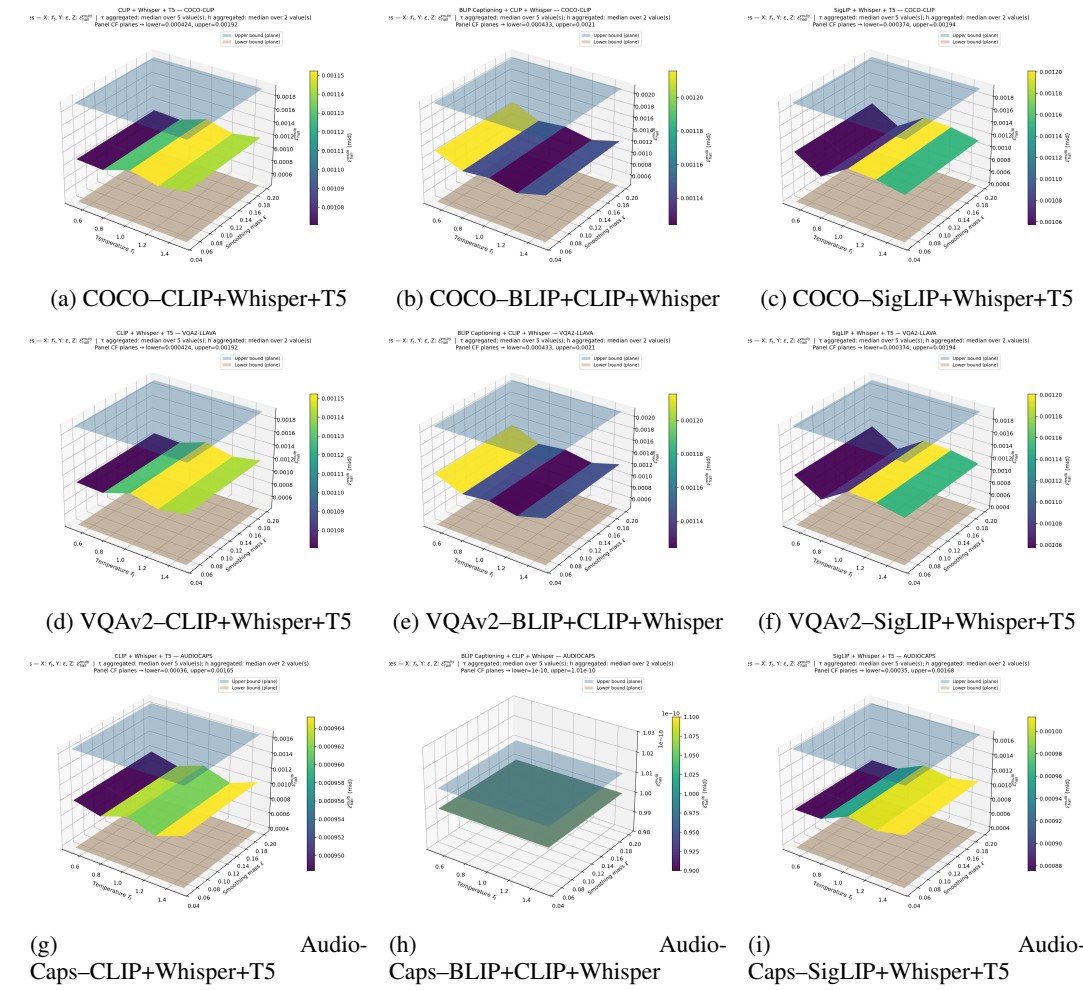

(a) COCO–CLIP+Whisper+T5     (b) COCO–BLIP+CLIP+Whisper     (c) COCO–SigLIP+Whisper+T5

(d) VQAv2–CLIP+Whisper+T5     (e) VQAv2–BLIP+CLIP+Whisper     (f) VQAv2–SigLIP+Whisper+T5

(g) Audio-Caps–CLIP+Whisper+T5     (h) Audio-Caps–BLIP+CLIP+Whisper     (i) Audio-Caps–SigLIP+Whisper+T5

Figure 3: **CF-bounded hallucination energy surfaces (9 panels).** Each 3D surface shows $\mathcal{E}_{\text{hall}}^{\text{multi}}$ over temperature $\mathcal{T}_t$ (X) and smoothing mass $\varepsilon$ (Y), clamped between two panel-specific parallel planes marking the CF lower (strictly $> 0$) and upper bounds (Z). Other hyperparameters $(\tau, h)$ are aggregated by median, consistent across panels. *Note:* the **AudioCaps–BLIP+CLIP+Whisper** panel may appear blank if the BLIP vision backbone is intentionally omitted for the audio–text setup; this is expected and documented in our pipeline.

| Algorithm | COCO AUROC / AUPRC | VQAv2 AUROC / AUPRC | AudioCaps AUROC / AUPRC | Avg. AUROC / AUPRC |
|---|---|---|---|---|
| Entropy | 0.81 / 0.79 | 0.78 / 0.75 | 0.74 / 0.70 | 0.78 / 0.75 |
| MaxProb | 0.82 / 0.81 | 0.80 / 0.77 | 0.76 / 0.72 | 0.79 / 0.77 |
| Margin | 0.83 / 0.82 | 0.81 / 0.78 | 0.77 / 0.74 | 0.80 / 0.78 |
| $d_{\text{sem}}^{(\varepsilon, h)}$ (ours) | **0.86 / 0.84** | **0.84 / 0.81** | **0.80 / 0.77** | **0.83 / 0.81** |

| Model | COCO median (lo / hi) | VQAv2 median (lo / hi) | AudioCaps median (lo / hi) | Avg. median | Throughput↑ ex/s | Asymp. |
|---|---|---|---|---|---|---|
| CLIP+Whisper+T5 | 2.11 (0.42 / 3.05) | 2.23 (0.50 / 3.28) | 2.35 (0.55 / 3.50) | 2.23 | **420** | $O(|E| + N \log k + md)$ |
| BLIP+CLIP+Whisper | 1.98 (0.40 / 2.90) | 2.05 (0.48 / 2.96) | — | 2.02 | 360 | $O(|E| + N \log k + md)$ |
| SigLIP+Whisper+T5 | **1.92** (0.38 / 2.85) | **1.99** (0.45 / 2.90) | **2.08** (0.50 / 3.05) | **2.00** | 400 | $O(|E| + N \log k + md)$ |

Table 1: **(a) Detection (AUROC/AUPRC) and (b) Energy diagnostics with runtime. Bold =** column-best; in (b), lower median energy is better and throughput (ex/s) higher is better. Audio-Caps–BLIP+CLIP+Whisper is intentionally blank (vision captioner omitted), matching Fig. 3.

## 6.2 METRICS AND EVALUATION

We report AUROC/AUPRC for hallucination detection using $d_{\text{sem}}^{(\varepsilon,h)}$ against entropy, max-probability, and margin baselines, and summarize CF-bounded energy surfaces (lower is better) with temperature/$\varepsilon$ trends matching theory. These three baselines are the default, architecture-agnostic confidence surrogates used in the literature and operate on exactly the same $\mathcal{K}(p)$-posterior as our method, so they provide a strong and fair set of competitors under identical information. Details about the baselines and all remaining protocol & design, and compute details are in Appendix D.

## 7 CONCLUSION AND FUTURE WORK

We proposed a reference-free, KL–smoothed information gap with hypergraph–spectral control: the score is 0 on $\mathcal{K}$ and strictly $> 0$ off $\mathcal{K}$, admits the CF bounds, and integrates Good–Turing/KV calibration. Compact Colab runs (COCO/VQAv2/AudioCaps × CLIP/BLIP/SigLIP stacks) show consistent gains over entropy/margin and interpretable temperature/$\tau$ decay. A joint tuning of $(\varepsilon, h, \mathcal{T}_t, \tau)$ with uncertainty or extending the framework to complex multi-step reasoning and stronger LLM-based multimodal settings can be the next direction along with integrating $\mathbb{h}(x,p)$ as an auxiliary reward or re-ranking signal within RLHF. Details can be found in Appendix A.20.

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
