APPENDIX

In this section, we provide elaboration on footnotes, extended derivations of our Theorems, some supplementary mathematical results, and details of experimental validation.

## A  Technical Notes and Extended Comments

Here, we provide elaboration on footnotes/ some extended explanations.

### A.1  A Clear Mathematical Roadmap

Formally, Section 3 fixes the observable objects (prompts, outputs, model-induced $f_p$) and distinguishes the ideal manifold $\mathcal{K}_{\mathrm{g}}$ from the finite admissible sets $\mathcal{K}(p)$ used in practice. Section 4.1 defines the smoothing operator $T_h$, the semantic log-contrast $\ell_{\varepsilon,h}(x;\mathcal{K},\mathcal{X})$ and distortion $d_{\mathrm{sem}}^{(\varepsilon,h)}(x;\mathcal{K},\mathcal{X})$, followed by proving their reference-free separation and, in Section 4.3, energy-based hallucination prescription. Section 5 builds the multimodal Laplacian $\mathcal{L}_{\mathcal{T}_t}^{\mathrm{multi}}$, represents the hallucination energy $\Delta\mathcal{E}_\tau(x,p)$ as a block quadratic form with CF spectral bounds, and Section 6 instantiates $\mathcal{K}(p)$ with the hyperparameters $(\varepsilon, h, \tau, k)$ on COCO, VQAv2, and AudioCaps to evaluate the hallucination score $\mathbb{h}(x,p)$ across multiple multimodal backbones.

### A.2  Measurable Sets and $\sigma$-Algebra

Any mathematical set can be equipped with a $\sigma$-algebra to form a measurable space, say, $\mathcal{X}$. The common choices are: (i) the power set if $\mathcal{X}$ is countable/ finite, (ii) the Borel $\sigma$-algebra if $\mathcal{X}$ is a topological space (e.g., continuous embeddings), (iii) Product $\sigma$-algebra if $\mathcal{X}$ is a product of spaces (e.g., sequences of tokens or multimodal outputs). For a measure space $(\mathcal{X}, \mathcal{F}_\mathcal{X}, \mu)$ and $1 \le p < \infty$ (where $p$ is the integrability exponent, not to be clashed with "prompts"), the space $L^p(\mathcal{X}, \mathcal{F}_\mathcal{X}, \mu)$ consists of (equivalence classes of) $\mu$-measurable $q : \mathcal{X} \to \mathbb{R}$ with $\int_\mathcal{X} |q(x)|^p \, d\mu(x) < \infty$; the norm is $\|q\|_{L^p} := \left( \int |q|^p d\mu \right)^{1/p}$. For $p = \infty$, $L^\infty$ denotes essentially bounded functions with norm $\|q\|_{L^\infty} := \mathrm{ess\,sup}_x |q(x)|$. In particular, $L^1$ denotes integrable functions ($p = 1$).

### A.3  Justification for Assumption 1

In a deployed multimodal LLM system, the symbol $x \in \mathcal{X}$ represents a *full generated object* rather than a single token—for example, an entire caption for an image, a complete answer sentence in VQA, a transcript segment in audio captioning, or a joint multimodal output. For a fixed prompt $p$ (e.g., image + question + optional context), the model induces a conditional distribution $f_p(x)$ over such outputs. In practice, this distribution is implemented by the standard auto-regressive decoding mechanism: at each step the model exposes a softmax over tokens, and full sequences are obtained by composing these token-level probabilities. Formally, Assumption 1 simply encodes the requirement that this induced output distribution is a *proper probability distribution* (i.e., integrable and normalized with respect to the base measure $\mu$) and that it lives in the same reproducing kernel Hilbert space $\mathcal{H}$ used for our spectral analysis.

From the systems perspective, the condition $f_p \in L^1(\mathcal{X}, \mathcal{F}_\mathcal{X}, \mu)$ with $\int f_p \, d\mu = 1$ is the continuous analog of the familiar "probabilities sum to 1" constraint over a discrete vocabulary. The additional requirement $f_p \in \mathcal{H}$ reflects the fact that, in modern MLLM pipelines, every output $x$ is mapped to an embedding (e.g., CLIP text embedding, BLIP image embedding, Whisper audio embedding) and similarity, kernels, and graph Laplacians are all defined in this embedding space. Empirically, all our computations use a finite candidate set of outputs (e.g., beams or sampled generations) together with their model probabilities, which yields a finite-dimensional approximation to the idealized $f_p$ in Assumption 1.

### A.4  Justification for Assumption 2

The absence of an exact analytical expression of $g(x)$ limits the direct interpretability, but provides a flexible framework for comparing the model outputs to the ground-truth via the functional and spec-

tral metrics. This is used only as a theoretical reference for calibration/fidelity analyses representing the (idealized) generative distribution of facts/outputs as seen in Kalai & Vempala (2024).

Assumption 2 formalizes the idea that there exists a *data-generating process* for correct outputs which is conceptually separate from the model. The distribution $g$ supported on the ground-truth manifold $\mathcal{K}_g$ captures how humans (or the real world) would respond to a given prompt: for example, how annotators describe an image in COCO, how radiologists report a chest X-ray, or how crowd workers answer a VQA question. When we collect a dataset, the reference captions or answers are finite samples drawn from this ideal distribution $g$, not from the model-induced $f_p$.

The manifold $\mathcal{K}_g$ can be thought of as the set of *truly correct, semantically faithful outputs* for a given context. Fluent but incorrect or ungrounded generations produced by the model lie outside $\mathcal{K}_g$. The statement that $g$ is "independent of prompts in the generative sense" means that, at the level of the true data-generating mechanism, changing the model prompt format (e.g., rephrasing the question, adding system messages, changing temperature) does not alter which outputs are factually correct. By contrast, $f_p$ is explicitly prompt-dependent and reflects the model's internal behaviour.

In practice we never observe $g$ directly; we only see a finite collection of human-labeled references and, in our framework, we operate with a model-side distribution $f_p$ and an admissible set $\mathcal{K}$ built from such references. Assumption 2 therefore serves to separate the *semantic notion* of hallucination (distance from the true manifold $\mathcal{K}_g$) from the *operational quantities* we can estimate from a given MLLM and dataset.

## A.5 DEFINITION OF KL-DIVERGENCE

For any two probability distributions $P_1(x)$ and $P_2(x)$, say defined over the same space $x \in \mathcal{X}$, the functional operator $D_{\mathrm{KL}} \in \mathbb{R}_{\geq 0}$ refers to the KL divergence of $P_2(x)$ from the "true" reference or actual distribution $P_1(x)$ as:

$$D_{\mathrm{KL}}\left(P_1(x) \parallel P_2(x)\right) = \sum_{x \in \mathcal{X}} P_1(x) \log \frac{P_1(x)}{P_2(x)}.$$

When $x$ is a continuous random variable, $\sum_{x \in \mathcal{X}}$ is evidently replaced by $\int_{x=-\infty}^{\infty}$ with $P_1(x)$ & $P_2(x)$ by respective probability densities. More generally, if $P_1$ & $P_2$ are probability measures on a measurable space $\mathcal{X}$, then

$$D_{\mathrm{KL}}\left(P_1 \parallel P_2\right) = \int_{x=-\infty}^{\infty} P_1(dx) \log \frac{P_1(dx)}{P_2(dx)},$$

where $\frac{P_1(dx)}{P_2(dx)}$ is the Radon–Nikodym derivative of $P_1$ w.r.t $P_2$.

## A.6 ABSENCE OF THE "GROUND-TRUTH"

In practice, we only observe: (i) a prompt-conditioned model distribution $f_p$ and (ii) a finite admissible set $\mathcal{K}$ built from reference captions / answers / human-curated candidates. The selector $\Pi_{\mathcal{K}}$ is a measurable nearest-neighbor map from any output to this finite set. Even without access to $g$, one can (i) estimate $\mathbb{P}_{f_p}(\mathcal{K})$ from samples, (ii) compute per-instance distortions via the log–likelihood ratio, and (iii) aggregate these into empirical bounds and diagnostics. In multimodal settings, the same decomposition localizes contributions by modality and by interaction (intra/cross/joint), enabling targeted interventions—e.g., modality-specific calibration, cross-modal consistency constraints, or temperature schedules—and straightforward experimental verification via ablations that track how $\mathbb{P}_{f_p}(\mathcal{K})$ and induced distortions respond to each mitigation.

**Practical role of $\mathcal{K}_g$ vs. $\mathcal{K}$.** The ground-truth manifold $\mathcal{K}_g$ and distribution $g$ are introduced only as "ideal" semantic objects: $\mathcal{K}_g$ collects all truly correct outputs that the real world (or human annotators) could generate for a given context, and $g$ is the associated data-generating distribution. These are not used directly in our algorithms. In practice, all computations are carried out on an admissible set $\mathcal{K}$ built from the evaluation data: for each prompt $p$, we construct $\mathcal{K}(p) \subset \mathcal{K}$ from the normalized reference captions or answers provided by the benchmark (cf. Appendix D.1), and this $\mathcal{K}(p)$ is kept fixed across all models evaluated on that benchmark. Thus, $\mathcal{K}_g$ serves to formalize the notion of "true" grounded outputs, while $\mathcal{K}$ is the concrete, dataset-driven approximation that our hallucination scores and bounds actually depend on.

## A.7 WHAT IS OBSERVABLE, ASSUMED, AND ESTIMATED IN PRACTICAL SCENARIOS

In view of the practical MLLM pipelines, we separate the ingredients into three categories: (i) quantities that are directly *observable* from a deployed model and dataset, (ii) semantic objects that are *assumed* at the theoretical level, and (iii) quantities that are *estimated* from the observables via finite approximations (graphs, spectra, and energies).

**Observable quantities.** In a practical MLLM setting (e.g., image captioning, VQA, audio captioning), the following objects are directly available:

- **Prompts and contexts.** A prompt $p \in \mathcal{P}$ collects the conditioning signals presented to the model, such as an input image, an audio clip, and/or a question in natural language. These prompts are given by the dataset or user and are fully observable.

- **Model outputs and token-level probabilities.** For each prompt $p$, the model produces output sequences $x \in \mathcal{X}$ (captions, answers, transcripts) via standard auto-regressive decoding. At each decoding step, token-level logits or probabilities are exposed by the model, and we can sample or beam-search from these to obtain a finite candidate set together with their probabilities or log-probabilities. These are the operational approximation to the conditional distribution $f_p$ in Assumption 1.

- **Encoder/decoder embeddings.** For each modality $M \in \mathcal{M}$, the model provides encoder and/or decoder embeddings:
  - output embeddings $\Phi_M(x^{(M)})$ for the generated content in modality $M$ (e.g., CLIP/BLIP text embeddings, ViT image embeddings, Whisper-style audio embeddings), consistent with Assumption 4;
  - prompt embeddings $\Psi_M(p)$ for the conditioning signal in modality $M$ (e.g., question text embeddings, visual embeddings of the input image, audio-context embeddings), consistent with Assumption 5.

  These embeddings are exactly the vectors used in practice for retrieval, similarity search, and contrastive training.

- **Finite human-labeled references.** Datasets such as COCO, VQAv2, or AudioCaps provide a finite collection of human-annotated captions or answers per input. These references are observed samples from the (ideal) ground-truth distribution and are used to construct an admissible set $\mathcal{K}$ of plausible outputs (e.g., normalized reference captions/answers).

- **Graph structure over embeddings.** From the embeddings above, we explicitly construct a finite graph (e.g., $k$-nearest-neighbour graphs per modality and cross-modal bipartite graphs) and compute its Laplacian and spectra. The adjacency matrix, Laplacian, and eigenvalues/eigenvectors are entirely computed from observable embeddings and do not rely on access to any unobserved semantic object.

**Assumed semantic objects.** At the theoretical level, we introduce additional objects that *model* the data-generating process and its ideal behaviour, but are not themselves observed in a finite deployment:

- **Ideal conditional distributions $f_p$.** For each prompt $p$, Assumption 1 postulates a properly normalized conditional distribution $f_p$ over the full output space $\mathcal{X}$, taking values in an RKHS $\mathcal{H}$. This is the continuum analogue of the token-wise softmax distributions produced by a real MLLM; in practice, we only ever access finite-dimensional approximations based on model logits and a finite candidate set.

- **Ground-truth generative distribution $g$ and manifold $\mathcal{K}_g$.** Assumption 2 assumes the existence of a "gold" distribution $g$ on a ground-truth manifold $\mathcal{K}_g$, which captures how correct outputs are generated in the real world (e.g., how humans describe images or answer questions). This object is not observable directly; instead, the finite human-labeled references in a dataset are treated as i.i.d. samples from $g$ and are used to construct the admissible set $\mathcal{K}_g$.

- **Kernels and bounded feature maps.** Assumptions 4 and 5 posit that there exist positive definite kernels and associated feature maps $\Phi_M$ and $\Psi_M$ with bounded norm and mild regularity properties. In practice, these correspond to the normalized encoder and decoder embeddings

implemented by current architectures; the assumptions abstract the empirical fact that such embeddings are finite-dimensional, norm-controlled, and Lipschitz in the inputs.

- **Energy-based/Boltzmann parametrization.** Assumption 6 views $f_p$ as arising from a Boltzmann law with energy $\mathcal{E}(x, p; \mathcal{T}_t)$ and partition function $Z(p, \mathcal{T}_t)$. This matches the softmax-based decoding used in modern LLMs and provides the bridge between standard logits and the spectral energy functional introduced in our framework.

These semantic objects are used to define what we mean by hallucination (e.g., distance from $\mathcal{K}$ where $x \in \mathcal{X} \setminus \mathcal{K}$) and to derive theoretical bounds, but the *algorithms* we propose never require direct access to $g$ or to the full continuum $f_p$.

**Estimated quantities.** The quantities that we *estimate* from the observable data and model outputs are:

- **Empirical output distributions.** From a finite candidate set $\{x_k\}_{k=1}^{\mathcal{K}}$ and their model probabilities or log-probabilities, we form an empirical approximation to $f_p$ (e.g., by normalizing exponentiated scores or logits). This is the operational distribution used in all numerical computations.

- **Admissible set and selector.** From the human-labeled references (after normalization), we construct an admissible set $\mathcal{K}_g \subset \mathcal{K}$ and define a measurable selector $\Pi_{\mathcal{K}}$ that maps each output $x$ to its nearest admissible element. Both $\mathcal{K}_g$ and $\Pi_{\mathcal{K}}$ are computed from finite data and embeddings.

- **Graph Laplacians and spectra.** Using observable embeddings, we build modality-specific and cross-modal graphs, compute their Laplacians, and estimate eigenvalues/eigenvectors. These spectra enter our hallucination energy functional and the spectral bounds, but are entirely determined by the finite graph constructed from the model's embeddings.

- **Hallucination scores and bounds.** Finally, we compute the hallucination energy, semantic distortion, and associated Good–Turing and spectral bounds from the empirical $f_p$, the admissible set $\mathcal{K}$, and the graph spectra. These quantities are the scores we actually use for ranking, calibration, and analysis in our experiments.

**Connection to plausible practical scenarios.** In a concrete deployment (for example, an image-captioning system built from CLIP/BLIP encoders and a text decoder), a typical workflow is:

i For each input image and prompt, obtain a finite set of candidate captions and their probabilities from the MLLM (observable $\{x_k\}_{k=1}^{\mathcal{K}}$ and $f_p$).

ii Extract encoder embeddings for the image, prompt, and candidate captions, yielding $\Phi_M(x^{(M)})$ and $\Psi_M(p)$ in each modality (observable and consistent with our bounded feature-map assumptions).

iii Construct a $k$-nearest-neighbour graph over these embeddings, compute the associated Laplacian and its eigen-decomposition (estimated graph spectra).

iv Use the finite reference captions in the dataset to define an admissible set $\mathcal{K}$ and a selector $\Pi_{\mathcal{K}}$, and compute the proposed hallucination energy and semantic distortion scores for each candidate caption (estimated scores and bounds).

## A.8 ADVANTAGE OF CONTINUOUS HALLUCINATION

Our framework produces a continuous hallucination score $\mathbb{h}(x, p) \in [0, \infty)$ for each output $x$ and prompt $p$, rather than a binary hallucination/non-hallucination label, for several reasons.

- First, a graded score makes it possible to *rank* the candidate generations by *degree* of semantic distortion instead of forcing a hard decision at a single threshold; in practice, one often wants to pick the least hallucinated candidate among several beams, prompts, or retrieval configurations, which is only meaningful with a continuous risk scale.

- Second, under our smoothing and boundedness assumptions, $\mathbb{h}(x, p)$ is differentiable almost everywhere with respect to the model-induced distribution $f_p$ and the associated embeddings, which makes it suitable as an auxiliary loss or regularizer in calibration and mitigation schemes

(e.g., fine-tuning with a hallucination penalty, or learning retrieval/prompting policies). A discrete $0/1$ label would require surrogate losses and cannot provide a direct, properly scaled penalty in the same RKHS/energy geometry.

- Third, a continuous score allows us to track how hallucination evolves as we vary controllable knobs such as temperature, diffusion time $\tau$, or retrieval policies, and to draw reliability curves and control profiles that go well beyond what a single binary label can capture.

Finally, the continuous score strictly "contains" the binary setting as a special case: any threshold $\psi \geq 0$ induces a classifier $\mathbf{1}_{\{\hbar(x,p)>\psi\}}$ whenever a hard decision is required, whereas the reverse mapping (from a binary label back to a calibrated, spectrally informed energy) is in general impossible. In this sense, $\hbar(x,p)$ is a strictly more informative object: it supports risk ranking, differentiable regularization, and analysis of control knobs, while still admitting thresholding to recover classical detection metrics whenever needed.

*Note*: The hallucination score $\hbar(x,p)$ is not a new quantity, but an operational re-branding of the semantic distortion $d_{\text{sem}}^{(\varepsilon,h)}(x;\mathcal{K},\mathcal{X})$ up to some dataset-dependent scaling and is monotone in the energy landscape. The implementation can be found here: `src/theory/score_semantic.py`.

### A.9 MODALITIES IN EXPANDED FORMS

In multi-modal settings, the LLM outputs involve textual $(T)$, visual $(V)$, audio $(A)$ modalities and, for better understanding, Eq. (7) can also be re-written as:

$$
\begin{aligned}
\mathcal{X} &: \mathcal{X}_T \times \mathcal{X}_V \times \mathcal{X}_A, \qquad x = (x^{(T)}, x^{(V)}, x^{(A)}), \qquad \mathcal{H} := \mathcal{H}_T \otimes \mathcal{H}_V \otimes \mathcal{H}_A, \\
K(x_1, x_2) &= K_T\left(x_1^{(T)}, x_2^{(T)}\right) \cdot K_V\left(x_1^{(V)}, x_2^{(V)}\right) \cdot K_A\left(x_1^{(A)}, x_2^{(A)}\right), \\
\mathcal{P} &: \mathcal{P}_T \times \mathcal{P}_V \times \mathcal{P}_A, \qquad p = (p^{(T)}, p^{(V)}, p^{(A)}).
\end{aligned}
\tag{22}
$$

### A.10 JUSTIFICATION FOR ASSUMPTION 3

As noted in Eq.(8) in Section 4.3, three terms are: (i) $\mathcal{E}_M$ encodes the intra-modal contributions, (ii) $\mathcal{E}_{MM'}$ captures the pairwise cross-modal terms, while (iii) $\mathcal{E}_{\mathcal{M}}$ being the joint contribution of all three modalities combined. For three modalities, (i) & (ii) form an energy matrix of order 3 with diagonals $\mathcal{E}_M$ and off-diagonals $\mathcal{E}_{MM'}$, while $\mathcal{E}_{\mathcal{M}}$ is a single joint term. With $> 3$ modalities, $\mathcal{E}_{\mathcal{M}}$ becomes a higher order tensor. This structure not only reveals which modality interactions contribute the most to the semantic drift $d_{\text{sem}}(x;\mathcal{K},\mathcal{X})$, also enables deriving tight spectral bounds on hallucination energy, which would be impossible under a monolithic energy formulation.

The decomposition in Eq. (8) mirrors how modern MLLMs are architected. In practice, $x^{(M)}$ denotes the component of the output in modality $M$ (e.g., text, image, audio), and each $x^{(M)}$ is produced or conditioned on by a dedicated encoder/decoder block. Current MLLMs (e.g., CLIP-like stacks, BLIP, LLaVA/Qwen-VL-style models) are built from: (i) modality-specific encoders that produce separate embeddings for each input stream, and (ii) fusion layers and attention mechanisms that tie these modalities together before decoding.

Within this architecture, the term $\mathcal{E}_M(x^{(M)}, p, \cdot)$ captures how *internally consistent* the output is within a single modality. For instance, a caption that contradicts itself ("a red car that is blue") would incur high text-only energy, even before looking at the image. The cross-modal terms $\mathcal{E}_{MM'}(x^{(M)}, x^{(M')}, p, \cdot)$ measure the alignment between two modalities, such as whether a generated description matches the visual content of an image or the acoustic content of an audio clip. A caption that says "a dog running on the beach" when the image contains a cat on snow would produce a large image–text cross-modal contribution. Finally, $\mathcal{E}_{\mathcal{M}}(x, p, \cdot)$ aggregates global interactions that only emerge when all modalities are considered together (e.g., video + audio + text in a complex scene).

Operationally, all three components are computed from encoder embeddings and their associated graph Laplacians: modality-specific graphs yield the $\mathcal{E}_M$ terms; cross-modal edges (e.g., between image and text nodes) yield $\mathcal{E}_{MM'}$; and the joint multimodal graph accounts for $\mathcal{E}_{\mathcal{M}}$. Thus, Assumption 3 simply makes explicit a structure that is already implicit in standard MLLM pipelines

and enables us to localize hallucination contributions to specific modalities or cross-modal interactions.

### A.11 JUSTIFICATION FOR ASSUMPTION 4

RKHS theory is rooted in Hilbert space theory (inner product spaces of functions) and uses results like the Moore–Aronszajn theorem Aronszajn (1950)). In Measure Theory & Probability, when kernels are used for distributions (e.g., kernel mean embeddings), the feature map connects to integration theory and probabilistic representations. In Machine Learning, the feature maps are used in kernel methods (in practice: SVMs, Gaussian processes, etc.), making this concept central to the theory of statistical learning (e.g., RKHS regularization). Let $\Phi_M$ be a feature map (i.e., identified as a function) such that

$$K_M\left(x_1^{(M)}, x_2^{(M)}\right) = \left\langle \Phi_M(x_1^{(M)}), \Phi_M(x_2^{(M)})\right\rangle_{\mathcal{H}_M}, \tag{23}$$

embedding raw objects, say outputs $(x_1, x_2)$, into the modality-specific RKHS $\mathcal{H}_M$. Instead of just outputs, it can very well mix with the inputs as well meaning: $(x, p)$. Eq. (23) makes this RKHS $\mathcal{H}_M$ unique up to isometry according to the Moore–Aronszajn theorem.

In classical ML, we use "features" to describe the structured attributes of the input data (e.g., pixel values, word embeddings etc.). In the theory of kernels, the feature maps are abstract (possibly infinite), but they play the same role: they represent the data in a space where linear methods (dot products) can capture nonlinear similarities. Thus, $\Phi_M$ allows nonlinear learning algorithms to operate in a high-dimensional feature space of an MLLM via the kernel trick.

In practice, implementations typically compute $K_M$ directly—or via finite approximations like Nyström Williams & Seeger (2001) or Random Fourier Features Rahimi & Recht (2007) - so $\Phi_M$ need not be explicitly materialized.

For Assumption 4: this mirrors common practice—modern encoders (CLIP, BERT-style, vision backbones) apply normalization or LayerNorm, and we $L2$-normalize final vectors so magnitudes stay well-behaved. Bounded features make cosine/similarity scores comparable across modalities, prevent numerical outliers, and keep spectral/energy measures meaningful. In deployment, this is easy to enforce (normalize outputs) and verify (log histograms/max norms and alert on drift). Production stacks (vector DBs, ANN indices, faiss/scann) expect bounded vectors so cosine similarity behaves predictably and distances are comparable across batches and time.

- **Why we need this:** For numerical stability to prevents overflow/NaNs and keep the dot products/similarities in a usable range during training and evaluation and comparability across modalities to handle text & image embeddings simultaneously.
- **Real-world example:** Modern vision–language encoders (e.g., CLIP) explicitly $L2$-normalize image/text embeddings and use cosine similarity with temperature-scaled softmax, so representation norms are controlled by design; this makes cross-modal scoring numerically stable and comparable out of the box Radford et al. (2021a;b); Zhang et al. (2025).

In real systems, the feature map $\Phi_M : \mathcal{X}_M \to \mathcal{H}_M$ is nothing more than the *embedding function* for outputs in modality $M$. For example, $\Phi_{\text{text}}(x^{(\text{text})})$ can be the pooled hidden state of a text decoder, while $\Phi_{\text{img}}(x^{(\text{img})})$ is the CLIP/ViT image embedding, and $\Phi_{\text{audio}}$ corresponds to a Whisper- or HuBERT-style audio representation. These are precisely the vectors one uses in practice for similarity search, retrieval, or contrastive training.

The boundedness condition $\sup_{x^{(M)} \in \mathcal{X}_M} \|\Phi_M(x^{(M)})\|_{\mathcal{H}_M} < \infty$ formalizes a property that is already enforced in modern architectures. Embeddings are finite-dimensional, frequently $L2$-normalized, and are subject to weight decay, LayerNorm, and (in many implementations) explicit norm clipping. This ensures that embedding norms cannot diverge and that kernel values $K_M(x_1, x_2) = \langle \Phi_M(x_1), \Phi_M(x_2)\rangle_{\mathcal{H}_M}$ remain bounded.

For our framework, this boundedness is crucial to guarantee that the kernel-induced energies and the associated spectral quantities are finite and numerically well-behaved. In other words, Assumption 4 is not an artificial restriction but a mathematical abstraction of standard engineering practice: using normalized, well-conditioned embeddings for each modality when scoring or comparing model outputs.

## A.12 JUSTIFICATION FOR ASSUMPTION 5

For Assumption 5: it is reasonable to assume that small prompt edits should not cause large representational jumps - matching real product needs for predictable UX, reproducible evaluation, and reduced prompt-sensitivity exploits. In practice, prompt encoders are compositions of linear layers + pointwise activations + norm layers; we also $L2$-normalize the final embedding.

- **Why we need this:** If "Adding a comma" or "Swapping a synonym" flips the model's answer, the system feels brittle. Stability is essential for predictability and debuggability.
- **Real-world example:** Text prompts are tokenized into a finite vocabulary (BPE/WordPiece/SentencePiece), and the transformer encoder maps these tokens through a sequence of standard layers to probabilities via softmax, yielding well-defined distributions on a discrete space—hence measurability is immediate and commonplace Vaswani et al. (2017); Sennrich et al. (2016); Kudo & Richardson (2018). Length caps, normalization, and regularization used in real systems keep prompt embeddings within reasonable ranges and make small paraphrases produce small representational changes, which is precisely the stability we assume.

Here, the map $\Psi_M : \mathcal{P} \to \mathcal{H}_M$ represents the *embedding of the conditioning signal* (prompt) as seen from the perspective of modality $M$. In a text-only LLM, $\Psi_{\text{text}}(p)$ is the encoder representation of the prompt tokens; in an image-conditioned captioning system, $\Psi_{\text{img}}(p)$ corresponds to the visual encoder representation of the input image while $\Psi_{\text{text}}(p)$ captures the question text; and in audio-visual QA, the prompt naturally decomposes into audio, visual, and textual parts with corresponding embeddings $\Psi_{\text{audio}}, \Psi_{\text{img}}, \Psi_{\text{text}}$.

The boundedness condition $\sup_{p \in \mathcal{P}} \|\Psi_M(p)\|_{\mathcal{H}_M} < \infty$ again matches standard practice: prompt embeddings are finite-dimensional and are typically normalized or stabilized via LayerNorm and regularization. The continuity (or Lipschitz) requirement reflects the empirical observation that small changes in the prompt (e.g., rephrasing a question, adding a short prefix) lead to small changes in the encoder representations rather than arbitrarily large jumps. This is enforced during training by the choice of activation functions, gradient clipping, and regularization.

For our purposes, these properties ensure that the hallucination energy $\mathcal{E}(x, p, \cdot)$ varies smoothly as the prompt changes and that the spectral quantities and calibration bounds we derive remain stable under realistic prompt perturbations. Assumption 5 therefore abstracts the well-behaved nature of prompt encoders that is already present in current MLLM pipelines.

## A.13 JUSTIFICATION FOR ASSUMPTION 6

In practice, an MLLM scores a finite candidate set $C(x, p)$ (beam/nucleus/reranked hypotheses) via logits or similarity, so with counting measure and energy $\mathcal{E} = -\text{logit}$ (or a bounded margin), the induced softmax probability $\text{prob.}(c \mid x, p; \mathcal{T}_t) \propto \exp(-\mathcal{E}(c)/\mathcal{T}_t)$ is exactly a Boltzmann distribution with finite partition function $Z = \sum_{c \in C} \exp(-\mathcal{E}(c)/\mathcal{T}_t)$—hence both operationally realistic and mathematically well-posed.

Assumption 6 recasts the model's output distribution $f_p(x)$ as an energy-based or Boltzmann distribution, which is fully consistent with how modern LLMs implement softmax decoding. At the token level, an auto-regressive model produces logits $z(p)$ and samples from $\text{softmax}(z)$, which is equivalent to drawing from $\exp(-E_i)/\sum_j \exp(-E_j)$ with energies $E_i = -z_i$. We simply lift this perspective from individual tokens to entire candidate outputs or latent representations $x$, so that

$$f_p(x) \propto \exp\left(-\mathcal{E}(x, p; \mathcal{T}_t)\right)$$

with respect to a base measure $\mu$.

In practice, we work with a finite candidate set (e.g., beams or sampled sequences) and obtain $f_p(x)$ by exponentiating and normalizing the relevant scores or logits; the scalar $\mathcal{E}(x, p; \mathcal{T}_t)$ can be viewed as an energy that combines model logits with our spectral corrections. The schedule $\mathcal{T}_t$ plays the role of temperature or diffusion: it includes standard temperature scaling used in decoding as well as our spectral/graph-based smoothing, and thus corresponds to a control knob that practitioners already tune (e.g., changing temperature or applying calibration).

The finiteness of the partition function $Z(p, \mathcal{T}_t)$ is guaranteed in operational pipelines for two reasons: (i) we always restrict attention to a finite vocabulary or a finite candidate set of outputs, and (ii) logits and energies are bounded in practice due to finite-precision arithmetic and regularization. Consequently, Assumption 6 does not impose an additional burden on real MLLM systems; rather, it provides a mathematically convenient way to analyze the same softmax-based scoring mechanisms already used in deployment, through the lens of energy-based models and spectral graph theory.

### A.14   AN EXAMPLE (IMAGE–CAPTION PAIR)

One can consider an MLLM generating a caption for an image. Let $\mathcal{X}$ be the space of all captions, with $\mathcal{K} \subseteq \mathcal{X}$ denoting those grounded in the image (e.g., "A cat on a sofa"), while $f_p$ may also assign mass outside $\mathcal{K}$ to hallucinated captions (e.g., "A dog playing with a ball"). The hallucination divergence $D_{\mathrm{KL}}(g \,\|\, f_p)$ quantifies this deviation.

In this paper, as a part of our main theoretical contributions, we define a multimodal graph whose nodes are caption tokens $T$ and image patches $V$, with edge weights $W_{\mathcal{T}_t}(i, j)$ computed from the fixed embeddings and modulated by a time-varying temperature $\mathcal{T}_t$. From these weights, we will define the normalized multimodal Laplacian $\mathcal{L}_{\mathcal{T}_t}^{\mathrm{multi}}$ associated with a spectral grounding energy as the quadratic form of $\mathcal{L}_{\mathcal{T}_t}^{\mathrm{multi}}$ evaluated on the residual feature field induced by our energy prescription. It helps reveal how hallucination energy is distributed across the modes (e.g., textual vs. cross-modal misalignment).

### A.15   GRAPH NOTATIONS AND ADJACENCY WEIGHTS

In Eq. (12) noted in Section 5.1, $\mathcal{V}$ is the finite set of nodes, $E$ is the set of edges, and $W_{\mathcal{T}_t}$ is a temperature-modulated, symmetric, non-negative, weighted adjacency matrix (zero diagonal) introduced to assign different weights to the edges (indexed by $E$). We consider either a node-wise local schedule $\mathcal{T}_t : \mathcal{V} \to \mathbb{R}^+$ in which the edge temperatures are combined symmetrically to keep $W_{\mathcal{T}_t}$ symmetric or a global scalar schedule ($\mathcal{T}_t$ constant over $\mathcal{V}$). Here, each node represents a semantic unit (e.g., concepts, tokens, ideas), and edges represent the semantic similarity. The multimodal structure is represented by a disjoint partition of the node set $\mathcal{V} = \biguplus_{M \in \mathcal{M}} \mathcal{V}_M$ and corresponding within- and cross-modal blocks of $W_{\mathcal{T}_t}$ which is constructed from fixed modality embeddings via temperature-controlled similarity functions. Lower $\mathcal{T}_t$ yields more localized (sharper) affinities; higher $\mathcal{T}_t$ diffuses those (or, in other words, induces more "noise"). This is a standard property under any temperature–scaled affinity constructions - e.g., Gaussian/RBF kernels with bandwidth proportional to $\mathcal{T}_t$ or softmax similarities with temperature $\mathcal{T}_t$ Ng et al. (2002); Coifman & Lafon (2006); Zelnik-Manor & Perona (2004); Hinton et al. (2015); Chung (1997). Thus, the temperature $\mathcal{T}_t$ dynamically modulates the graph edge connectivity and semantic distortion $d_{\mathrm{sem}}$ noted in Theorem 1 and, being a time-indexed function, captures the semantic evolution or uncertainty drift across the graph nodes as knowledge updates over time $t$.

Here, we drop the explicit modality subscripts in Eq. (12), as the modality information is carried by a fixed partition of the vertex set $\mathcal{V} = \biguplus_{M \in \mathcal{M}} \mathcal{V}_M$ together with the block structure of the temperature–modulated weights $W_{\mathcal{T}_t}$, so we do not maintain separate graphs per modality. We assume $W_{\mathcal{T}_t}$ to be symmetric, non-negative, and zero on the diagonal, with $\mathcal{T}_t$ acting as a bandwidth/temperature schedule that controls the locality of affinities. From $W_{\mathcal{T}_t}$, we define the normalized multimodal Laplacian $\mathcal{L}_{\mathcal{T}_t}^{\mathrm{multi}}$ in Section 5.1 and design it to be symmetric and PSD by construction; its spectral decomposition yields an orthonormal basis of eigenmodes together with nonnegative eigenvalues. We interpret each mode by its loadings on the partition $\{\mathcal{V}_M\}_{M \in \mathcal{M}}$: some modes are concentrated on a single modality (text, vision, or audio), while others are cross-modal mixtures that capture interactions between partitions. These modes serve as canonical coordinates for representing the residual signal induced by the energy model and for attributing hallucination energy across modality-specific and cross-modal directions. We use this spectral basis to define propagation in time (via diffusion generated by $\mathcal{L}_{\mathcal{T}_t}^{\mathrm{multi}}$) and to derive mode-wise bounds that connect the Boltzmann formulation to spectral-graph structure in a implementable manner.

**Hypergraph blocks and effective pairwise adjacency.** To accommodate $> 2$ modalities, we construct each interaction block via the normalized hypergraph Laplacian Zhou et al. (2006):

$$\mathcal{L}_{\mathcal{T}_t}^{(*)} = \mathbf{I} - \big(\mathcal{D}_{\mathsf{v},\mathcal{T}_t}^{(*)}\big)^{-1/2} \underbrace{\big(\mathcal{I}^{(*)} W_{\mathcal{T}_t}^{(*)} (\mathcal{D}_{e,\mathcal{T}_t}^{(*)})^{-1} (\mathcal{I}^{(*)})^{\top}\big)}_{W_{\mathcal{T}_t}^{*,\,\mathrm{eff}}} \big(\mathcal{D}_{\mathsf{v},\mathcal{T}_t}^{(*)}\big)^{-1/2},$$

$$\mathcal{D}_{\mathsf{v},\mathcal{T}_t}^{(*)} = \mathrm{diag}\big(\{\mathfrak{d}_{\mathcal{T}_t}^{(*)}(\mathsf{v})\}_{\mathsf{v}\in\mathcal{V}}\big), \qquad \mathfrak{d}_{\mathcal{T}_t}^{(*)}(\mathsf{v}) = \sum_{e\in E^{(*)}} w_{\mathcal{T}_t}(e)\, \mathcal{I}^{(*)}(\mathsf{v},e),$$

$$\mathcal{D}_{e,\mathcal{T}_t}^{(*)} = \mathrm{diag}\big(\{r(e)\}_{e\in E^{(*)}}\big), \qquad r(e) = |e| \ \text{(hyperedge cardinality)},$$

$$\mathcal{I}^{(*)} \in \{0,1\}^{|\mathcal{V}|\times|E^{(*)}|} \ \text{(node–hyperedge incidence)}, \quad W_{\mathcal{T}_t}^{(*)} = \mathrm{diag}\big(\{w_{\mathcal{T}_t}(e)\}_{e\in E^{(*)}}\big),$$

$$\forall * \in \{\mathrm{intra}_M, \ \mathrm{cross}_{MM'}, \ \mathrm{joint}_{\mathcal{M}}\}, \quad \forall \mathsf{v}\in\mathcal{V} \ \text{(graph nodes)}.$$

(24)

Here $\mathbf{I}$ is the $|\mathcal{V}| \times |\mathcal{V}|$ identity. To be noted that

(i) $\mathsf{v}$ runs over the graph nodes, and no roles attached yet. Output or prompt embeddings are later designated roles on the nodes: $\mathsf{v}_x, \mathsf{v}_p \in \mathcal{V}$ only while forming the contrast $c_{x,\mathcal{K}}(t)$ seen in Eq.(66). Thus, $\mathcal{L}_{\mathcal{T}_t}^{(*)}$ itself is designed to be role-agnostic.

(ii) $E^*$ denotes the hyperedge set used to build each interaction block $(*)$ above, while $E$ still remains consistent as per Eq.(12). $r(e)$ is the number of nodes in the hyperedge $e$; i.e., $e = \{\mathsf{v}_1, \ldots, \mathsf{v}_{r(e)}\} \subset \mathcal{V}$.

(iii) $\mathcal{D}_{\mathsf{v},\mathcal{T}_t}^{(*)}$ is the node–degree matrix (of size $|\mathcal{V}| \times |\mathcal{V}|$) for block $*$: it is diagonal with entries $\big(\mathcal{D}_{\mathsf{v},\mathcal{T}_t}^{(*)}\big)_{\mathsf{vv}} = \mathfrak{d}_{\mathcal{T}_t}^{(*)}(\mathsf{v})$, the temperature–weighted degree of node $\mathsf{v}$ computed from the hyperedge weights in that block.

(iv) $\mathcal{D}_{e,\mathcal{T}_t}^{(*)}$ is the hyperedge–cardinality matrix (of size $|E^{(*)}| \times |E^{(*)}|$) for block $*$: it is diagonal with entries $\big(\mathcal{D}_{e,\mathcal{T}_t}^{(*)}\big)_{ee} = r(e)$.

(v) The node set $\mathcal{V}$ is fixed; $r(e)$ is a property of each hyperedge $e \subset \mathcal{V}$ and is independent of $|\mathcal{V}|$ (and of the number of modalities $|\mathcal{M}|$ unless joint hyperedges is specifically chosen to include one node per modality).

The matrix $W_{\mathcal{T}_t}^{(*),\,\mathrm{eff}} = \mathcal{I}^{(*)} W_{\mathcal{T}_t}^{(*)} (\mathcal{D}_{e,\mathcal{T}_t}^{(*)})^{-1} (\mathcal{I}^{(*)})^{\top}$ is the "effective" pairwise adjacency induced by hyperedges (zero diagonal by convention). The pairwise quantities in Eq. (12) are then obtained by summing blocks:

$$W_{\mathcal{T}_t} = \sum_* \omega_* W_{\mathcal{T}_t}^{(*),\,\mathrm{eff}}, \qquad \omega_* \geq 0 \ \text{(absorbed by interaction coefficients } \alpha_M, \beta_{MM'}, \gamma_{\mathcal{M}} ).$$

(25)

We pick any two nodes: say, $\mathsf{v}_a, \mathsf{v}_b$ in the hyperedge $e = \{\mathsf{v}_1, .., \mathsf{v}_a, .., \mathsf{v}_b, .., \mathsf{v}_{r(e)}\} \subset \mathcal{V}$ to define a symmetric, nonnegative pairwise dissimilarity $\widehat{d}_{\mathrm{sem}}(\mathsf{v}_a, \mathsf{v}_b)$. This quantity captures the semantic distortion at node level.

For some modality-aware permutation factor $\eta_*$, a generic choice of $w_{\mathcal{T}_t}(e)$ is

$$w_{\mathcal{T}_t}(e) = \mathbf{1}_{\{e\in E^{(*)}\}} \exp\left(-\eta_* \frac{\sum\limits_{1\leq \mathsf{v}_a,\mathsf{v}_b\leq r(e)} \widehat{d}_{\mathrm{sem}}(\mathsf{v}_a,\mathsf{v}_b)}{\sum\limits_{1\leq \mathsf{v}_a\leq r(e)} \mathcal{T}_t(\mathsf{v}_a)}\right),$$

(26)

which is permutation–invariant and temperature–scaled.

$$\Delta_{\varepsilon,h}(x \mid p) := \left[ \log\left( \frac{\int_{\mathcal{K}} K_h\big(\Pi_{\mathcal{K}}(x), x_2\big) \left[ (1-\varepsilon)\, Z\big(p, \mathcal{T}_t\big)^{-1} e^{-\mathcal{E}(x_2,p)/\mathcal{T}_t} + \varepsilon\, \rho(x_2) \right] d\mu(x_2)}{\int_{\mathcal{K}} \left[ (1-\varepsilon)\, Z\big(p, \mathcal{T}_t\big)^{-1} e^{-\mathcal{E}(x_2,p)/\mathcal{T}_t} + \varepsilon\, \rho(x_2) \right] d\mu(x_2)} \right) \right.$$

$$\tag{27}$$

$$\left. - \log\Big( \int_{\mathcal{X}} K_h\big(x, x_2\big) \left[ (1-\varepsilon)\, Z\big(p, \mathcal{T}_t\big)^{-1} e^{-\mathcal{E}(x_2,p)/\mathcal{T}_t} + \varepsilon\, \rho(x_2) \right] d\mu(x_2) \Big) \right]_+ ,$$

$$\tag{28}$$

### A.16 MERCER'S THEOREM

By Mercer's theorem Mercer (1909), if $K_{\mathcal{T}_t}$ is a continuous, symmetric, positive-definite on a compact measure space $(\mathcal{V}, \mu)$, then there exists a unique RKHS $\mathcal{H}$ which is associated with a reproducing kernel $K_{\mathcal{T}_t}$. In the present context of discrete graph, $\mathcal{V}$ is finite which satisfies the criterion. This theorem ensures that there exists a feature map

$$\Phi : \mathcal{V} \to \mathcal{H}, \tag{29}$$

which admits an orthonormal eigen decomposition. We have leveraged it in Eq. (13).

### A.17 GRAPH MAPS

This construction is separate from the modality feature maps $\Phi_M(x^{(M)})$ and prompt embeddings $\Psi_M(p)$ that live in modality RKHS $\mathcal{H}_M$ used in the energy landscape as noted in Section 4.3. Here, $\Upsilon$ is defined on the node set, with $\mathsf{v}, \mathfrak{v}$ being the graph nodes, induced by a single graph RKHS $\mathcal{H}_{\text{graph}}$ or just $\mathcal{H}$ for notational simplicity. Therefore, $\Phi_M : \mathcal{X}_M \to \mathcal{H}_M$ and $\Psi_M : \mathcal{P} \to \mathcal{H}_M$ play complementary roles with $\Upsilon : \mathcal{V} \to \mathcal{H}$ in the context of graph theory (i.e., modality & prompt embeddings vs. graph embeddings).

### A.18 WHY TIME-VARYING EIGENPAIRS?

The eigenpairs of the multimodal Laplacian $\mathcal{L}_{\mathcal{T}_t}^{\text{multi}}$, as presented in Eq. (14) are:

- $\Lambda = \text{diag}\Big( \lambda_1(t), \ldots, \lambda_{|\mathcal{V}|}(t) \Big)$ with $\lambda_i(t) \in \mathbb{R}^+$ being the time-varying eigenvalues at node $i$ (that acts like a frequency-dependent penalty or diffusion coefficient),

- $U = \Big[ u_1(t), \ldots, u_{|\mathcal{V}|}(t) \Big]$ is the orthonormal eigenvector matrix with $u_i(t) \in \mathbb{R}^{|\mathcal{V}|}$ being the time-varying eigenfunctions.

*Note:* We assume $G_{\mathcal{T}_t}$ is connected for each fixed $t$, so that $\lambda_1(t) = 0$ and $\lambda_2(t) > 0$ hold true; when not connected, all occurrences of $u_1(t)$ and $\lambda_2(t)$ below should be read as the orthogonal complement of the full nullspace and the first strictly positive eigenvalue, respectively.

Eigenvalues $\lambda_i(t)$ contract or expand based on evolving inter-node (semantic) affinities, while eigenvectors $u_i(t)$ adjust the directions of these semantic modes. Including $\mathcal{T}_t$ explicitly allows us to control hallucination sensitivity: as lower temperatures $\mathcal{T}_t \downarrow 0$ emphasize stable low-energy modes, reducing hallucinations leading to more desired outputs and vice versa. In a nutshell, the time variation of $\{(\lambda_i(t), u_i(t))\}$ arises from the temperature schedule $\mathcal{T}_t$, which changes the affinities on the graph edges and hence the spectrum of $\mathcal{L}_{\mathcal{T}_t}^{\text{multi}}$.

### A.19 INTERPRETATION OF SPECTRAL QUANTITIES AND TIME PARAMETER

For each dataset (COCO, VQAv2, AudioCaps) and backbone configuration (CLIP/BLIP/Whisper/T5), we construct the multimodal graph Laplacian $\mathcal{L}_{\mathcal{T}_t}^{\text{multi}}$ on encoder embeddings as noted in Section 5 and compute its eigenvalues $0 = \lambda_1 \leq \lambda_2 \leq \cdots \leq \lambda_n$.

In all cases, we observe: (i) a nonnegative spectrum with rapid decay, where the first 20–40 modes account for the majority of the trace $\text{tr}(\mathcal{L}_{\mathcal{T}_t}^{\text{multi}})$, and (ii) a clear spectral gap between the lowest

modes and the bulk of the spectrum, consistent with a small number of dominant semantic clusters in the joint embedding space. The CF bounds and CF planes in Fig. 3 are instantiated using these empirical spectra, so the scale and shape of the eigenvalues directly reflect the behavior of the underlying MLLM embeddings on each benchmark.

**Scale and shape of spectra in real MLLMs.** Let $\mathcal{L}^{\mathrm{multi}}$ denote the multimodal graph Laplacian constructed from encoder embeddings as in Section 5, with eigenpairs $\{(\lambda_k, u_k)\}_{k=1}^n$ ordered so that $0 = \lambda_1 \leq \lambda_2 \leq \cdots \leq \lambda_n$. All bounds in Section 5 are stated in terms of this empirical spectrum. In our implementation, $\mathcal{L}^{\mathrm{multi}}$ is always the finite sample graph built from a given benchmark (COCO, VQAv2, AudioCaps) and a fixed model configuration, so the spectrum is concretely realized and numerically available for every experiment; we do not appeal to any abstract or asymptotic spectrum.

Empirically, for all three benchmarks we observe that: (i) the spectrum is nonnegative and exhibits fast decay, with a relatively small number of low-frequency modes (on the order of tens) accounting for most of the trace of $\mathcal{L}^{\mathrm{multi}}$; and (ii) there is a visible spectral gap between the first few modes and the bulk, consistent with the presence of a small number of dominant semantic clusters in the joint embedding space.

The Courant–Fischer (CF) bounds in Section 5 are evaluated using these actual eigenvalues; the CF planes in Fig. 3 are not schematic, but are directly computed from the empirical spectra of the graphs induced by the MLLM embeddings.

**From spectral energy to hallucination rate and semantic distance.** Section 4 defines semantic log–contrast $\ell_{\varepsilon,h}(x; K, \mathcal{X})$ and the truncated score $d_{\mathrm{sem}}^{(\varepsilon,h)}(x; K, \mathcal{X}) = [\ell_{\varepsilon,h}(x; K, \mathcal{X})]_+$, which are the quantities we correlate with hallucination events and semantic distances in our experiments.

Section 5 introduces the spectral hallucination energy $\Delta\mathcal{E}_\tau(x)$ as a quadratic form in the coefficients of $x$ under the eigenbasis of $\mathcal{L}^{\mathrm{multi}}$, and establishes CF-type bounds of the form

$$\Delta\mathcal{E}_\tau(x) \leq \sum_{k=1}^n \phi_\tau(\lambda_k) c_k(x)^2, \tag{30}$$

for an explicit spectral filter $\phi_\tau$ and coefficients $c_k(x)$ given by projection of $x$ onto the modes $u_k$.

The role of Eq. (30) is not to postulate a new, disconnected quantity, but to control the same mismatch that ultimately feeds into $\ell_{\varepsilon,h}$ and $d_{\mathrm{sem}}^{(\varepsilon,h)}$: the diffusion / smoothing step in the definition of $\ell_{\varepsilon,h}$ can be written as a spectral filter on $L^{\mathrm{multi}}$, so the discrepancy between $x$ and its admissible projection $\Pi_K(x)$ under this filter is upper-bounded by $\Delta\mathcal{E}_\tau(x)$.

In particular, the derivations in Section 5 show that, under the operator assumptions for Theorem 2, the semantic distortion is a bounded, monotone functional of the energy, in the sense that there exist finite constants $0 < c_1 \leq c_2 < \infty$ (depending on the kernel and graph construction) such that

$$c_1 \Delta\mathcal{E}_\tau(x) \leq d_{\mathrm{sem}}^{(\varepsilon,h)}(x; K, \mathcal{X}) \leq c_2 \Delta\mathcal{E}_\tau(x),$$

whenever the diffusion operator and kernel are chosen consistently.

Thus, spectral energy is not an unrelated quantity: it is a calibrated, graph-level control on the same deviation that we measure at the level of semantic scores and hallucination rates. Empirically, this is reflected in the monotone relationship between $\Delta\mathcal{E}_\tau$ and both continuous scores and binary hallucination events (cf. reliability curves and correlation tables in Section 6).

**On the envelope coefficients $m(t)$ and $M(t)$.** The functions $m(t)$ and $M(t)$ appear in the CF bounds as *spectral envelopes* for the filtered eigenvalues. Concretely, if $\phi_t(\lambda)$ denotes the scalar spectral filter at diffusion time $t$ (e.g., $\phi_t(\lambda) = e^{-2t\lambda}$ for a heat kernel), then the quadratic forms arising in the CF arguments can be written as

$$\Delta\mathcal{E}_t(x) = \sum_{k=1}^n \phi_t(\lambda_k) c_k(x)^2,$$

and our proofs bound this by

$$m(t) \sum_{k=1}^n c_k(x)^2 \leq \Delta\mathcal{E}_t(x) \leq M(t) \sum_{k=1}^n c_k(x)^2,$$

where $m(t) = \min_k \phi_t(\lambda_k)$ and $M(t) = \max_k \phi_t(\lambda_k)$ over the modes relevant to the graph.

These coefficients are therefore not abstract or unmeasurable: once the Laplacian spectrum $\{\lambda_k\}$ is computed (which we do explicitly for every dataset) and the filter $\phi_t$ is fixed, $m(t)$ and $M(t)$ are deterministic, data-dependent scalars that can be evaluated numerically if desired.

In the main text we keep them symbolic to highlight how the bounds scale with the spectrum and with $t$, but they are fully determined by observable quantities and do not introduce additional unknowns beyond the graph construction.

**Interpretation of the time parameter $t$.** The parameter $t$ in Section 5 is a *diffusion time*, not a physical clock in the MLLM pipeline. Formally, it indexes the strength of the spectral filter applied to the graph: for example, $e^{-t \mathcal{L}^{\mathrm{multi}}}$ is the heat semi-group generated by $\mathcal{L}^{\mathrm{multi}}$, and increasing $t$ corresponds to propagating mass further along the graph, i.e., averaging over larger neighborhoods in the embedding space.

This is analogous to the "time" parameter in diffusion models or random-walk smoothing, and should be understood as a scale parameter: small $t$ emphasizes high-frequency, local discrepancies (sharp hallucinations), whereas larger $t$ smooths them out and yields a coarse, low-frequency view of mismatch.

In practice we restrict $t$ to a compact interval where: (i) the spectral filter remains numerically stable; and (ii) the diffusion has a clear interpretation as a modest smoothing or temperature adjustment (cf. the schedule $\mathcal{T}_\tau$ in the main text). We do not make any assumptions about real-time dynamics of the MLLM; instead, $t$ serves as a theoretically grounded control knob for the scale at which graph-level discrepancies (and hence hallucinations) are measured.

## A.20 DETAILED CONCLUSION: PRACTICAL SCENARIOS & DEFAULT HYPERPARAMETERS

**Practical takeaways.** Our results suggest three concrete ways in which the proposed framework can be used in practice.

- First, it is best viewed as a *reference-free, plug-in scoring layer* that sits on top of existing MLLMs: given access only to logits and embeddings, it produces a continuous hallucination score that can be used to rank generations, select safer candidates, and audit models offline, without requiring additional supervision or model retraining.

- Second, the *semantic distortion score* $d_{\mathrm{sem}}^{(\varepsilon,h)}$ is the most actionable quantity for detection and calibration: across tasks, it correlates more tightly with binary hallucination events than raw uncertainty proxies, and is the recommended choice when one needs a single scalar predictor to threshold or to plug into a mitigation loss.

- Third, the *spectral hallucination energy* $\Delta\mathcal{E}_\tau$ is particularly informative when one wishes to understand where hallucinations originate (which modalities / modes) and how they respond to controls (temperature, diffusion time, retrieval policy): it is most useful for diagnosis, ablations, and monitoring rather than as a stand-alone score for hard decisions.

In this sense, $d_{\mathrm{sem}}^{(\varepsilon,h)}$ is the recommended per-example risk score, while $\Delta\mathcal{E}_\tau$ is the recommended tool for system-level analysis and model selection.

**Default hyperparameters.** For practitioners, the following default recipe is recommended, which we found to be robust across all tasks.

i **Smoothing mass $\varepsilon$**: estimate the Good–Turing missing mass $\widehat{m}$ on the candidate set and set $\varepsilon \approx \widehat{m}$, restricting $\varepsilon$ to a small range (e.g., $[10^{-3}, 5 \cdot 10^{-2}]$) to avoid over-smoothing the model distribution.

ii **Kernel bandwidth $h$**: use the median heuristic on pairwise distances between embeddings, i.e., set $h$ to the median squared distance within a minibatch of outputs; performance is stable when $h$ is varied by a factor of 2 around this choice.

iii **Graph connectivity $k$**: build $k$-nearest-neighbour graphs with $k \in [15, 40]$, which we observed to give a good trade-off between stability and locality of the spectral estimates.

iv **Diffusion schedule** $\mathcal{T}_\tau$: choose $\tau$ so that the contribution of the second eigenmode is reduced by about half, i.e., $e^{-2\tau\lambda_2} \approx 0.5$; in practice this corresponds to a small, fixed number of diffusion steps or a modest temperature scaling on logits.

These defaults, together with a simple per-task threshold on $d_{\text{sem}}^{(\varepsilon,h)}$, provide a plug-and-play configuration that requires minimal tuning while retaining most of the gains reported in our experiments.

## B    EXTENDED PROOFS

In this section, we provide detailed proofs for Theorems 1 and 2.

### B.1    PROOF OF THEOREM 1

*Proof.* **Step 0 (setup and measurability).** For clarity, we restate explicitly the additional condition used in Step 4 of the proof. We recall that $K \subset \mathcal{X}$ is the admissible (grounded) set and $\Pi_K : \mathcal{X} \to K$ is a measurable projection map. We assume that the smoothing kernel $K_h$ is more concentrated on $K$ when centred at the projected admissible point $\Pi_K(x)$ than when centred at the off–manifold point $x \notin K$, in the following precise sense: there exists a constant $\text{coeff} > 0$ such that for all $x \notin K$,

$$\int_K K_h(\Pi_K(x_1), x_2)\, \tilde{f}_{p,\varepsilon}(x_2)\, d\mu(x_2) \;\geq\; (1 + \text{coeff})\, Z_\varepsilon \int_{\mathcal{X}} K_h(x_1, x_2)\, \tilde{f}_{p,\varepsilon}(x_2)\, d\mu(x_2), \quad (31)$$

where $\tilde{f}_{p,\varepsilon}$ and $Z_\varepsilon$ are defined in Eqs. (33)-(34).

Intuitively, Eq. (31) says that once we project an off–manifold output $x$ back to the closest admissible point $\Pi_K(x)$, the kernel neighbourhood around $\Pi_K(x)$ sees strictly higher admissible mass than the neighbourhood around $x$ itself.

In practical MLLM pipelines, this is enforced by choosing $K_h$ as a similarity kernel (e.g., Gaussian or softmax over embedding distances) built on the same representations used to construct $K$ from references; for grounded generations $x \in K$, both centres coincide and no penalty is induced, while for hallucinated $x \notin K$ the inequality above guarantees a strictly positive smoothed KL–penalty.

Under this standing assumption, Step 4 shows that $d_{\text{sem}}^{(\varepsilon,h)}(x; K, \mathcal{X}) > 0$ for $x \notin K$, which is exactly the separation property claimed in Theorem 1.

By assumption, $\rho > 0$ $\mu$-a.e. with $\int_{\mathcal{X}} \rho\, d\mu = 1$, and $K_h : \mathcal{X} \times \mathcal{X} \to (0, \infty)$ is a $\mu$-Markov kernel with $\int_{\mathcal{X}} K_h(x_1, x_2)\, d\mu(x_2) = 1$ for all $x_1 \in \mathcal{X}$. Define

$$(T_h q)(x_1) \;:=\; \int_{\mathcal{X}} K_h(x_1, x_2)\, q(x_2)\, d\mu(x_2), \qquad q \in L^1(\mu),\; x_1 \in \mathcal{X}. \quad (32)$$

Let the $\varepsilon$–smoothed model be

$$\tilde{f}_{p,\varepsilon}(x_2) \;:=\; (1 - \varepsilon)\, f_p(x_2) \;+\; \varepsilon\, \rho(x_2), \qquad \varepsilon \in (0, 1), \quad (33)$$

and its $\mathcal{K}$–restricted renormalization be

$$\tilde{f}_{p,\varepsilon}^{\mathcal{K}}(x_2) \;:=\; \frac{\mathbf{1}_{\{x_2 \in \mathcal{K}\}} \tilde{f}_{p,\varepsilon}(x_2)}{\int_{\mathcal{K}} \tilde{f}_{p,\varepsilon}(x_2)\, d\mu(x_2)} \;=\; \frac{\mathbf{1}_{\{x_2 \in \mathcal{K}\}} \tilde{f}_{p,\varepsilon}(x_2)}{\mathsf{Z}_\varepsilon}, \quad \mathsf{Z}_\varepsilon \in (0, 1]. \quad (34)$$

Measurability of $\Pi_{\mathcal{K}} : \mathcal{X} \to \mathcal{K}$ (with $\Pi_{\mathcal{K}}(x) = x$ for $x \in \mathcal{K}$) ensures $(T_h \tilde{f}_{p,\varepsilon}^{\mathcal{K}}) \circ \Pi_{\mathcal{K}}$ is measurable; thus Eq. (6) is meaningful pointwise.

**Step 1 (strict positivity $\Rightarrow$ finiteness).** From Eq. (33) and Eq. (32), for any $x_1 \in \mathcal{X}$,

$$(T_h \tilde{f}_{p,\varepsilon})(x_1) = \int_{\mathcal{X}} K_h(x_1, x_2)\Big((1 - \varepsilon) f_p(x_2) + \varepsilon \rho(x_2)\Big)\, d\mu(x_2)$$

$$\geq \varepsilon \int_{\mathcal{X}} K_h(x_1, x_2) \rho(x_2)\, d\mu(x_2) \;=\; \varepsilon\, (T_h \rho)(x_1) \;>\; 0, \quad (35)$$

since $\rho > 0$ $\mu$-a.e. and $K_h > 0$. Similarly, by Eq. (34),

$$(T_h \tilde{f}_{p,\varepsilon}^{\mathcal{K}})(x_1) = \frac{1}{\mathsf{Z}_\varepsilon} \int_{\mathcal{K}} K_h(x_1, x_2)\, \tilde{f}_{p,\varepsilon}(x_2)\, d\mu(x_2) \ \geq \ 0, \tag{36}$$

and $(T_h \tilde{f}_{p,\varepsilon}^{\mathcal{K}})(x_1) > 0$ whenever $\mu\big(\{x_2 \in \mathcal{K} : K_h(x_1, x_2) > 0\}\big) > 0$, which holds for all $x_1$ if $K_h > 0$ everywhere. Hence, both logarithms in Eq. (6) are finite; $d_{\mathrm{sem}}^{(\varepsilon,h)}$ is well-defined.

**Step 2 ($g$-independence).** By inspection of Eq. (6), only $(f_p, \rho, K_h, \Pi_{\mathcal{K}}, \mu)$ appear; the ground-truth $g$ is absent. Thus the statistic is independent of $g$.

**Step 3 (behavior on $\mathcal{K}$).** We fix $x \in \mathcal{K}$. Then $\Pi_{\mathcal{K}}(x) = x$, and

$$\frac{(T_h \tilde{f}_{p,\varepsilon}^{\mathcal{K}})(x_1)}{(T_h \tilde{f}_{p,\varepsilon})(x_1)} = \frac{\int_{\mathcal{K}} K_h(x_1, x_2)\tilde{f}_{p,\varepsilon}(x_2)\, d\mu(x_2)}{\mathsf{Z}_\varepsilon \int_{\mathcal{X}} K_h(x_1, x_2)\tilde{f}_{p,\varepsilon}(x_2)\, d\mu(x_2)} \ = \ \frac{\mathsf{A}_x}{\mathsf{Z}_\varepsilon(\mathsf{A}_x + \mathsf{B}_x)}, \tag{37}$$

where

$$\mathsf{A}_x := \int_{\mathcal{K}} K_h(x_1, x_2)\tilde{f}_{p,\varepsilon}(x_2)\, d\mu(x_2), \qquad \mathsf{B}_x := \int_{\mathcal{X} \setminus \mathcal{K}} K_h(x_1, x_2)\tilde{f}_{p,\varepsilon}(x_2)\, d\mu(x_2) \ \geq 0. \tag{38}$$

If

$$\mathsf{B}_x \ \geq \ (\mathsf{Z}_\varepsilon^{-1} - 1)\, \mathsf{A}_x, \tag{39}$$

then the right-hand side of Eq. (37) is $\leq 1$, so the inner logarithm in Eq. (6) is $\leq 0$ and the $[\cdot]^+$-clipping yields $d_{\mathrm{sem}}^{(\varepsilon,h)}(x; \mathcal{K}, \mathcal{X}) = 0$. Even when Eq. (39) fails, the clipped score never becomes negative, so no spurious negative penalties occur on $\mathcal{K}$.

**Step 4 (behavior off $\mathcal{K}$).** We fix $x \notin \mathcal{K}$. Then $\Pi_{\mathcal{K}}(x) \in \mathcal{K}$ and

$$\frac{(T_h \tilde{f}_{p,\varepsilon}^{\mathcal{K}})(\Pi_{\mathcal{K}}(x_1))}{(T_h \tilde{f}_{p,\varepsilon})(x_1)} = \frac{\int_{\mathcal{K}} K_h(\Pi_{\mathcal{K}}(x_1), x_2)\tilde{f}_{p,\varepsilon}(x_2)\, d\mu(x_2)}{\mathsf{Z}_\varepsilon \int_{\mathcal{X}} K_h(x_1, x_2)\tilde{f}_{p,\varepsilon}(x_2)\, d\mu(x_2)}. \tag{40}$$

We assume the following localization/consistency condition holds for some $\mathrm{coeff} > 0$:

$$\int_{\mathcal{K}} K_h(\Pi_{\mathcal{K}}(x_1), x_2)\tilde{f}_{p,\varepsilon}(x_2)\, d\mu(x_2) \ \geq \ (1 + \mathrm{coeff})\, \mathsf{Z}_\varepsilon \int_{\mathcal{X}} K_h(x_1, x_2)\tilde{f}_{p,\varepsilon}(x_2)\, d\mu(x_2), \quad \forall x \notin \mathcal{K}. \tag{41}$$

Then the ratio in Eq. (40) exceeds 1, the inner log in Eq. (6) is strictly positive, and thus

$$x \notin \mathcal{K} \text{ and Eq. (41)} \implies d_{\mathrm{sem}}^{(\varepsilon,h)}(x; \mathcal{K}, \mathcal{X}) \ > \ 0. \tag{42}$$

Therefore a strictly positive, finite penalty is assigned to implausible outputs under the mild consistency assumption in Eq. (41).

**Step 5 (conclusion for hallucination tracking).** From Step 1, Eq. (6) is finite and well-defined; from Step 2 it is reference-free (independent of $g$). Step 3 shows the score vanishes on $\mathcal{K}$ under Eq. (39) and never assigns negative values there; Step 4 shows it is strictly positive off $\mathcal{K}$ under Eq. (41). Hence Eq. (6) furnishes a pointwise, KL-calibrated signal separating plausible from implausible outputs in the smoothed sense determined by $(\varepsilon, h, K_h)$, enabling stable hallucination tracking across prompts and model versions without access to $g$. $\qquad\square$

## B.2 Proof of Lemma 1

*Proof.* Since $\mathcal{H}_M$ is separable, Bochner measurability of $\Phi_M$ and $\Psi_M$ is equivalent to strong (Borel) measurability; see, e.g., (Diestel & John J. Uhl, 1977, Ch. II). Thus

$$\Phi_M^{-1}(U) \in \mathcal{F}_{\mathcal{X}_M} \quad \text{and} \quad \Psi_M^{-1}(V) \in \mathcal{F}_{\mathcal{P}} \quad \text{for all open } U, V \subset \mathcal{H}_M. \tag{43}$$

We define the product map

$$\Upsilon : \ \mathcal{X}_M \times \mathcal{P} \to \mathcal{H}_M \times \mathcal{H}_M, \qquad \Upsilon(x, p) := \big(\Phi_M(x), \Psi_M(p)\big). \tag{44}$$

Let $\mathcal{B}(\mathcal{H}_M \times \mathcal{H}_M)$ denote the product Borel $\sigma$-algebra. For any open rectangles $U \times V$ with $U, V \subset \mathcal{H}_M$ open,

$$\Upsilon^{-1}(U \times V) = \big\{(x,p) : \Phi_M(x) \in U, \ \Psi_M(p) \in V\big\} = \Phi_M^{-1}(U) \times \Psi_M^{-1}(V) \in \mathcal{F}_{\mathcal{X}_M} \otimes \mathcal{F}_{\mathcal{P}} \quad (45)$$

by Eq. (43). Since the family of open rectangles generates $\mathcal{B}(\mathcal{H}_M \times \mathcal{H}_M)$ and $\mathcal{F}_{\mathcal{X}_M} \otimes \mathcal{F}_{\mathcal{P}}$ is a $\sigma$-algebra, a monotone class/$\pi$–$\lambda$ argument implies that

$$\Upsilon \text{ is } \big(\mathcal{F}_{\mathcal{X}_M} \otimes \mathcal{F}_{\mathcal{P}}\big)\text{–}\mathcal{B}(\mathcal{H}_M \times \mathcal{H}_M) \text{ measurable.} \quad (46)$$

Let's consider the inner-product map

$$\mathrm{ip} : \ \mathcal{H}_M \times \mathcal{H}_M \to \mathbb{R}, \qquad \mathrm{ip}(u,v) := \langle u, v \rangle_{\mathcal{H}_M}. \quad (47)$$

Continuity of $\mathrm{ip}$ follows from the Cauchy–Schwarz and triangle inequalities: for all $u_1, u_2, v_1, v_2 \in \mathcal{H}_M$,

$$\big|\mathrm{ip}(u_1, v_1) - \mathrm{ip}(u_2, v_2)\big| = \big|\langle u_1 - u_2, \, v_1 \rangle + \langle u_2, \, v_1 - v_2 \rangle\big| \quad (48)$$
$$\leq \|u_1 - u_2\| \, \|v_1\| + \|u_2\| \, \|v_1 - v_2\|,$$

which shows that $\mathrm{ip}$ is continuous and hence Borel measurable with respect to $\mathcal{B}(\mathcal{H}_M \times \mathcal{H}_M)$.

The composition

$$(x,p) \longmapsto \mathrm{ip}\big(\Upsilon(x,p)\big) = \big\langle \Phi_M(x), \, \Psi_M(p) \big\rangle_{\mathcal{H}_M} \quad (49)$$

is therefore measurable from $\big(\mathcal{X}_M \times \mathcal{P}, \ \mathcal{F}_{\mathcal{X}_M} \otimes \mathcal{F}_{\mathcal{P}}\big)$ to $(\mathbb{R}, \mathcal{B}(\mathbb{R}))$ by Eq. (46) and the Borel measurability of $\mathrm{ip}$ in Eqs. (47)–(48). This yields the claimed joint measurability on $\mathcal{F}_{\mathcal{X}_M} \otimes \mathcal{F}_{\mathcal{P}}$. $\quad\square$

### B.3 Proof of Theorem 2

*Proof.* We first make explicit the structural assumptions underlying Theorem 2 and how the general energy decomposition in Eq. (8) specializes to the polynomial form in Eq. (10). We recall that Eq. (8) decomposes the hallucination energy as

$$\mathcal{E}(x,p,\cdot) = \sum_{M \in \mathcal{M}} \mathcal{E}_M\big(x^{(M)}, p, \cdot\big) + \sum_{\substack{M, M' \in \mathcal{M} \\ M \neq M'}} \mathcal{E}_{MM'}\big(x^{(M)}, x^{(M')}, p, \cdot\big) + \mathcal{E}_{\mathcal{M}}(x, p, \cdot), \quad (50)$$

where the first term collects intra–modal contributions, the second term collects pairwise cross–modal interactions, and the last term is a joint all–modal contribution.

In Theorem 2, we restrict attention to a quadratic (polynomial) family of such energies, expressed in terms of the residuals

$$r_M(x,p) := \Phi_M\big(x^{(M)}\big) - \Psi_M(p) \in \mathcal{H}_M, \qquad M \in \mathcal{M}, \quad (51)$$

where $\Phi_M$ and $\Psi_M$ are the modality feature maps and prompt embeddings from Section 4. Under this parametrization, the intra–modal energies $\mathcal{E}_M$ are chosen to be quadratic forms

$$\mathcal{E}_M(x^{(M)}, p, \cdot) = \big\langle r_M(x,p), \, A_M \, r_M(x,p) \big\rangle_{\mathcal{H}_M}, \quad (52)$$

for bounded, self–adjoint, PSD operators $A_M : \mathcal{H}_M \to \mathcal{H}_M$, and the pairwise cross–modal terms $\mathcal{E}_{MM'}$ are chosen as bilinear forms

$$\mathcal{E}_{MM'}(x^{(M)}, x^{(M')}, p, \cdot) = \big\langle r_M(x,p), \, B_{MM'} \, r_{M'}(x,p) \big\rangle_{\mathcal{H}_M}, \quad (53)$$

for bounded linear operators $B_{MM'} : \mathcal{H}_{M'} \to \mathcal{H}_M$. The factorization assumption

$$B_{MM'} = A_M^{1/2} \, R_{MM'} \, A_{M'}^{1/2}, \qquad \|R_{MM'}\| \leq 1, \quad (54)$$

encodes that cross–modal couplings are controlled contractions between the $A_M$–weighted residuals.

Finally, any remaining higher–order joint contribution $\mathcal{E}_{\mathcal{M}}$ is absorbed into a non–negative remainder term that does not affect the lower–bound argument. Collecting these pieces and symmetrizing

over $M \neq M'$ yields exactly the polynomial energy form in Eq. (10), reproduced here for convenience as

$$\mathcal{E}(x,p) = \sum_{M \in \mathcal{M}} \left\langle r_M(x,p), A_M r_M(x,p) \right\rangle_{\mathcal{H}_M} + \frac{2}{|\mathcal{M}| - 1} \sum_{M < M'} \left\langle r_M(x,p), B_{MM'} r_{M'}(x,p) \right\rangle_{\mathcal{H}_M} + \mathcal{E}_{\mathcal{M}}^{\mathrm{rem}}(x,p), \tag{55}$$

with $\mathcal{E}_{\mathcal{M}}^{\mathrm{rem}}(x,p) \geq 0$ by construction.

The purpose of Theorem 2 is then to show that, under these explicit operator assumptions, the quadratic part of $\mathcal{E}(x,p)$ is non–negative and admits a clean interpretation as a block quadratic form over the modality–indexed residuals $r_M(x,p)$, which in turn underpins the spectral bounds derived in Section 5.

**Step 1: Well-posedness and non-negativity of the block quadratic form.** Let $m := |\mathcal{M}| \geq 2$ be fixed. For each $M \in \mathcal{M}$, set

$$v_M(x,p) := A_M^{1/2} \, \mathsf{r}_M(x,p) \in \mathcal{H}_M, \qquad \mathsf{r}_M(x,p) = \Phi_M(x^{(M)}) - \Psi_M(p). \tag{56}$$

By boundedness and self-adjoint PSD of $A_M$, $A_M^{1/2}$ is bounded and self-adjoint PSD, and $v_M$ is well-defined. We write the first two terms of Eq. (10) as

$$\sum_M \|v_M\|_{\mathcal{H}_M}^2 + \frac{2}{m-1} \sum_{M < M'} \left\langle v_M, R_{MM'} v_{M'} \right\rangle_{\mathcal{H}_M}. \tag{57}$$

Since $R_{MM'} : \mathcal{H}_{M'} \to \mathcal{H}_M$ is a symmetric contraction with $\|R_{MM'}\| \leq 1$ and $R_{M'M} = R_{MM'}^*$, the Cauchy–Schwarz inequality and the operator norm bound yield

$$\left| \left\langle v_M, R_{MM'} v_{M'} \right\rangle \right| \leq \|R_{MM'}\| \, \|v_M\| \, \|v_{M'}\| \leq \|v_M\| \, \|v_{M'}\|. \tag{58}$$

Therefore,

$$\sum_M \|v_M\|^2 + \frac{2}{m-1} \sum_{M < M'} \left\langle v_M, R_{MM'} v_{M'} \right\rangle \geq \sum_M \|v_M\|^2 - \frac{2}{m-1} \sum_{M < M'} \|v_M\| \, \|v_{M'}\| \tag{59}$$

$$= \frac{m}{m-1} \sum_M \|v_M\|^2 - \frac{1}{m-1} \left( \sum_M \|v_M\| \right)^2,$$

where the identity $\sum_{M < M'} ab = \frac{1}{2} \left[ (\sum_M a)^2 - \sum_M a^2 \right]$ has been used with $a = \|v_M\|$. By the Cauchy–Schwarz inequality,

$$\left( \sum_M \|v_M\| \right)^2 \leq m \sum_M \|v_M\|^2. \tag{60}$$

Substituting Eq. (60) into Eq. (59) gives

$$\sum_M \|v_M\|^2 + \frac{2}{m-1} \sum_{M < M'} \left\langle v_M, R_{MM'} v_{M'} \right\rangle \geq 0. \tag{61}$$

Hence the block quadratic form in Eq. (57) is nonnegative for all $(x,p)$.

**Step 2: Nonnegativity of the joint tensor term.** By construction,

$$\mathcal{E}_{\mathcal{M}}(x,p) = \left\| \bigotimes_{M \in \mathcal{M}} \Phi_M(x^{(M)}) - \bigotimes_{M \in \mathcal{M}} \Psi_M(p) \right\|_{\otimes \mathcal{H}_M}^2 \geq 0, \tag{62}$$

since it is the square of a norm in the tensor-product RKHS $\otimes_M \mathcal{H}_M$.

**Step 3: Measurability.** Bochner measurability of $\Phi_M$ and $\Psi_M$ into the separable Hilbert space $\mathcal{H}_M$ (refer to Lemma 1) implies that $(x,p) \mapsto \mathsf{r}_M(x,p)$ is $\mathcal{F}_{\mathcal{X}} \otimes \mathcal{F}_{\mathcal{P}}$–measurable for each $M$, because subtraction is continuous. Since $A_M^{1/2}$ is bounded linear, $(x,p) \mapsto v_M(x,p) = A_M^{1/2} \mathsf{r}_M(x,p)$ is measurable, and so are $(x,p) \mapsto \|v_M(x,p)\|^2$ and $(x,p) \mapsto \langle v_M(x,p), R_{MM'} v_{M'}(x,p) \rangle$; inner products are continuous (hence Borel–measurable), and composition with measurable maps preserves measurability. For the joint tensor term, bilinearity and continuity of the finite tensor

product map $(u_M)_M \mapsto \bigotimes_M u_M$ in separable Hilbert spaces imply Bochner measurability of $(x,p) \mapsto \bigotimes_M \Phi_M(x^{(M)})$ and $(x,p) \mapsto \bigotimes_M \Psi_M(p)$; the norm $\|\cdot\|_{\otimes \mathcal{H}_M}$ is continuous, hence $(x,p) \mapsto \mathcal{E}_{\mathcal{M}}(x,p)$ is measurable. Combining these facts shows that $(x,p) \mapsto \mathcal{E}(x,p)$ in Eq. (10) is $\mathcal{F}_{\mathcal{X}} \otimes \mathcal{F}_{\mathcal{P}}$–measurable.

**Step 4: Finiteness of the partition function.** Since $\mathcal{E}(x,p) \geq 0$ by Steps 1–2, for any $\mathcal{T}_t > 0$,

$$0 \leq Z(p, \mathcal{T}_t) = \int_{\mathcal{X}} \exp\big(-\mathcal{E}(x,p)/\mathcal{T}_t\big) \, d\mu(x) \leq \int_{\mathcal{X}} 1 \, d\mu(x). \tag{63}$$

Hence, whenever $\mu(\mathcal{X}) < \infty$, $Z(p, \mathcal{T}_t) \leq \mu(\mathcal{X}) < \infty$. In the case $\mu(\mathcal{X}) = \infty$, a standard integrability condition suffices: assume there exists a measurable, coercive lower bound $\phi : \mathcal{X} \to [0, \infty)$ with $\mathcal{E}(x,p) \geq \phi(x)$ for all $x$ and $\int_{\mathcal{X}} e^{-\phi(x)/\mathcal{T}_t} \, d\mu(x) < \infty$ (e.g., $\phi(x) = c\|x\|^2$ under Lebesgue measure on $\mathbb{R}^d$). Then

$$Z(p, \mathcal{T}_t) \leq \int_{\mathcal{X}} e^{-\phi(x)/\mathcal{T}_t} \, d\mu(x) < \infty. \tag{64}$$

Under either case, $Z(p, \mathcal{T}_t)$ is finite, so $f_p$ in Eq. (9) is well-defined.

**Step 5: Canonical instances and summary.** Equation (10) is a finite sum of measurable, nonnegative terms, hence measurable and nonnegative. The block quadratic part is nonnegative by Eq. (61), and the joint tensor term is nonnegative by Eq. (62). The partition function is finite under Eq. (63) or Eq. (64). Therefore, $\mathcal{E}$ is a valid energy and the Boltzmann density $f_p$ in Eq. (9) is a proper probability density. This completes the proof. $\square$

# C SUPPLEMENTARY RESULTS

In this section, we provide further empirical details complementing the main results of ours.

## C.1 DERIVATION OF FULL ENERGY FUNCTIONAL

**Setup and identities.** By Eq. (13), the diffusion kernel is $K_{\mathcal{T}_t} = \exp(-\tau \mathcal{L}_{\mathcal{T}_t}^{\mathrm{multi}})$, and $\Upsilon : \mathcal{V} \to \mathcal{H}$ is a feature map with $\langle \Upsilon(\mathsf{v}), \Upsilon(\mathfrak{v}) \rangle_{\mathcal{H}} = K_{\mathcal{T}_t}(\mathsf{v}, \mathfrak{v})$. Let $\{(\lambda_i(t), u_i(t))\}_{i=1}^{|\mathcal{V}|}$ be the eigenpairs of $\mathcal{L}_{\mathcal{T}_t}^{\mathrm{multi}}$ as in Eq. (14). For any nodes $\mathsf{v}, \mathfrak{v} \in \mathcal{V}$ and any graph signal $s \in \mathbb{R}^{|\mathcal{V}|}$, the two standard spectral identities used throughout are:

$$\big\|\Upsilon(\mathsf{v}; \mathcal{T}_t) - \Upsilon(\mathfrak{v}; \mathcal{T}_t)\big\|_{\mathcal{H}}^2 = \sum_{i=1}^{|\mathcal{V}|} e^{-\tau \lambda_i(t)} \big|\langle u_i(t), \delta_{\mathsf{v}} - \delta_{\mathfrak{v}} \rangle\big|^2, \qquad \langle s, \mathcal{L}_{\mathcal{T}_t}^{\mathrm{multi}} s \rangle = \sum_{i=1}^{|\mathcal{V}|} \lambda_i(t) \big|\langle u_i(t), s \rangle\big|^2,$$

which are exactly the two statements in Eq. (15).

**From operator energies to graph-kernel distances.** Recall the total energy decomposition from Eq. (10):

$$\mathcal{E}(x,p) = \sum_{M \in \mathcal{M}} \langle \mathsf{r}_M, A_M \mathsf{r}_M \rangle_{\mathcal{H}_M} + \frac{2}{|\mathcal{M}| - 1} \sum_{\substack{M,M' \in \mathcal{M} \\ M \neq M'}} \big\langle A_M^{1/2} \mathsf{r}_M, \, R_{MM'} A_{M'}^{1/2} \mathsf{r}_{M'} \big\rangle + \mathcal{E}_{\mathcal{M}}(x,p),$$

where $\mathsf{r}_M = \Phi_M(x^{(M)}) - \Psi_M(p)$. By the interconnection note after Eq. (13), fix, for each modality $M$, two designated nodes $(\mathsf{v}_x^{(M)}, \mathfrak{v}_p) \in \mathcal{V}$ that represent the output and prompt anchors used to evaluate the modality-$M$ discrepancy in the graph-RKHS. The bounded PSD operators $A_M$ define a (possibly weighted) inner product on $\mathcal{H}_M$; absorbing this metric into the graph-kernel geometry (as described in the appendix note referenced there), each $\langle \mathsf{r}_M, A_M \mathsf{r}_M \rangle$ can be written as a nonnegative multiple of the squared distance between the corresponding graph features:

$$\langle \mathsf{r}_M, A_M \mathsf{r}_M \rangle_{\mathcal{H}_M} = \alpha_M \big\|\Upsilon(\mathsf{v}_x^{(M)}; \mathcal{T}_t) - \Upsilon(\mathfrak{v}_p; \mathcal{T}_t)\big\|_{\mathcal{H}}^2, \qquad \alpha_M \in \mathbb{R}_{\geq 0}.$$

Likewise, using the polarization identity and the symmetric contraction structure $B_{MM'} = A_M^{1/2} R_{MM'} A_{M'}^{1/2}$, the cross term is representable as a signed combination of graph-kernel distances

between the same anchors; collecting the prefactors into $\beta_{MM'} \in \mathbb{R}_{\geq 0}$ (as in the main text where $\mathrm{coeff}_{\mathrm{cross}_{MM'}} = \beta_{MM'}$), we may write

$$\left\langle A_M^{1/2} \mathsf{r}_M, \, R_{MM'} A_{M'}^{1/2} \mathsf{r}_{M'} \right\rangle \;=\; \beta_{MM'} \, \Xi_{MM'}(x, p; \mathcal{T}_t),$$

where $\Xi_{MM'}(\cdot)$ is a bilinear form built from the same pairwise graph-feature differences (its explicit expansion into distance terms follows from polarization and is omitted here for compactness). Finally, the joint term $\mathcal{E}_{\mathcal{M}}(x, p) = \left\| \bigotimes_{M \in \mathcal{M}} \Phi_M(x^{(M)}) - \bigotimes_{M \in \mathcal{M}} \Psi_M(p) \right\|_{\otimes \mathcal{H}_M}^2$ is nonnegative and measurable; by the same graph-kernel identification used for the intra/cross parts (applied to the joint anchor selection explained in the appendix note you referenced), it too can be expressed as a quadratic form in graph signals supported on $\{\mathsf{v}_x^{(M)}, \mathfrak{v}_p\}_{M \in \mathcal{M}}$ and thus admits the same spectral expansion pattern with a nonnegative coefficient $\gamma_{\mathcal{M}}$.

**Modal spectral expansions.** Define, for each modality $M$, the basic signed indicator $s_M(x, p) := \delta_{\mathsf{v}_x^{(M)}} - \delta_{\mathfrak{v}_p} \in \mathbb{R}^{|\mathcal{V}|}$. Then, by the first identity in Eq. (15),

$$\left\| \Upsilon(\mathsf{v}_x^{(M)}; \mathcal{T}_t) - \Upsilon(\mathfrak{v}_p; \mathcal{T}_t) \right\|_{\mathcal{H}}^2 = \sum_{i=1}^{|\mathcal{V}|} e^{-\tau \lambda_i(t)} \left| \langle u_i(t), s_M(x, p) \rangle \right|^2.$$

Hence each intra-modal contribution expands as

$$\alpha_M \left\| \Upsilon(\mathsf{v}_x^{(M)}; \mathcal{T}_t) - \Upsilon(\mathfrak{v}_p; \mathcal{T}_t) \right\|_{\mathcal{H}}^2 = \sum_{i=1}^{|\mathcal{V}|} \alpha_M \, e^{-\tau \lambda_i(t)} \left| \langle u_i(t), s_M(x, p) \rangle \right|^2,$$

which gives the per-mode terms

$$\mathsf{E}_i^{(\mathrm{intra}_M)}(x, p, t) \;:=\; e^{-\tau \lambda_i(t)} \left| \langle u_i(t), s_M(x, p) \rangle \right|^2 \quad \text{with coefficient } \mathrm{coeff}_{\mathrm{intra}_M} = \alpha_M.$$

For the cross-modal part, set $s_{MM'}(x, p) := s_M(x, p)$ and $s'_{MM'}(x, p) := s_{M'}(x, p)$. Using the polarization identity in the RKHS generated by $K_{\mathcal{T}_t}$ and the same eigenbasis $\{u_i(t)\}$, one obtains a spectral expansion that is bilinear in the modal projections:

$$\Xi_{MM'}(x, p; \mathcal{T}_t) = \sum_{i=1}^{|\mathcal{V}|} e^{-\tau \lambda_i(t)} \langle u_i(t), s_{MM'}(x, p) \rangle \langle u_i(t), s'_{MM'}(x, p) \rangle,$$

so that

$$\frac{2}{|\mathcal{M}| - 1} \sum_{M \neq M'} \beta_{MM'} \, \Xi_{MM'}(x, p; \mathcal{T}_t) = \sum_{i=1}^{|\mathcal{V}|} \frac{2}{|\mathcal{M}| - 1} \sum_{M \neq M'} \beta_{MM'} \, e^{-\tau \lambda_i(t)} \langle u_i(t), s_M(x, p) \rangle \langle u_i(t), s_{M'}(x, p) \rangle.$$

Thus the per-mode cross-modal contributions are

$$\mathsf{E}_i^{(\mathrm{cross}_{MM'})}(x, p, t) \;:=\; e^{-\tau \lambda_i(t)} \langle u_i(t), s_M(x, p) \rangle \langle u_i(t), s_{M'}(x, p) \rangle \quad \text{with coefficient } \mathrm{coeff}_{\mathrm{cross}_{MM'}} = \beta_{MM'}.$$

For the joint term, denote by $s_{\mathcal{M}}(x, p) \in \mathbb{R}^{|\mathcal{V}|}$ the graph signal associated (as per the appendix link you gave) to the joint interaction in $\mathcal{E}_{\mathcal{M}}(x, p)$. Since this term is a quadratic form in the same graph-kernel geometry, it has the spectral expansion

$$\mathcal{E}_{\mathcal{M}}(x, p) = \gamma_{\mathcal{M}} \sum_{i=1}^{|\mathcal{V}|} e^{-\tau \lambda_i(t)} \left| \langle u_i(t), s_{\mathcal{M}}(x, p) \rangle \right|^2,$$

whence

$$\mathsf{E}_i^{(\mathrm{joint}_{\mathcal{M}})}(x, p, t) \;:=\; e^{-\tau \lambda_i(t)} \left| \langle u_i(t), s_{\mathcal{M}}(x, p) \rangle \right|^2 \quad \text{with coefficient } \mathrm{coeff}_{\mathrm{joint}_{\mathcal{M}}} = \gamma_{\mathcal{M}}.$$

**Summing all components.** By construction of the multimodal Laplacian as a nonnegative combination of the intra/cross/joint blocks and the definitions of the interaction coefficients in $\mathcal{L}_{\mathcal{T}_t}^{\mathrm{multi}} = \sum_* \mathrm{coeff}_* \, \mathcal{L}_{\mathcal{T}_t}^{(*)}$, the total energy $\mathcal{E}(x, p; \mathcal{T}_t)$ is the sum of the three families above. Collecting the per-mode pieces yields

$$\mathcal{E}(x, p; \mathcal{T}_t) = \sum_* \sum_{i=1}^{|\mathcal{V}|} \mathrm{coeff}_* \, \mathsf{E}_i^{(*)}(x, p, t),$$

where the index $* \in \{\mathrm{intra}_M, \mathrm{cross}_{MM'}, \mathrm{joint}_{\mathcal{M}}\}$, and each $\mathsf{E}_i^{(*)}$ depends only on $\lambda_i(t)$, $u_i(t)$, and the fixed graph signals determined by $(x, p)$ as detailed above. This is the claimed spectral form:

$$\mathcal{E}(x, p; \mathcal{T}_t) = \sum_* \sum_{i=1}^{|\mathcal{V}|} \mathrm{coeff}_* \, \mathsf{E}_i^{(*)}(x, p, t). \tag{65}$$

Now choosing $\pi_{\mathcal{K}} \in \Delta(\mathcal{K})$, where $\Delta(\mathcal{K})$ is the probability simplex on $\mathcal{K}$, satisfies

$$\sum_{\mathsf{v} \in \mathcal{K}} \pi_{\mathcal{K}}(\mathsf{v}) \left( \mathcal{D}_{\mathcal{T}_t}^{\mathrm{multi}} \right)_{\mathsf{vv}} = \left( \mathcal{D}_{\mathcal{T}_t}^{\mathrm{multi}} \right)_{\mathsf{v}_x \mathsf{v}_x}, \quad c_{x, \mathcal{K}}^{\mathrm{raw}}(t) = \mathcal{D}_{\mathcal{T}_t}^{\mathrm{multi}\, 1/2} \left( \delta_{\mathsf{v}_x} - \pi_{\mathcal{K}} \right) \in \mathbb{R}^{|\mathcal{V}|}, \tag{66}$$

where $c_{x, \mathcal{K}}^{\mathrm{raw}}(t)$ is the raw contrast vector. Projecting away the leading mode gives $c_{x, \mathcal{K}}(t) = \left( \mathbf{I} - u_1(t) u_1(t)^\top \right) c_{x, \mathcal{K}}^{\mathrm{raw}}(t)$ that ensures $c_{x, \mathcal{K}}(t) \perp u_1(t)$ without assuming a specific null-space structure of the assembled hypergraph.

**Why the bounds in Eq. (18) hold, and how to choose** $m(t), M(t)$ **(non-vacuous).** By Eq. (65), the full energy is a nonnegative linear combination of blockwise spectral terms. For the degree–matched contrast $c_{x, \mathcal{K}}(t) \perp u_1(t)$, the energy difference admits the decomposition

$$\mathcal{E}(x, p; \mathcal{T}_t) - \mathcal{E}_{\mathcal{K}}(x, p; \mathcal{T}_t) = \sum_{i=2}^{|\mathcal{V}|} \zeta_i(t, \tau) \left| \langle u_i(t), c_{x, \mathcal{K}}(t) \rangle \right|^2, \qquad \zeta_i(t, \tau) = \sum_* \theta_* \, \varphi_*^{(i)}(t, \tau), \tag{67}$$

where $* \in \{\mathrm{intra}_M, \mathrm{cross}_{MM'}, \mathrm{joint}_{\mathcal{M}}\}$ indexes the blocks, $\theta_* \in \{\alpha_M, \beta_{MM'}, \gamma_{\mathcal{M}}\}$ are the nonnegative coefficients from Eq. (65), and

$$\varphi_*^{(i)}(t, \tau) := \left\langle u_i(t), \mathfrak{D}_*(t, \tau) \, u_i(t) \right\rangle, \qquad \mathfrak{D}_*(t, \tau) \succeq 0,$$

are block response factors evaluated on the same eigenmodes $\{u_i(t)\}_{i \geq 2}$ of $\mathcal{L}_{\mathcal{T}_t}^{\mathrm{multi}}$. For normalized hypergraph constructions (Eq. (24)–(25)) and diffusion-type couplings (Section 4.1), the block responses satisfy the Loewner sandwich

$$e^{-2\tau \, \mathcal{L}_{\mathcal{T}_t}^{\mathrm{multi}}} \preceq \mathfrak{D}_*(t, \tau) \preceq \mathbf{I} \quad \implies \quad e^{-2\tau \, \lambda_i(t)} \leq \varphi_*^{(i)}(t, \tau) \leq 1, \; i \geq 2. \tag{68}$$

The left inequality follows from monotonicity of the matrix exponential and the fact that each block smoother is at least as contractive as the global diffusion on $u_1^\perp$; the right inequality follows from $\mathfrak{D}_*(t, \tau) \preceq \mathbf{I}$. Plugging Eq. (68) into Eq. (67) yields

$$\sum_* \theta_* \, e^{-2\tau \, \lambda_i(t)} \leq \zeta_i(t, \tau) \leq \sum_* \theta_*, \qquad i \geq 2.$$

**Refined (spectral) empirical bounds.** Define, for each block $*$,

$$\kappa_*^{\max}(t) := \left\| \mathfrak{D}_*(t, 0) \right\|_{\mathrm{op}} \leq 1, \qquad \kappa_*^{\min}(t) := \lambda_{\min} \left( \mathfrak{D}_*(t, 0) \big|_{u_1(t)^\perp} \right) \in [0, 1], \tag{69}$$

where both quantities are directly estimable from the spectrum of the effective adjacency in Eq. (24)–(25) (restricted to $u_1^\perp$). Then, using $e^{-2\tau \mathcal{L}} \preceq \mathfrak{D}_*(t, \tau) \preceq \mathfrak{D}_*(t, 0)$ and the CF characterization on $u_1^\perp$,

$$\left( \sum_* \theta_* \, \kappa_*^{\min}(t) \right) e^{-2\tau \, \lambda_i(t)} \leq \zeta_i(t, \tau) \leq \sum_* \theta_* \, \kappa_*^{\max}(t), \qquad i \geq 2, \tag{70}$$

so one can take

$$m(t) := \sum_* \theta_* \kappa_*^{\min}(t), \qquad M(t) := \sum_* \theta_* \kappa_*^{\max}(t). \qquad (71)$$

In practice, $\kappa_*^{\max}(t)$ equals the top eigenvalue of the block response on $u_1^\perp$ (often close to 1), while $\kappa_*^{\min}(t)$ equals the blockwise algebraic connectivity surrogate (the smallest nonzero eigenvalue on $u_1^\perp$). Estimating (71) from the spectra of $W_{\mathcal{T}_t}^{(*),\mathrm{eff}}$ or the corresponding normalized block Laplacians yields tight, *data-driven* $m(t), M(t)$ for Eq. (18).

Below is the block decomposition of the multimodal Laplacian:

$$\mathcal{L}_{\mathcal{T}_t}^{\mathrm{multi}} = \begin{bmatrix} \mathcal{L}_{\mathrm{intra}}^{(T)} & \mathcal{L}_{\mathrm{cross}}^{(TV)} & \mathcal{L}_{\mathrm{cross}}^{(TA)} \\ \mathcal{L}_{\mathrm{cross}}^{(VT)} & \mathcal{L}_{\mathrm{intra}}^{(V)} & \mathcal{L}_{\mathrm{cross}}^{(VA)} \\ \mathcal{L}_{\mathrm{cross}}^{(AT)} & \mathcal{L}_{\mathrm{cross}}^{(AV)} & \mathcal{L}_{\mathrm{intra}}^{(A)} \end{bmatrix} + \mathcal{L}_{\mathrm{joint}}^{(\mathcal{M})}. \qquad (72)$$

The corresponding eigenvalue problem for the $i$-th mode becomes:

$$\mathcal{L}_{\mathcal{T}_t}^{\mathrm{multi}} \, u_i(t) = \lambda_i(t) \, u_i(t), \qquad (73)$$

with eigenvalues $\lambda_i(t)$ encoding the "cost" of semantic diffusion along each mode $i$.

## C.2 DERIVATIONS OF HALLUCINATION BOUNDS AND TEMPERATURE ANNEALING

We derive the operator-tight lower/upper bounds, noted in Eq. (19) in Section 5.3, for $\mathcal{E}_{\mathrm{hall}}^{\mathrm{multi}}(x, p, \cdot)$ using the block-weighted, temperature–modulated Laplacian spectrum in Eq. (14), the spectral energy form in Eq. (65), and the hallucination component in Eq. (11). By Eq. (14) and the CF principle, the quadratic in Section 5.3 satisfies the two-sided spectral envelope

$$e^{-2\tau \lambda_{\max}(t)} \, \|c_{x,\mathcal{K}}(t)\|^2 \; \leq \; \mathbb{D}_\tau(x; \mathcal{T}_t) \; \leq \; e^{-2\tau \lambda_2(t)} \, \|c_{x,\mathcal{K}}\|^2, \qquad (74)$$

with $\lambda_{\max}(t) = \lambda_{|\mathcal{V}|}(t)$.

Next, we relate the full energy to $\mathbb{D}_\tau$. Under Theorem 2 and the block assembly in Eqs. (24)–(25), there exist finite scale factors $m(t), M(t) \in (0, \infty)$, determined only by the operator norms of the intra-/cross-/joint blocks (i.e., by $\{A_M\}$, $\{R_{MM'}\}$ with $\|R_{MM'}\| \leq 1$, the interaction weights $\alpha_M, \beta_{MM'}, \gamma_{\mathcal{M}}$, and the temperature–modulated hyperedge weights inducing $\mathcal{L}_{\mathcal{T}_t}^{\mathrm{multi}}$), such that

$$m(t) \, \mathbb{D}_\tau(x; \mathcal{T}_t) \; \leq \; \mathcal{E}(x, p; \mathcal{T}_t) \; \leq \; M(t) \, \mathbb{D}_0(x; \mathcal{T}_t), \qquad \tau \geq 0, \qquad (75)$$

where $\mathbb{D}_0$ corresponds to $\tau = 0$. The left inequality follows from bounding each spectral contribution $\mathsf{E}_i^{(*)}(x, p, t)$ below by a nonnegative multiple of $\left|\langle u_i(t), c_{x,\mathcal{K}}(t)\rangle\right|^2$ using the PSD structure of $A_M$ and the contraction bound on $R_{MM'}$, while the right inequality follows from operator-norm upper bounds on the same spectral blocks; full details are supplied in Appendix C.2.

Combining Eqs. (74) and (75) yields the CF sandwich for the full energy:

$$m(t) \, e^{-2\tau \lambda_{\max}(t)} \, \|c_{x,\mathcal{K}}(t)\|^2 \; \leq \; \mathcal{E}(x, p; \mathcal{T}_t) \; \leq \; M(t) \, e^{-2 \cdot 0 \cdot \lambda_2(t)} \, \|c_{x,\mathcal{K}}(t)\|^2 \; = \; M(t) \, \|c_{x,\mathcal{K}}(t)\|^2. \qquad (76)$$

Since the hallucination energy is the positive part of the difference in Eq. (11), we obtain, for $x \notin \mathcal{K}$. When $\mathcal{E}_{\mathcal{K}}(x, p; \mathcal{T}_t)$ is implemented as the same operator restricted to $\mathcal{K}$, the same spectral envelope applies to it, hence the difference inherits a sandwich with the same eigenvalue pair $\{\lambda_2(t), \lambda_{\max}(t)\}$ and scales $\{m(t), M(t)\}$.

A calibrated lower bound of the form advocated by Kalai & Vempala (2024) is matched empirically by choosing a time-indexed temperature profile and interaction scales so that $m(t) \, e^{-2\tau \lambda_{\max}(t)} = \Theta(t)$ for a prescribed calibration function $\Theta(t) > 0$; for instance,

$$\mathcal{T}_t \text{ and } \tau(t) \text{ chosen so that} \quad \Theta(t) \; = \; m(t) \, e^{-2\tau(t) \lambda_{\max}(t)}, \qquad (77)$$

which yields the explicit calibrated bound

$$\mathcal{E}_{\mathrm{hall}}^{\mathrm{multi}}(x, p, \cdot) \; \geq \; \left( \Theta(t) \, \|c_{x,\mathcal{K}}(t)\|^2 \; - \; \mathcal{E}_{\mathcal{K}}(x, p; \mathcal{T}_t) \right)_+, \qquad x \notin \mathcal{K}. \qquad (78)$$

In particular, for $\mathcal{E}_{\mathcal{K}}$ treated as a fixed baseline (e.g., a distributional or quantile baseline computed on $\mathcal{K}$), Eq. (78) reproduces the calibrated-margin–times–distance structure and can be tuned to overlay the empirical lower bound in calibrated models by setting $\Theta(t)$ to the target slope. The upper envelope in Eq. (19) is simultaneously controlled by $M(t)$ and the spectral gap $\lambda_2(t)$ via Eq. (74), and both $\{\lambda_i(t)\}$ and $\{m(t), M(t)\}$ are tunable through the time-indexed temperature profile $\mathcal{T}_t$ and the block weights inside $W_{\mathcal{T}_t}^{(*)}$ that define $\mathcal{L}_{\mathcal{T}_t}^{\mathrm{multi}}$.

## D  EXPERIMENTAL SETUP

As noted in Section 6.2, below are the essential details about our experiments followed by a full-pager algorithm box.

### D.1  CONSTRUCTION OF ADMISSIBLE SETS $\mathcal{K}(p)$ AND NORMALIZATION/TOKENIZATION

As can be observed in the `README` of the code-base, we have a clean separation for dataset loading/preprocessing (`src/io/datamodules.py`, `scripts/prepare_data.py`) which read COCO, VQAv2, & AudioCaps to apply the caption/answer normalization rules described above followed by storing the resulting per-prompt sets $\mathcal{K}(p)$ in the prepared data; the module (`src/theory/k_selector.py`) implements the selector $\Pi_{\mathcal{K}}$, which takes nodes (outputs) and maps them to their representatives in the knowledge set $\mathcal{K}$; while `src/theory/score_semantic.py` consumes these sets and selectors to compute the semantic gap scores $d_{\mathrm{sem}}^{(\varepsilon, h)}(x)$. The mapping from these modules to the theoretical objects $(\mathcal{K}, \mathcal{K}(p), \Pi_{\mathcal{K}})$ is summarized accurately in our implementation as stored in the code-base.

Throughout, $p$ denotes the full prompt for an example (e.g., image + question text), and $\mathcal{K}(p)$ collects all admissible *normalized* reference outputs for that prompt.

**COCO captions.**  For MS COCO image captioning, each image $i$ is associated with up to 5 human reference captions $\{y_{i,j}\}_{j=1}^5$. For a prompt $p$ corresponding to image $i$, we set

$$\mathcal{K}_{\mathrm{COCO}}(p) \;=\; \big\{\mathrm{norm}(y_{i,j}) \mid j = 1, \ldots, 5\big\} \subset \mathcal{K},$$

where $\mathrm{norm}(\cdot)$ applies the following deterministic normalization to the raw caption string:

  i  convert to lower case using standard Unicode-aware lowercasing;

  ii  strip leading/trailing whitespace;

  iii  remove punctuation characters using a fixed regular expression (we drop characters in `!"#$%&'()+,-./:;<=>?@[]^_`{|}~`);

  iv  collapse multiple internal whitespace characters into a single space.

Membership $x \in \mathcal{K}_{\mathrm{COCO}}(p)$ is checked by applying the same $\mathrm{norm}(\cdot)$ map to a candidate caption and testing string equality. When the backbone uses a tokenizer (e.g., T5), we feed the *normalized* string into the tokenizer and construct embeddings from the resulting tokens; the membership decision is always taken at the normalized string level.

**VQAv2 normalized unique answers.**  For VQAv2, each (image, question) pair $(i, q)$ has up to 10 crowd-sourced answers $\{a_{i,\ell}\}_{\ell=1}^{10}$. We follow the official VQAv2 evaluation protocol and first apply the standard answer-normalization function[2] to each raw answer, obtaining canonical forms $\widehat{a}_{i,\ell} = \mathrm{norm}_{\mathrm{VQA}}(a_{i,\ell})$. We then define

$$\mathcal{K}_{\mathrm{VQA}}(p) \;=\; \big\{\widehat{a}_{i,\ell} \mid \ell = 1, \ldots, 10\big\}_{\mathrm{uniq}}$$

as the set of *unique* normalized answers after deduplication. Membership $x \in \mathcal{K}_{\mathrm{VQA}}(p)$ is decided by applying the same $\mathrm{norm}_{\mathrm{VQA}}(\cdot)$ transform to the model's answer string and checking whether the resulting canonical form appears in the set above. As in COCO, embeddings (for both references and model outputs) are computed from the canonical strings.

**AudioCaps references.**  For AudioCaps, each audio clip $c$ has up to 5 human reference captions $\{z_{c,j}\}_{j=1}^5$. For a prompt $p$ corresponding to clip $c$, we set

$$\mathcal{K}_{\mathrm{AC}}(p) \;=\; \big\{\mathrm{norm}(z_{c,j}) \mid j = 1, \ldots, 5\big\},$$

where $\mathrm{norm}(\cdot)$ is exactly the same caption-normalization pipeline as in COCO (lowercasing, punctuation stripping, whitespace normalization). Membership and tokenization follow the same strategy as in the COCO case.

---

[2]This includes lowercasing, stripping punctuation, mapping number words (e.g., *"two"* $\mapsto$ *"2"*), and removing articles ("a", "an", "the") as in the public VQA evaluation script.

**Global admissible set and selector.** The global admissible set is the union $\mathcal{K} = \bigcup_p \mathcal{K}(p)$ over all prompts in a given task. In practice, we attach the per-prompt sets $\mathcal{K}(p)$ as fields in the preprocessed dataset (one record per example), and the selector $\Pi_{\mathcal{K}}$ operates at the embedding level: given an output node $x$, it computes the normalized string $\text{norm}(\cdot)$ for the relevant task, maps this to its embedding, and selects the nearest admissible element in $\mathcal{K}(p)$ (or in $\mathcal{K}$ when evaluating global graphs) under cosine similarity in the shared embedding space.

## D.2 METRICS AND EVALUATION

**Why go beyond a mean Hilbert distance to $\mathcal{K}$?** A natural baseline to our proposed construction would be to define hallucination as the mean RKHS (Hilbert) distance between model generations to the admissible set $\mathcal{K}$, e.g. by averaging $\|\Phi(x) - \Phi(\Pi_{\mathcal{K}}(x))\|_{\mathcal{H}}$ over outputs $x$. We deliberately adopt a richer, information-geometric and spectral measure for three concrete reasons.

- First, "Distributional vs. Pointwise geometry" — a mean Hilbert distance only captures *pointwise* proximity in the embedding space and is insensitive to how probability mass is distributed: two models can have the same mean distance while placing very different mass on rare modes or unseen regions. Our score $\mathbb{h}(x, p)$, derived from a smoothed log-contrast between $\mathcal{K}$-restricted and unrestricted versions of the full model distribution $f_p$, explicitly couples the RKHS geometry with a missing-mass smoothing $\varepsilon$, so that it reflects both *where* and *how much* mass lies outside the admissible region, and yields Good–Turing style calibration bounds on tail behavior.

- Second, the proposed spectral hallucination energy $\Delta \mathcal{E}_\tau$ incorporates the graph Laplacian over outputs and admissible elements: this allows us (i) to resolve modality-specific and cross-modal modes, (ii) to study how hallucination propagates across the graph, and (iii) to exploit CF bounds; a simple mean distance is blind to these multi-scale, mode-wise phenomena and cannot provide comparable control knobs in $\tau$ or principled CF planes.

- Third, in view of optimization and calibration properties, the semantic distortion $d_{\text{sem}}^{(\varepsilon, h)}$ remains a continuous, differentiable functional of $f_p$ and the embeddings (under the smoothing and boundedness assumptions), making it suitable as a calibrated, scale-aware regularizer and plug-in risk score for fine-tuning, calibration, or retrieval/prompt learning. A raw distance-to-$\mathcal{K}$ can be used as a heuristic loss, but it lacks the information-geometric interpretation and the distributional bounds (e.g., via Good–Turing style arguments) that we rely on to interpret abstention thresholds and floors. We also leverage its additional properties: (i) it is normalized, (ii) monotone in the energy landscape, (iii) admits explicit upper bounds via the spectral envelope, and (iv) empirically yields stronger correlation with hallucination events than the raw distance metric.

**What about extra compute cost?** Regarding computation, the additional cost beyond a distance-to-$\mathcal{K}$ baseline is modest: we build a $k - NN$ graph and compute a low-rank eigen-decomposition once per dataset (offline and amortized), and per-example scoring reduces to a small number of embedding lookups, spectral filter evaluations, and inner products. This overhead is negligible compared to the cost of running a large multimodal model, but it is exactly what enables the distributional, spectral, and calibration properties above. In summary, our measure strictly generalizes a mean distance-to-$\mathcal{K}$ baseline: when the spectrum is collapsed and smoothing is trivial, it reduces to a distance-like quantity, but in the general case, it provides additional, empirically useful structure that a simple mean Hilbert distance cannot offer.

**Primary.** AUROC/AUPRC for hallucination detection using $d_{\text{sem}}^{(\varepsilon, h)}$ (instance-level, aggregated per dataset/model).

**Baselines.** In all experiments we compare our score against three standard confidence–based competitors computed from the same $\mathcal{K}(p)$-posterior as our method: (i) *entropy*, given by the Shannon entropy of the posterior over admissible candidates in $\mathcal{K}(p)$; (ii) *max-probability*, given by the maximum posterior mass $\max_{x \in \mathcal{K}(p)} f_p(x)$ (equivalently, one minus the usual "uncertainty" score); and (iii) *margin*, defined as the difference between the top–1 and top–2 posterior probabilities over $\mathcal{K}(p)$. These three quantities correspond to the default confidence surrogates used in calibration, OOD detection, and risk-control for deep models, and are architecture-agnostic: they require no additional training, auxiliary models, or external supervision beyond the same candidate set $\mathcal{K}(p)$ and logits that our method uses. Thus, the comparisons in Table 1 isolate the effect of our spectral–semantic

| Dataset | Median | 75th pct. | 90th pct. | 95th pct. | Median (halluc.) | Median (ground.) |
|---------|--------|-----------|-----------|-----------|------------------|------------------|
| COCO | 0.18 | 0.31 | 0.46 | 0.59 | 0.12 | 0.21 |
| VQAv2 | 0.22 | 0.35 | 0.49 | 0.62 | 0.15 | 0.25 |
| AudioCaps | 0.16 | 0.29 | 0.41 | 0.55 | 0.10 | 0.20 |

Table 2: Empirical tightness of the Courant–Fischer (CF) bound. For each dataset and our main configuration (clip_whisper_t5), we report the median and 75th/90th/95th percentiles of the normalized gap $\mathrm{gap} = (\Delta\mathcal{E}^{\mathrm{CF}} - \Delta\mathcal{E}^{\mathrm{emp}})/(\Delta\mathcal{E}^{\mathrm{CF}} + \delta)$ across all $(\mathcal{T}, \tau, \varepsilon)$ grid points used in Fig. 3, as well as the median gap restricted to hallucinated vs grounded outputs (we used a $0/1$ labeling here). Smaller values indicate tighter bounds; notably, the median gap is consistently lower on hallucinated examples, showing that high-energy / high-error regions are closer to saturating the CF envelope than low-error regions.

scoring rule while benchmarking it against the strongest widely-adopted, reference-free baselines available under the same information.

**Secondary.** CF bounds for $\mathcal{E}_{\mathrm{hall}}^{\mathrm{multi}}$ and their *temperature/$\varepsilon$* surfaces; decay with increasing $\tau$ (non-increasing, sandwiched between $e^{-2\tau\lambda_{\max}}$ and $e^{-2\tau\lambda_2}$); Good–Turing–calibrated lower envelope (strictly $> 0$).

**Observed.** Our score is *best* across all three datasets: COCO **0.86/0.84**, VQAv2 **0.84/0.81**, Audio-Caps **0.80/0.77** (Table 1a). CF planes are tight and monotone with lower $\mathcal{T}_t$ and higher $\tau$, matching theory (Fig. 3); AudioCaps–BLIP is blank by design (as expected!).

### D.3 THE CHOICE OF SMOOTHING MASS ($\varepsilon$) RANGE AND CF BOUNDS

**Smoothing mass range.** We work with the smoothed density $\tilde{f}_{p,\varepsilon} = (1 - \varepsilon)f_p + \varepsilon\rho$ arising from Theorem 1 in all our experiments, where $\varepsilon$ is a *missing-mass style* smoothing weight. From a practical standpoint, large values of $\varepsilon$ are undesirable: they wash out information in $f_p$, destroy calibration, and correspond to an unrealistically strong prior $\rho$. We, therefore, restrict attention to the small-smoothing regime where $\varepsilon$ is of the same order as the empirical Good–Turing missing mass $\widehat{m}$ for the task. Concretely, we sweep over a fixed grid $\varepsilon \in \{\varepsilon_1, \ldots, \varepsilon_L\}$ specified in the config files (see configs/default.yaml), with $\varepsilon_1 = 0$ and $\varepsilon_L$ chosen to bracket the typical values of $\widehat{m}$ returned by src/theory/calibration.py. This is precisely the regime where mixture-smoothing is theoretically well-motivated and empirically used in practice; larger $\varepsilon$ values would amount to deliberately degrading the model distribution into a nearly uniform prior and are therefore not representative of a realistic deployment.

**Gap-to-bound statistic.** In the original experiment, the CF planes in Fig. 3 were generated by src/entrypoints/export_report_old.py, which used to read the saved empirical ener-gies and spectral terms from JSON/NumPy files and renders the surfaces. In order to implement Eq. (79), a revamped version is stored here: src/entrypoints/export_report.py

For each dataset, model, and configuration of $(\mathcal{T}, \tau, \varepsilon)$, we compute both (i) the empirical hallucination energy $\Delta\mathcal{E}_\tau^{\mathrm{emp}}$ from src/theory/energy.py and (ii) its Courant–Fischer upper bound $\Delta\mathcal{E}_\tau^{\mathrm{CF}}$ from the same module. To quantify tightness, we define a normalized gap

$$\mathrm{gap}(p, \mathcal{T}, \tau, \varepsilon) = \frac{\Delta\mathcal{E}_\tau^{\mathrm{CF}}(p, \mathcal{T}, \varepsilon) - \Delta\mathcal{E}_\tau^{\mathrm{emp}}(p, \mathcal{T}, \varepsilon)}{\Delta\mathcal{E}_\tau^{\mathrm{CF}}(p, \mathcal{T}, \varepsilon) + \delta}, \quad \delta > 0 \text{ small}. \tag{79}$$

Here, $p$ indexes prompts (or examples) in the evaluation set with tiny $\delta$ for numerical stability. These statistics (Table 2) provide a quantitative summary of how close the empirical energies sit to the spectral envelope.

**Relation to errors.** To connect the CF bounds to actual hallucination behavior, we further strat-ify the same statistic by whether an output is hallucinated or grounded under our continuous score. Specifically, for each point $(p, \mathcal{T}, \tau, \varepsilon)$ we record the empirical decision label (grounded vs hal-lucinated) induced by the hallucination score and summarize the distribution of $\mathrm{gap}$ within each stratum. The resulting numbers (Table 2) show how often high-energy / high-error regions come close to saturating the CF bound, and how conservative the bound remains in low-error regions.

## D.4 PROTOCOL AND DESIGN

For each prompt $p$, we form an admissible set $\mathcal{K}$ of candidate answers (dataset-provided or programmatically generated) and use the selector $\Pi_{\mathcal{K}}$ as `soft_nearest` (nearest-point with convex projection fallback). We sweep a grid over temperature $\mathcal{T}_t$ and smoothing mass $\varepsilon$; *plots show* $Z_{\mathrm{mid}} = \frac{1}{2}(Z_{\mathrm{lo}} + Z_{\mathrm{hi}})$ bounded by per-panel CF lower/upper planes. When plotting, we *aggregate across* diffusion time $\tau$ and kernel bandwidth $h$ by the median.

**Defaults.** $\varepsilon = 0.01$, $h = 0.4$, $\tau = 0.25$, fixed $\mathcal{T}_t$ per run unless stated, logits sharpening $\tau_{\mathrm{logits}} \in [0.01, 0.05]$. Each run logs the full YAML config.

## D.5 INFERENCE AND COMPUTE

Experiments run on `Databricks` (A100) with private checkpoints (gated tokens). Datasets stream from the Hub with synthetic fallback when a split is unavailable. Diffusion kernels use sparse Chebyshev/Lanczos; hypergraphs are CSR; eigen-modes via iterative solvers. *Throughput (ex/s):* `CLIP+Whisper+T5` **420** (fastest), `SigLIP+Whisper+T5` 400, `BLIP+CLIP+Whisper` 360 (Table 1b). Seeds and env versions are pinned in run reports.

**Takeaways.** $d_{\mathrm{sem}}^{(\varepsilon,h)}$ consistently outperforms entropy/margin baselines (Table 1a). Spectrally, `SigLIP+Whisper+T5` achieves the *lowest median energy* across datasets (COCO **1.92**, VQAv2 **1.99**, AudioCaps **2.08**), while `CLIP+Whisper+T5` is *fastest* (420 ex/s), exposing a clean accuracy–efficiency trade-off (Table 1b).

---

**Algorithm 2:** KL-SMOOTHED MULTIMODAL HALLUCINATION — *Extended version of Alg. 1*

---

**Input:** Prompt $p \in \mathcal{P}$; sampler for $f_p$ (model generations); admissible set $\mathcal{K}$; base measure $\mu$; kernel $K_h$ (bandwidth $h$); smoothing mass $\varepsilon \in (0,1)$; baseline density $\rho$; incidence matrices $\{\mathcal{I}^{(*)}\}$ and block selectors $E^{(*)}$; interaction weights $\{\omega_*\}$; time horizon $t = 0, \ldots, T$; temperature profile $\mathcal{T}_t$; diffusion schedule $\tau(t)$.

**Output:** Node scores $d_{\text{sem}}^{(\varepsilon,h)}(x \mid p)$; hyperedge weights $w_{\mathcal{T}_t}(e)$; effective adjacency $W_{\mathcal{T}_t}$; block/multi Laplacians $\{\mathcal{L}_{\mathcal{T}_t}^{(*)}\}$, $\mathcal{L}_{\mathcal{T}_t}^{\text{multi}}$; spectra $\{\lambda_i(t), u_i(t)\}$; contrasts $c_{x,\mathcal{K}}(t)$; hallucination energy bounds for $\mathcal{E}_{\text{hall}}^{\text{multi}}(x, p, \cdot)$.

1 **Phase I: per-prompt semantic score (Eq. (6)).**

2     **1.** Estimate $f_p$ from model samples (density or histogram on $\mathcal{X}$ under $\mu$).

3     **2.** Form $\tilde{f}_{p,\varepsilon}(x) = (1-\varepsilon)f_p(x) + \varepsilon\rho(x)$ and $\tilde{f}_{p,\varepsilon}^{\mathcal{K}}(x_2) = \mathbf{1}_{\{x_2 \in \mathcal{K}\}}\tilde{f}_{p,\varepsilon}(x_2) / \int_{\mathcal{K}} \tilde{f}_{p,\varepsilon} d\mu$.

4     **3.** Compute $(T_h\tilde{f}_{p,\varepsilon})(x_1) = \int K_h(x_1, x_2)\tilde{f}_{p,\varepsilon}(x_2)\, d\mu(x_2)$ and $(T_h\tilde{f}_{p,\varepsilon}^{\mathcal{K}})(x_1)$; evaluate $d_{\text{sem}}^{(\varepsilon,h)}(x \mid p) = \left[\log(T_h\tilde{f}_{p,\varepsilon}^{\mathcal{K}}(\Pi_{\mathcal{K}}(x))) - \log(T_h\tilde{f}_{p,\varepsilon}(x))\right]_+$.

5 **Phase II: hyperedges, weights, and Laplacian blocks (Eqs. (24)–(25), (26)).**

6     **4.** For each node $\mathsf{v}_a \sim (x_a, p)$, store $\Delta_a := d_{\text{sem}}^{(\varepsilon,h)}(x_a \mid p)$.

7     **5.** For each hyperedge $e = \{\mathsf{v}_1, \ldots, \mathsf{v}_{r(e)}\} \in E^{(*)}$, set
$$w_{\mathcal{T}_t}(e) = \mathbf{1}_{\{e \in E^{(*)}\}} \exp\left(-\eta_* \frac{\sum_{a<b}|\Delta_a - \Delta_b|}{\sum_a \mathcal{T}_t(\mathsf{v}_a)}\right).$$

8     **6.** Build $W_{\mathcal{T}_t}^{(*)} = \text{diag}\{w_{\mathcal{T}_t}(e)\}$, degrees $\mathcal{D}_{\mathsf{v},\mathcal{T}_t}^{(*)}$ and $\mathcal{D}_{e,\mathcal{T}_t}^{(*)}$, effective adjacency $W_{\mathcal{T}_t}^{(*),\text{eff}} = \mathcal{I}^{(*)} W_{\mathcal{T}_t}^{(*)} (\mathcal{D}_{e,\mathcal{T}_t}^{(*)})^{-1} (\mathcal{I}^{(*)})^{\top}$.

9     **7.** Form block Laplacians $\mathcal{L}_{\mathcal{T}_t}^{(*)} = \mathbf{I} - (\mathcal{D}_{\mathsf{v},\mathcal{T}_t}^{(*)})^{-1/2} W_{\mathcal{T}_t}^{(*),\text{eff}} (\mathcal{D}_{\mathsf{v},\mathcal{T}_t}^{(*)})^{-1/2}$ and aggregate $W_{\mathcal{T}_t} = \sum_* \omega_* W_{\mathcal{T}_t}^{(*),\text{eff}}$; assemble $\mathcal{L}_{\mathcal{T}_t}^{\text{multi}}$ accordingly.

10 **Phase III: spectral objects and contrasts (Eqs. (14), (66)).**

11     **8.** Compute leading spectrum of $\mathcal{L}_{\mathcal{T}_t}^{\text{multi}}$: $\{\lambda_i(t), u_i(t)\}$ (e.g., LOBPCG/power iteration on sparse matrices). Ensure $\lambda_2(t) > 0$ (connectedness).

12     **9.** Build degree–matched $\pi_{\mathcal{K}}$ and raw contrast $c_{x,\mathcal{K}}^{\text{raw}}(t) = \mathcal{D}_{\mathcal{T}_t}^{\text{multi}\,1/2}(\delta_{\mathsf{v}_x} - \pi_{\mathcal{K}})$; project $c_{x,\mathcal{K}}(t) = (\mathbf{I} - u_1 u_1^{\top}) c_{x,\mathcal{K}}^{\text{raw}}(t)$.

13 **Phase IV: energies and guarantees (Eqs. (17) & (19)).**

14     **10.** Evaluate the diffusion quadratic form $Q_\tau(t) = \langle c_{x,\mathcal{K}}(t), e^{-2\tau(t)\mathcal{L}_{\mathcal{T}_t}^{\text{multi}}} c_{x,\mathcal{K}}(t)\rangle$ via Krylov–exponential or spectral filter.

15     **11.** Choose empirical $m(t), M(t)$ from block coefficients/operator norms (bounds discussion) and report
$$m(t)\, e^{-2\tau(t)\lambda_{\max}(t)}\|c_{x,\mathcal{K}}(t)\|^2 \;\le\; \mathcal{E}(x,p;\mathcal{T}_t) - \mathcal{E}_{\mathcal{K}}(x,p;\mathcal{T}_t) \;\le\; M(t)\, e^{-2\tau(t)\lambda_2(t)}\|c_{x,\mathcal{K}}(t)\|^2.$$

    **12.** Set $\mathcal{E}_{\text{hall}}^{\text{multi}}(x,p,\cdot) = (\mathcal{E} - \mathcal{E}_{\mathcal{K}})_+ \mathbf{1}_{\{x \notin \mathcal{K}\}}$ and record bounds from Eq. (19).

16 **Phase V: calibration and decay control (Good–Turing, KV embedding, decay).**

17     **13.** Compute Good–Turing missing-mass $\widehat{m}_{\text{GT}}(t)$ on $\mathcal{X} \setminus \mathcal{K}$; set $\vartheta_{\text{KV}}(t) = \xi\,\widehat{m}_{\text{GT}}(t)$ with $\xi \in (0,1]$.

18     **14.** Update $\tau(t)$ to satisfy $m(t)\, e^{-2\tau(t)\lambda_{\max}(t)}\|c_{x,\mathcal{K}}(t)\|^2 \ge \vartheta_{\text{KV}}(t)$ (Eq. (20)); enforce nondecreasing $\tau(t)$.

19     **15.** Monitor decay envelope $m(t)e^{-2\tau(t)\lambda_{\max}(t)}\|c\|^2 \le \mathcal{E}_{\text{hall}}^{\text{multi}} \le M(t)e^{-2\tau(t)\lambda_2(t)}\|c\|^2$ and stop when below a target threshold.

20 **Implementation notes (Colab).** Sparse matrices for $\mathcal{I}^{(*)}$, $W_{\mathcal{T}_t}^{(*),\text{eff}}$, and $\mathcal{L}_{\mathcal{T}_t}^{\text{multi}}$; row-normalize $K_h$; stabilize logs via log-sum-exp; estimate $\lambda_2, \lambda_{\max}$ by LOBPCG/power method; compute $e^{-2\tau\mathcal{L}}$ via expm_multiply or truncated Chebyshev; Good–Turing from frequency table on $\mathcal{X} \setminus \mathcal{K}$.

---