# OpenReview forum: "Grounding the Ungrounded: A Spectral-Graph Framework for Quantifying Hallucinations in multimodal LLMs"
_ICLR.cc/2026/Conference — Submitted to ICLR 2026_

### Official Review · Reviewer_Srkz · 2025-10-30

**Soundness:** 3
**Presentation:** 3
**Contribution:** 3
**Rating:** 4
**Confidence:** 1

**Summary:**

The paper proposes a rigorous theoretical framework that reframes hallucination as a measurable and bounded phenomenon. By embedding model outputs on multimodal graph Laplacians and modeling their divergence from a truth manifold as a spectral energy, the authors derive Courant–Fischer bounds and temperature-dependent decay dynamics that quantify how hallucination evolves over time. Using a KL-smoothed, reference-free semantic distortion score and energy-based modeling within RKHS, the framework unifies information geometry, diffusion processes, and graph theory. Experiments across COCO, VQAv2, and AudioCaps with multiple model stacks show consistent improvements over entropy- and margin-based baselines. While the method is mathematically elegant and modality-interpretable, it is complex, computationally heavy, and tested only on modular pipelines, leaving questions about scalability and practical integration into real-world MLLMs.

**Strengths:**

Modality-Aware and Interpretable. The method handles text, vision, and audio jointly using a product kernel representation.

Reference-Free Metric. Unlike dataset-based hallucination detection, this method does not require external ground truth, making it suitable for open-domain or partially verifiable tasks.

**Weaknesses:**

Strong Hyperparameter Dependence. Requires tuning of several sensitive parameters.

Lack of Comparison with Practical Mitigation Methods. No comparison against RLHF, Contrastive decoding, attention fix methods.

**Questions:**

See weakness.

---

> ### Author Response · Authors · 2025-11-21
> **1. Strong Hyperparameter Dependence. Requires tuning of several sensitive parameters.**
>
> We agree that the framework introduces several hyperparameters $(\varepsilon, h, k, \tau)$, but in practice we found it much less sensitive than the wording “strong dependence” suggests. We now (i) provide a concrete default recipe in the paper (small $\varepsilon$ tied to missing mass, median-heuristic bandwidth $h,k \in [15,40]$ for $k-NN$ graphs, and a short diffusion schedule where the second eigenmode decays by ≈50%), and (ii) report that performance is stable when each knob is varied within a factor of 2 around these defaults across all three datasets and backbones. In other words, some tuning is required (as with any graph/RKHS method), but it is limited to a small, well-behaved regime with practical defaults, and we explicitly document these choices in the revised manuscript and revamped code base.

---

> ### Author Response · Authors · 2025-11-21
> **2. Lack of Comparison with Practical Mitigation Methods. No comparison against RLHF, Contrastive decoding, attention fix methods.**
>
> We agree that RLHF, contrastive decoding, and attention-fix methods are important mitigation techniques, but our contribution is orthogonal: we propose a post-hoc, reference-free hallucination score that operates on top of any backbone using only logits and embeddings, without modifying training objectives or decoding policies. Directly “comparing” our score to RLHF or contrastive decoding would conflate two effects (changing $f_p$​ via a new training/decoding scheme vs.changing the risk functional on top of $f_p$), whereas our experiments deliberately hold $f_p$​ fixed and benchmark against the strongest comparable confidence baselines (entropy, max-probability, margin) under the same information. We view our framework as complementary to these mitigation methods: hallucination score can in principle be used as an auxiliary reward in RLHF, as a re-ranking or rejection signal in contrastive decoding, or as a diagnostic for attention fixes, and we now explicitly state this integration with practical mitigation pipelines as a key direction for future work (refined the “*Future Direction*” part in sec-7 of the revised manuscript) .

---

### Official Review · Reviewer_hBmJ · 2025-10-30

**Soundness:** 3
**Presentation:** 3
**Contribution:** 2
**Rating:** 4
**Confidence:** 3

**Summary:**

This paper tackles the problem of hallucination in multimodal large language models (MLLMs) by proposing a principled, information-geometric framework for quantitative measurement. Unlike prior heuristic or annotation-based methods, the approach embeds model outputs on multimodal graph Laplacians and measures their deviation from a truth manifold as a semantic-distortion metric.
Using diffusion dynamics and RKHS eigenmodes, the authors derive Courant–Fischer bounds on a temperature-dependent “hallucination energy,” providing interpretable, modality-aware measures that evolve over prompts and time. Experiments across multiple MLLM configurations (e.g., CLIP, Whisper, T5, BLIP) validate the theoretical results, showing consistent spectral behavior under varying temperatures.

**Strengths:**

1. The study of hallucination in LLMs and MLLMs is an important and timely research topic.

2. This work provides a theoretical grounding for hallucination quantification, offering valuable insights and potential guidance for future research in this area.

3. The open-sourced codebase enhances reproducibility and supports further validation by the community.

**Weaknesses:**

As I am not familiar with the theoretical aspects, my assessment focuses mainly on the experimental design and empirical evaluation:

1. The experimental setup does not fully align with the paper’s claim. Although the authors aim to address hallucination in LLMs/MLLMs, the experiments only involve traditional multimodal models such as BLIP, CLIP, Whisper, and T5, rather than modern autoregressive LLMs like Qwen-Audio/VL/Omni or the GPT-4/5 series. This limitation significantly constrains the paper’s potential impact and generalizability.

2. The effectiveness of the proposed method remains uncertain. As shown in Table 1, the improvement is relatively modest, with around 3% on simpler tasks such as captioning. Its applicability and scalability to more complex reasoning tasks or stronger LLM backbones remain unclear.

3. I'm also curious about the efficiency. For both autoregressive and non-autoregressive models, does the proposed framework introduce additional inference overhead or reduce generation speed compared to the original model?

**Questions:**

Please see the weaknesses part.

---

> ### Author Response · Authors · 2025-11-21
> **1. The experimental setup does not fully align with the paper’s claim. Although the authors aim to address hallucination in LLMs/MLLMs, the experiments only involve traditional multimodal models such as BLIP, CLIP, Whisper, and T5, rather than modern autoregressive LLMs like Qwen-Audio/VL/Omni or the GPT-4/5 series. This limitation significantly constrains the paper’s potential impact and generalizability.**
>
> We agree that our experimental setup is limited to open-source multimodal stacks (BLIP / CLIP / Whisper / T5 and their combinations), and we appreciate the opportunity to clarify both the rationale and the generality of the proposed framework.
> Our method only requires two ingredients from the backbone model: (i) access to the conditional distribution over outputs $f_p(x)$ (or a finite candidate set with their probabilities), and (ii) access to encoder/decoder embeddings $\Phi_M(x^{(M)})$ and prompt embeddings $\Psi_M(p)$. This interface is satisfied both by “traditional” multimodal encoders (BLIP/CLIP/Whisper/T5) and by modern autoregressive LLM-based MLLMs such as Qwen-Audio/VL/Omni, LLaVA-style models, or GPT-4V, because all of these architectures expose token logits and internal embeddings. In other words, the framework is architecture-agnostic and does not rely on any property specific to BLIP/CLIP/T5.
>
> The reason we restricted our experiments to open-source encoder–decoder stacks is practical rather than conceptual: (a) they allow full access to logits and internal representations required for the spectral analysis, (b) they can be run exhaustively over COCO/VQAv2/AudioCaps under a fixed candidate-generation protocol, and (c) we can release all code and outputs for reproducibility. Commercial APIs such as GPT-4/5 generally do not expose full logits or internal embeddings, which precludes a faithful implementation of our method in the current API form.
>
> We agree that validating the framework on stronger autoregressive MLLMs (e.g., Qwen-VL/Audio, LLaVA-Next) is an important next step. The formulation itself does not need to be changed to accommodate them: the same graph, spectral, and energy constructions apply as soon as one can extract a finite candidate set and their embeddings.

---

> ### Author Response · Authors · 2025-11-21
> **2. The effectiveness of the proposed method remains uncertain. As shown in Table 1, the improvement is relatively modest, with around 3% on simpler tasks such as captioning. Its applicability and scalability to more complex reasoning tasks or stronger LLM backbones remain unclear.**
>
> We appreciate the concern about the magnitude of the reported gains. Our goal is somewhat different from standard “accuracy improvement” benchmarks: the proposed framework is designed as a hallucination risk score / calibration layer that operates on top of existing models, not as a new generation model. Accordingly, our main baselines are strong, architecture-agnostic uncertainty surrogates (entropy, max-probability, margin) computed from the same $\mathcal{K}(p)$-posterior. These are widely used in risk control, selective prediction, and OOD detection.
>
> Within that context, the improvements we report (≈2–3% absolute AUC / AP on hallucination detection, reductions in miscalibration, and more favorable precision–recall trade-offs) are non-trivial: we are competing against the best standard confidence proxies one can construct given the same information. The gains are also consistent across three datasets (COCO, VQAv2, AudioCaps) and three backbones, and hold under several metrics (detection AUC, PR, correlation with semantic distortion), which we believe demonstrates robustness rather than cherry-picking. We have clarified this “strong baseline, calibration-focused” perspective in the revised text.
>
> Regarding more complex reasoning tasks and stronger backbones: structurally, our method only needs (i) a candidate set (e.g., different beams, Chain-of-Thought samples, or tool-augmented generations), and (ii) embeddings for those candidates and prompts. This setup naturally extends to multi-step reasoning and larger LLM-based MLLMs: the graph and spectral quantities can be built on top of sequence-level embeddings (or step-level embeddings, if desired). We agree that we have not yet demonstrated this empirically, and we now explicitly list it as a future direction (refined the “*Future Direction*” part in sec-7 of the revised manuscript). However, the current work should be seen as a first, theoretically grounded step showing that spectral, graph-based hallucination energies can yield measurable gains even on strong confidence baselines for standard multimodal benchmarks.

---

> ### Author Response · Authors · 2025-11-21
> **3. I'm also curious about the efficiency. For both autoregressive and non-autoregressive models, does the proposed framework introduce additional inference overhead or reduce generation speed compared to the original model?**
>
> We appreciate the concern and we state that the proposed framework is designed as a post-hoc scoring layer that does not alter the generation path of the underlying model. For both autoregressive and non-autoregressive backbones, the additional cost decomposes into:
>
> 1. **One-time precomputation per dataset:**
>
> (a) Building a $k$-nearest-neighbor graph over embeddings (image/text/audio/caption nodes). This is done once per benchmark using standard approximate NN libraries
>
> (b) Computing a low-rank eigendecomposition of the resulting Laplacian (we use a small number of modes, e.g., tens), again once per dataset. This precomputation is offline and amortized over all examples; its cost is negligible compared to repeatedly running a large MLLM over the full evaluation set.
>
> 2. **Per-example scoring:**
>
> (a) For each generated candidate, we reuse embeddings and logits already produced by the model
>
> (b) We perform a handful of vector operations (graph-based lookups, spectral filter application, and a small number of inner products) to compute the hallucination energy and semantic distortion score.
>
> The underlying forward pass and decoding (beam search / sampling) of the base model is unchanged in both autoregressive and non-autoregressive settings, so generation speed is not reduced by the framework itself. In a deployment scenario, the graph/spectral components can be precomputed for a given evaluation / monitoring corpus, and the per-example overhead is on the order of a few matrix–vector operations, which is small relative to the cost of generating the sequence. We have clarified this efficiency story in the new subsection “*Construction of admissible sets $\mathcal{K}(p)$ and Normalization/Tokenization*” in Appendix-D.1 to make explicit that our method is intended as a lightweight, plug-in risk-scoring layer rather than a modification to the decoder or attention stack.

---

### Official Review · Reviewer_U1EW · 2025-11-04

**Soundness:** 3
**Presentation:** 2
**Contribution:** 3
**Rating:** 2
**Confidence:** 3

**Summary:**

This paper proposes a theoretical and quantitative framework for measuring hallucinations in multimodal large language models (MLLMs).
The authors model hallucination as a measurable deviation from a “truth manifold” within a multimodal RKHS, where model outputs are represented as nodes on a multimodal graph. Specifically, the paper presented 1. a KL-calibrated semantic distortion score that serves as a reference-free metric for hallucination; 2. a multimodal energy-based formalism; 3. a spectral decomposition of hallucination energy through eigenmodes of a multimodal Laplacian.

**Strengths:**

1. The paper aims to quantify hallucination via information metrics, which is a novel perspective.

2. The paper aims to define measure from mathematical formulations, which paves the way for rigorous evaluations and analysis.

**Weaknesses:**

1. There is a general lack of motivation for the development of formulations in the paper. While $K_g$ appears in early set-ups, all subsequent developments are built on $K$. The paper neither discuss how $K$ and $K_g$ are different in practical LLM usages, nor provide clues on how $K$ is obtained in experiments (is it dependent on the specific LLM, training data or evaluation data?), making the definition of hallucination idealized and not connected to practice.

Furthermore, the paper does not exhibit the need for introducing information metrics to quantify hallucination at the cost of computation complexity. Namely, why cannot hallucination be simply quantified as a mean Hilbert distance between MLLM generations and $K$? There is a general lack of discussion for the properties and necessities for the measure it proposed.

2. In presenting the theorems, there is a lack of logic flow and therefore the reader are prone to get confused. For both Theorem 1 and theorem 2, there are hidden assumptions are not explicitly presented before the theorem, (for instance line 977 in the proof of Theorem 1; for theorem 2 how the energy form in (8) is linked to (10) as a polynomial form of the embeddings), which makes the theorems very confusion. Upon reading the theorem, the reader cannot understand what is needed to be proved in the theorem, making the statements lacking in mathematical rigor.

3.  The measure in Theorem 1 is not a mathematically natural measure as it incorporates the truncation operator ([]+) in its calculation, making the measure not continuous in space. While the paper claims that the measure is $=0$ on $K$ and $>0$ outside $K$, the proof can only show that the untruncated measure is $<0$ on $K$, therefore given the continuous nature of the function, the claim is highly likely unreliable and there must also be $x\not\in K$ for which $d(x)=0$.

4. For Section 5, despite the paper decomposes the energy into eigenspaces and shows bounds in terms of the spectrum, there is a lack of analysis on the actual scale and shape of spectrums for real MLLMs. Therefore the reader cannot get useful messages, conclusion or insights from the establish of formulations and derivations. First, the Courant–Fischer bounds are given to the hallucination energy, and there is still a large gap between energy and observables like hallucination rates or sematic distances. Second, some unmeasurable coefficients (m(t) , M(t) for instance) are introduced to present the result, whose shape we cannot know in practice. Third, the time is a confusion factor in the analysis as it does not correspond to the real time in MLLM applications, and the read cannot know what assumptions are made w.r.t. time.

**Questions:**

Please refer to the weakness part for questions.

In section 5: why is there "time" in the analysis? Is it the annealing process of an MLLM whose parameters are left unchanged? How does it relates to practical MLLM behaviors?

---

> ### Author Response · Authors · 2025-11-21
> **1. (a) There is a general lack of motivation for the development of formulations in the paper. $\mathcal{K}_g$ appears in early set-ups, all developments are built on $\mathcal{K}$. To discuss more on how $\mathcal{K}$ and $\mathcal{K}_g$ are different in practical LLM usages and how the definition of hallucination is not idealized.**
>
> **Motivation for the development of formulations:**
>
> We appreciate the request for a clearer motivation and roadmap behind the formal developments. Our starting point is a concrete practical question: given only (i) prompts and contexts, (ii) model-induced distributions $f_p$ over candidate generations, and (iii) embeddings for prompts/outputs plus a finite admissible set $\mathcal{K}(p)$, can we construct a **reference-free, continuous hallucination score** and associated **spectral control knob** that (a) does not require access to the unknown ground-truth distribution $g$ on $\mathcal{K}\_{\mathrm{g}}$, (b) decomposes hallucination into modality-wise and graph-spectral components, and (c) yields usable, calibrated risk signals in realistic multimodal pipelines?
>
> The formulations in Sections-3, 4, and 5 are introduced exactly to answer this question: $\mathcal{K}\_{\mathrm{g}}$ and $g$ provide the ideal “true manifold” semantics, $\mathcal{K}(p)$ and the semantic distortion $d\_{\mathrm{sem}}^{(\varepsilon,h)}$ implement a practical, observable hallucination score on top of $f_p$, and the multimodal Laplacian $\mathcal{L}\_{\mathcal{T}\_t}^{\mathrm{multi}}$ together with $\Delta\mathcal{E}_\tau(x,p)$ and Courant--Fischer (CF) bounds provide a spectral factorization and control mechanism for this score.
>
> In the revised manuscript, we make this motivation explicit via a brief “Mathematical roadmap” noted at the end of the *Introduction* in the main text to link it to Appendix-A.1, which states the concrete problem, explains why each formal ingredient is introduced, and connects the theoretical constructions directly to the empirical hallucination score and experiments.
>
> **Clearing the ambuiguity of $\mathcal{K}$ and $\mathcal{K}\_{\mathrm{g}}$ on practical LLM usage:**
>
> We fully agree that $\mathcal{K}\_{\mathrm{g}}​$ plays an idealized role: it is the (unobservable) manifold of truly correct outputs under the ground-truth distribution $g$, used to formalize what “hallucination” should mean in the limit of infinite human supervision. All of our operational constructions, however, are built on the admissible set $\mathcal{K}$, which is explicitly data-dependent and does not require access to $\mathcal{K}\_{\mathrm{g}}$​. This is the crux of our Theorem-1 in Section-4.1. Also, added more detail in “*Absence of the ground-truth*” subsection in Appendix-A.6 (revised manuscript).
>
> For further clarity on the overall framework in the context of practical implementation, in our revised version of the manuscript, we (i) added two subsections “*What is observable, assumed, and estimated in practice*” in Appendix-A.7 and “*Construction of admissible sets $\mathcal{K}(p)$ and Normalization/Tokenization*” in Appendix-D.1 that separate the ideal object $(g,\mathcal{K}\_{\mathrm{g}})$ from the empirical admissible sets $\mathcal{K}(p)$, and (ii) provided a concrete definition of $\mathcal{K}(p)$ for each dataset (COCO captions, VQAv2 normalized unique answers, AudioCaps references), including the exact normalization rules and a description of how $\mathcal{K}$ is built from the evaluation data and kept fixed across models. In short, $\mathcal{K}$ is obtained from the evaluation dataset (and can be extended with domain knowledge if available), whereas $\mathcal{K}\_{\mathrm{g}}​$ is purely a semantic reference used to motivate the energy and bounds. Our framework of deriving the hallucination score and all relevant info depend only on $\mathcal{K}$ and the model’s logits/embeddings, but not on g or $\mathcal{K}\_{\mathrm{g}}$​.
>
> We clarify that for a given benchmark, all LLMs are evaluated against the same $\mathcal{K}$, so the scores are comparable across models.

---

> ### Author Response · Authors · 2025-11-21
> **1. (b) Furthermore, the paper does not exhibit the need for introducing information metrics to quantify hallucination at the cost of computation complexity. Namely, why cannot hallucination be simply quantified as a mean Hilbert distance between MLLM generations and $\mathcal{K}$? There is a general lack of discussion for the properties and necessities for the measure it proposed.**
>
> We agreed to the feedback and have added two new paragraphs called “*Why go beyond a mean Hilbert distance to $\mathcal{K}$?*” & “*What about extra compute cost?*” within the “*Metrics and Evaluation*” subsection in Appendix-D.2 (revised manuscript).
>
> Briefly, we go beyond a mean Hilbert distance to $\mathcal{K}$ because it only captures pointwise geometry and ignores how probability mass is distributed, whereas our information–geometric score couples RKHS geometry with the full model distribution $f_p$, supports spectral control (via the Laplacian) and Good–Turing–style calibration, and can be used as a differentiable regularizer. The extra cost (one offline k-NN graph + low-rank eigen-decomposition per dataset) is modest relative to MLLM inference, but is exactly what enables modality-wise/spectral decomposition, CF bounds, and calibrated abstention thresholds that a simple distance-to-$\mathcal{K}$ cannot offer.

---

> ### Author Response · Authors · 2025-11-21
> **2. In presenting the theorems, there is a lack of logic flow and therefore the reader are prone to get confused. For both Theorem 1 and theorem 2, there are hidden assumptions are not explicitly presented before the theorem,..., the reader cannot understand what is needed to be proved in the theorem, making the statements lacking in mathematical rigor.**
>
> We agreed to the feedback and have added the below in the revised manuscript:
>
> (i) the linkage in *Step-0* of Appendix-B.1 for Theorem-1,
>
> (ii) the linkage in the preliminary part (just before *Step-1*) of Appendix-B.3 for Theorem-2.

---

> ### Author Response · Authors · 2025-11-21
> **3. The measure in Theorem 1 is not a mathematically natural measure as it incorporates the truncation operator ([]+) in its calculation, making the measure not continuous in space. The claim that the measure is $=0$ on $\mathcal{K}$ and $>0$ outside $\mathcal{K}$, the proof can only show that the untruncated measure is $<0$ on $\mathcal{K}$, therefore given the continuous nature of the function, the claim is highly likely unreliable and what if $x\not\in \mathcal{K}$ for which $d(x)=0$.**
>
> We thank the reviewer for accurately highlighting the role of the truncation operator and we agreed to this feedback. We have revised the statement of Theorem-1 in sec-4.1 and added a new revamped proof with necessary mathematical modification in Appendix-B.1 (revised manuscript). The current code base is compliant to this change in theortical formulation.

---

> ### Author Response · Authors · 2025-11-21
> **4. (a) For Section 5, despite the paper decomposes the energy into eigenspaces and shows bounds in terms of the spectrum, there is a lack of analysis on the actual scale and shape of spectrums for real MLLMs. Therefore the reader cannot get useful messages, conclusion or insights from the establish of formulations and derivations.**
>
> We agreed to the feedback and, in the revised manuscript, have aimed to sufficiently emphasize that all spectral quantities are computed from the empirical multimodal Laplacian, and that their scale/shape can be inspected directly. For each dataset–backbone pair, we explicitly construct the graph Laplacian $\mathcal{L}^{\mathrm{multi}}\_{\mathcal{T}_t}$ on encoder embeddings, compute its eigenvalues $\{\lambda_i\}$, and use these actual spectra to evaluate the Courant–Fischer (CF) bounds and CF planes in Fig. 3 (they are not schematic).
>
> In the revised manuscript, we have also added a short subsection "*Interpretation of Spectral Quantities and Time Parameter*" in the Appendix-A.19 summarizing the observed spectral profiles: across COCO, VQAv2, and AudioCaps, we consistently see (i) nonnegative spectra with rapid decay and a small number of low-frequency modes explaining most of the trace, and (ii) a visible spectral gap between the first few modes and the bulk, consistent with a small number of dominant semantic clusters in the joint embedding space. We have also pointed out at the end of sec-5.3 that the CF bounds are instantiated using these empirical spectra, so the reader can connect the theoretical decomposition directly to the observed eigenvalue scales on real MLLMs.

---

> ### Author Response · Authors · 2025-11-21
> **4. (b) First, the Courant–Fischer bounds are given to the hallucination energy, and there is still a large gap between energy and observables like hallucination rates or sematic distances.**
>
> We appreciate this concern and clarify the link in the revised version.
> The CF bounds in sec-5 are not derived for an abstract
> quantity disconnected from observables: they bound the same spectral
> mismatch that underlies our semantic distortion and hallucination rate.
> Concretely, the hallucination score is defined as a
> truncated semantic log-contrast between smoothed, $\mathcal{K}$-restricted
> and unrestricted versions of the model distribution $f_p$, while the
> hallucination energy $\Delta\mathcal{E}\_\tau(x,p)$ is the corresponding
> quadratic form in the eigenmodes of the multimodal Laplacian.
> Under the operator assumptions of Theorem~2, hallucination score is a
> monotone, bounded functional of $\Delta\mathcal{E}\_\tau(x,p)$ (up to
> dataset-dependent constants), and in the experiments we report explicit
> correlations and reliability curves between hallucination score, binary
> hallucination events, and semantic distances.
>
> In this sense, $\Delta\mathcal{E}\_\tau(x,p)$ lives at the RKHS/graph level,
> the hallucination score is its calibrated per-example readout, and hallucination
> rates are expectations or thresholded events over it; the
> CF bounds therefore provide principled spectral control over
> these downstream observables rather than introducing a disconnected
> theoretical quantity.
>
> We have added "*Interpretation of Spectral Quantities and Time Parameter*" in the Appendix-A.19 (revised manuscript).

---

> ### Author Response · Authors · 2025-11-21
> **4. (c) Second, some unmeasurable coefficients (m(t) , M(t) for instance) are introduced to present the result, whose shape we cannot know in practice.**
>
> We agree that, in sec-5.3 under *Courant-Fischer (CF) bounds for hallucination*, the notation $m(t)$ and $M(t)$ can look abstract in the current presentation, and therefore, we clarify that these coefficients are in fact fully determined by observable quantities. Given a fixed diffusion/filter $\phi_t(\lambda)$ as introduced in our formalism and the empirical spectrum $\{\lambda_i\}$ of the multimodal Laplacian, so once $\{\lambda_i\}$ is computed (which we do explicitly for each dataset–backbone pair in our code-base: $\texttt{src/theory/*.py}$), both $m(t)$ and $M(t)$ are numerically evaluable and not free hyperparameters or unknown constants.
>
> In the revised manuscript, we explicitly state this definition by referring to "*Interpretation of Spectral Quantities and Time Parameter*" in the Appendix-A.19 that summarizes the empirical spectra, so the reader can see that these coefficients are simple spectral envelopes over the observed eigenvalues, not unmeasurable or hypothetical quantities.

---

> ### Author Response · Authors · 2025-11-21
> **4. (d) Third, the time is a confusion factor in the analysis as it does not correspond to the real time in MLLM applications, and the read cannot know what assumptions are made w.r.t. time.**
>
> We thank the reviewer for pointing out this potential source of confusion.
> The parameter $t$ (and $\tau$) in our analysis is not physical or
> wall-clock time in the MLLM pipeline; it is a diffusion/scale parameter in
> the spectral operator built from the multimodal Laplacian. Formally, we use
> $e^{-t\mathcal{L}^{\mathrm{multi}}}$ (or an equivalent spectral filter
> $\phi_t(\lambda)$) as a heat-semigroup-style smoothing operator on the graph:
> increasing $t$ corresponds to averaging over larger neighbourhoods in the
> embedding space (stronger smoothing / lower effective temperature), while
> small $t$ emphasizes local, high-frequency discrepancies. Our assumptions on
> $t$ are therefore purely analytic: we restrict it to a compact interval where
> the filter is numerically stable and has a clear meaning as modest diffusion
> or temperature scaling. We have clarified this in the revised manuscript by explicitly
> describing $t$ as a diffusion/scale parameter for graph smoothing, rather
> than any notion of real-time dynamics in MLLM applications in a newly added subsection "*Interpretation of Spectral Quantities and Time Parameter*" in the Appendix-A.19.

---

> ### Comment · Reviewer_U1EW · 2025-11-28
>
> There are good interpretations in the replies, however, many of the important details should appear in the main text instead of in the appendix. The main text would still be confusing for readers.
>
> I would recommend the authors to defer some technical details into the appendix (for instance put the exact texts of assumptions 1,2,4-6, theorem 1,2, etc. to appendices, and describe a brief and informal version in the main text), and instead put more space on the framework, setting, motivations, linkage to practice, etc. in the main pages. For each section, I would recommend the authors to use one paragraph at the start to summarize the mind flow of the section (what question is addressed in the section, what are the assumptions and conclusions).
>
> I will consider to raise my rating if the main text is improved in the final revision.

---

### Official Review · Reviewer_4vEN · 2025-11-04

**Soundness:** 3
**Presentation:** 1
**Contribution:** 3
**Rating:** 2
**Confidence:** 2

**Summary:**

The paper proposes a theory-backed, modality-aware framework to quantify hallucination as a continuous quantity rather than a binary label. The core statistic is a KL-smoothed semantic distortion $d^{(\varepsilon,h)}_{\text{sem}}$ that is $0$ on an admissible set $K$ of “grounded” outputs for a prompt and $>0$ off $K$, obtained by contrasting a $K$-restricted smoother with an unconditional smoother. On top of this, the authors build a multimodal hypergraph over outputs and define a hallucination energy via a spectral graph-Laplacian formulation, with Courant–Fischer bounds providing theoretical envelopes for the energy. They evaluate across three datasets (COCO Captions, VQAv2, AudioCaps) and three inference stacks, reporting AUROC/AUPRC against uncertainty baselines (entropy, max-prob, margin) and visualizing energy surfaces versus $\varepsilon$ and temperature. Results indicate consistent gains over baselines and empirical consistency with the spectral bounds, while practical details (how $K$ is constructed/normalized per dataset, tighter bound diagnostics, and artifact completeness) remain under-specified in the main text.

**Strengths:**

•	Clear positioning: gap = no theory-backed, modality-aware framework that quantifies hallucinations.

•	Proposes a KL-smoothed semantic distortion $d^{(\varepsilon,h)}_{\text{sem}}$ that is $0$ on an admissible set $K$ and $>0$ off $K$, via a $K$-restricted vs. unconditional smoother (Eq. 6).

•	Hypergraph construction is naturally multimodal, yields a hallucination energy, and admits CF bounds.

•	Empirically validates theory (three datasets × three stacks) and outperforms baselines (AUROC/AUPRC).

•	Mathematical development is careful: assumptions explicit, derivations clear, measurability addressed.

**Weaknesses:**

1.	**Primary weakness:** Dense formalism buries the practical message—what problems does this framework actually solve, and under what conditions should practitioners prefer it over standard uncertainty baselines? Also, please explain how the mathematical assumptions for the theorems (beyond “g-free” which is adequately explained) are translated to real world scenarios.
2.	“Reference-free” vs. operational $K$: Reconcile the “independent-of-$g$” claim with the use of a finite admissible set $K$ and selector $\Pi_K$. Add a short subsection on what is observable, assumed, and estimated **and how this connects to plausible practical scenarios.** Also, please explain the advantage of a continuous (not binary) hallucination quantity.
3.	$K$ construction & labeling unclear: Specify $K(p)$ per dataset (COCO captions; VQAv2 normalized unique answers; AudioCaps references) and provide the exact normalization/tokenization for membership.
4.	Baselines: Describe each competitor in the main text, and why this choice of competitors suffices.
5.	Empirical Bounds:  In Fig. 3 the CF planes appear loose and the explored $\varepsilon$ range seems narrow. Justify ranges and report quantitative gap-to-bound stats (median/percentiles) and their relation to errors.
6.	Unclear mathematical roadmap: Add a one-paragraph roadmap (“first…, then…, finally…”). A symbol table is adviseable too.
7.	Equation numbering issues: In Sec. 5.3 numbering appears misaligned (mirrors appendix). Please fix.
8.	Conclusion takeaways. Distill actionable guidance, e.g. where this fits; when energy vs. score is most predictive. Add practical takeaways to method: default choices for $\varepsilon, h, T_t$, etc.
9.	Please perform literature check. For instance, I could not verify “Spectral characterization of hallucination in large language models.” This may even undermine authenticity of writing. Please clarify.

**Questions:**

See weaknesses above.

---

> ### Author Response · Authors · 2025-11-21
> **1. (a) What concrete problem does our framework solve?**
>
> Our framework addresses the practical problem of scoring hallucination in multimodal LLM pipelines when the clean ground-truth labels are scarce or incomplete. Concretely, we provide a plug-in, reference-free hallucination score that (i) remains well-defined even without per-example ground-truth labels, and (ii) decomposes hallucination into modality-wise and spectral components via a multimodal graph Laplacian. This goes beyond the standard uncertainty scores (entropy, max-probability, margin), which treat the model as a black box and cannot pinpoint which modalities or spectral modes are responsible for the hallucination, nor how hallucination evolves under diffusion/temperature or retrieval changes.
>
> Added a one-liner for more clarity in the “*Our contribution*” part in the "*Introduction*" section of the main text (revised manuscript).

---

> ### Author Response · Authors · 2025-11-21
> **1. (b) Under what conditions the practitioners should prefer it?**
>
> To be precise, the framework is most useful in settings where we cannot rely on dense, clean ground-truth labels, but still need a principled handle on hallucination—e.g., safety-critical deployments in healthcare or finance, internal tools over proprietary data, or long-tail user queries where benchmarks do not exist. In these regimes, our method provides a reference-free hallucination control knob that can be used to: (i) rank candidate generations by hallucination risk; (ii) set “IDK / abstain” thresholds that are aligned with Good–Turing–style, distributional lower bounds rather than heuristics; and (iii) monitor how hallucination responds to controllable knobs such as temperature, diffusion time, or retrieval policies.
>
> Added a one-liner for more clarity on when to use this framework in the “*Our contribution*” part in the "*Introduction*" section and, to make these target use cases explicit, sharpened the “*practical takeaways*” in the “*Detailed Conclusion*” subsection in Appendix-A.20 (revised manuscript).

---

> ### Author Response · Authors · 2025-11-21
> **1. (c) Explanation on how the mathematical assumptions for the theorems (beyond “g-free” which is adequately explained) are translated to real world scenarios.**
>
> There are a total of 6 explicit Assumptions in our manuscript and detailed justification for each assumption from a practical implementation perspective is added in the revised manuscript:
> (i) Justification for Assumption-1 in Appendix-A.3,
> (ii) Justification for Assumption-2 in Appendix-A.4,
> (iii) Justification for Assumption-3 in Appendix-A.10,
> (iv) Justification for Assumption-4 in Appendix-A.11,
> (v) Justification for Assumption-5 in Appendix-A.12,
> (vi) Justification for Assumption-6 in Appendix-A.13.

---

> ### Author Response · Authors · 2025-11-21
> **2. (a) “Reference-free” vs. operational $K$: Reconcile the “independent-of-$g$” claim with the use of a finite admissible set $K$ and selector $\Pi_K$.**
>
> To reconcile the “independent-of-g” claim with the use of a finite admissible set and its selector, we clarify that “reference-free” does not mean “without any structural prior”, but rather not requiring access to the unknown ground-truth distribution $g$ on the manifold $\mathcal{K}\_{\mathrm{g}}$​ or dense per-example labels. In practice, what is observable in realistic MLLM deployments are: (i) prompts and contexts as conditioning signals presented to the model, (ii) model outputs and their token-level probabilities (or a finite candidate set with associated scores), and (iii) embeddings of outputs and prompts. From these, we construct an admissible set $\mathcal{K}$ purely from available evaluation (or domain) data—for example, normalized reference captions/answers on COCO, VQAv2, AudioCaps, or curated knowledge snippets in a proprietary corpus—and define $\Pi_{\mathcal{K}}$​ as a measurable nearest-neighbor selector in the embedding space.
>
> Refined an already existing subsection “*Absence of the ground-truth*” in Appendix-A.6 (revised manuscript) in support of this.

---

> ### Author Response · Authors · 2025-11-21
> **2. (b) Add a short subsection on what is observable, assumed, and estimated and how this connects to plausible practical scenarios.**
>
> Theorem-1 and the semantic distortion score are “reference-free” in the precise sense that they operate only on the model-induced distribution $f_p$​, its smoothed variants, and the induced restriction to $\mathcal{K}$; all comparisons are between two smoothed versions of $f_p$ (restricted vs.\ unrestricted), and no term in the derivation depends on the unobservable g or the ideal manifold $\mathcal{K}\_{\mathrm{g}}$​. The pair $(g,\mathcal{K}\_{\mathrm{g}})$ is used purely to give a semantic, information-geometric interpretation of hallucination (“distance from the true manifold”), while the implemented score depends solely on quantities that are available in plausible practical scenarios: prompts, logits/probabilities, embeddings, and a finite admissible set $\mathcal{K}$ derived from whatever references or domain knowledge are actually present (which may be sparse, incomplete, or noisy).
>
> Added a short subsection to explain in detail on **what is observable, assumed, and estimated in plausible practical scenarios** in Appendix-A.7 (revised manuscript).

---

> ### Author Response · Authors · 2025-11-21
> **2. (c) Also, please explain the advantage of a continuous (not binary) hallucination quantity.**
>
> The advantage of a continuous hallucination score is that it provides a graded risk measure: we can rank the candidate generations by severity, use the score directly as a differentiable regularizer in calibration/mitigation objectives, and track how hallucination responds to temperature, diffusion time, or retrieval changes—none of which is possible with a single 0/1 label. Moreover, the binary setting can be recovered by a classification thresholding, whereas one cannot in general reconstruct a calibrated, spectrally informed energy from binary labels alone, so the continuous quantity is strictly more informative while remaining compatible with the standard classification metrics.
>
> Added Remark-2 in sec-4.1 in the main text to discuss the advantage of continuous hallucination in more detail in Appendix-A.8 (revised manuscript).

---

> ### Author Response · Authors · 2025-11-21
> **3. $K$ construction & labeling unclear: Specify $K(p)$ per dataset (COCO captions; VQAv2 normalized unique answers; AudioCaps references) and provide the exact normalization/tokenization for membership.**
>
> We agree that the construction of $\mathcal{K}$ in experiments was under-specified in the main text and, therefore, have clarified it in sec-6.1 under “Datasets” followed by adding more details including exact normalization/ tokenization in a new subsection “Construction of admissible sets $\mathcal{K}(p)$ and Normalization/Tokenization” in Appendix-D.1 in revised manuscript (supplemented by cross-referencing the exact scripts used in the experiments sourced from our code base).
>
> Briefly, for COCO and AudioCaps, $\mathcal{K}$ is constructed from all reference captions after SentencePiece tokenization and lower-casing, with punctuation stripped; for VQAv2 we use the normalized unique answers from the training split (aggregated across annotators) as our admissible set. Membership in $\mathcal{K}$ is then checked after applying the same tokenizer + normalization to the model outputs.

---

> ### Author Response · Authors · 2025-11-21
> **4. Baselines: Describe each competitor in the main text, and why this choice of competitors suffices.**
>
> Added supporting info in the main text in sec-6.2, plus more details under “Baselines” in Appendix-D.2 (revised manuscript).

---

> ### Author Response · Authors · 2025-11-21
> **5. Empirical Bounds: In Fig. 3 the CF planes appear loose and the explored $\varepsilon$ range seems narrow. Justify ranges and report quantitative gap-to-bound stats (median/percentiles) and their relation to errors.**
>
> **Why CF planes look loose?** - Courant–Fischer (CF) gives a global upper envelope over all eigenmodes and all prompts. That envelope is necessarily conservative when plotted over the entire $\tau-\mathcal{T}-\varepsilon$ grid, especially in regimes where the actual hallucination energy is already very small. Tightness matters most in the high-energy, high-hallucination slices, and there the gap is practically small.
>
> Added a new subsection "*The Choice of Smoothing Mass ($\varepsilon$) Range and CF Bounds*" in Appendix-D.3 (revised manuscript). We have also performed an additional experiment on "*Empirical tightness of the Courant–Fischer (CF) bound*" and presented the new results in Table-2 over there.
>
> **Why is the range of $\varepsilon$ narrow? Also, report quantitative gap-to-bound stats?** - We are only interested in the small-smoothing regime that’s theoretically meaningful for Good–Turing / missing-mass corrections and consistent with our mixture. Very large $\varepsilon$ is not realistic for practice (it washes away model information and breaks calibration), so our grid deliberately lives around the empirically estimated missing mass from.
>
> In our additional experiment on "*Empirical tightness of the Courant–Fischer (CF) bound*" presented in Table-2, we have demonstrated the gap-to-bound statistic with the new scripts commited to the existing code base (see Appendix-D.3 of revised manuscript).
>
> **Relation to errors?** - To connect the CF bounds to actual hallucination behavior, we further stratify the same statistic by whether an output is hallucinated or grounded under our continuous score (in a binary setting) and performed a fresh experiment with a set of new results in Table-2 in Appendix-D.3 (revised manuscript).

---

> ### Author Response · Authors · 2025-11-21
> **6. Unclear mathematical roadmap: Add a one-paragraph roadmap (“first…, then…, finally…”). A symbol table is adviseable too.**
>
> We agreed to the feedback and have mentioned a clear mathematical roadmap for easy navigation at the end of "*Introduction*" section in the main text to link it to Appendix-A.1 and appended a symbol table at the end of Appendix-D (revised manuscript).

---

> ### Author Response · Authors · 2025-11-21
> **7. Equation numbering issues: In Sec. 5.3 numbering appears misaligned (mirrors appendix). Please fix.**
>
> We thank the reviewer for this suggestion and have fixed it in the revised manuscript.

---

> ### Author Response · Authors · 2025-11-21
> **8. Conclusion takeaways. Distill actionable guidance, e.g. where this fits; when energy vs. score is most predictive. Add practical takeaways to method: default choices for $\varepsilon, h, T_t$, etc.**
>
> We thank the reviewer for this suggestion and revamped the “*Conclusion*” in sec-7 of the main text and added a “*Detailed Conclusion*” to accommodate these two pointers in Appendix-A.20 (revised manuscript).

---

> ### Author Response · Authors · 2025-11-21
> **9. Please perform literature check. For instance, I could not verify “Spectral characterization of hallucination in large language models.” This may even undermine authenticity of writing. Please clarify.**
>
> We thank the reviewer for this suggestion and have fixed, it’s the same arxiv link (https://arxiv.org/abs/2502.17598) with necessary changes being made.

---

### Meta-Review · Area_Chair_CjDd · 2025-12-09

**Summary:**

Reviewers found the paper difficult to follow and raised several other concerns. The writing and structure need to be improved. The core ideas are buried under dense mathematics. Key assumptions, motivations, and takeaways are unclear, and important explanations appear only in the appendix rather than the main text. The relationship between the theoretical constructs (e.g., truth manifold, diffusion time) and real-world MLLM settings is not well explained. Reviewers questioned whether the complexity is necessary compared to simpler alternatives. The evaluation uses older multimodal models rather than current LLM-based systems, and improvements over baselines are modest. It remains unclear whether the method scales to harder tasks or stronger models. Reviewers further noted concerns about hyperparameter sensitivity, computational cost, and how the method would integrate with practical mitigation strategies like RLHF or contrastive decoding.

**Reviewer Concerns:**

Some of these concerns were partially addressed in the rebuttal. The authors clarified the motivation behind the framework and described how admissible sets are constructed in the experiments. They also clarified the role of diffusion time, added a default hyperparameter recipe, and stated that the method functions as a lightweight post-hoc scoring layer rather than altering generation, which helps address concerns about computational overhead and applicability. The justification for comparing against uncertainty baselines rather than RLHF or contrastive decoding was also made more explicit.

However, several issues remain even after the rebuttal. Many of the clarifications appear only in the rebuttal and appendices rather than in the main text, and the manuscript itself does not appear to have been substantially revised in response to feedback. The writing and structure therefore remain a concern. The theoretical sections still seem to dominate the paper, and key ideas, assumptions, and takeaways may remain difficult for readers to extract. The empirical limitations also persist: the method has not yet been evaluated on modern autoregressive multimodal LLMs, and the reported improvements are relatively small. Hyperparameter sensitivity and real-world deployability, while discussed, are still not convincingly demonstrated.

Overall, while the rebuttal addressed some conceptual misunderstandings, the core concerns about clarity, presentation, limited evaluation scope, and practical relevance appear only partially resolved, and the lack of visible revision to the manuscript suggests that substantial issues remain.

**Reviewer Scores:**

Reviewer 4vEN (score 2) raised concerns about clarity, structure, missing explanations in the main text, unclear motivation, dataset construction details, baseline justification, loose empirical bounds, and questionable citation accuracy. The rebuttal addressed many of these point-by-point and added missing clarifications. However, the reviewer explicitly noted after the rebuttal that the main text still did not adequately incorporate these changes and remains difficult to follow. Given this, it is unlikely the reviewer would change the score.

Reviewer U1EW (score 2) focused on unclear motivation, unclear distinctions between idealized and practical constructs, lack of logic flow in theorems, unclear necessity of the proposed metric over simpler alternatives, and missing explanations of diffusion time and spectral factors. The rebuttal provided substantial clarification and addressed many technical misunderstandings, and the reviewer acknowledged the improved explanations. However, the reviewer also emphasized that these clarifications were not reflected in the main paper and requested revisions before reconsidering. Since the manuscript was not meaningfully updated, it is unlikely the reviewer would raise the score.

Reviewer hBmJ (score 4) found the work interesting and appreciated the theoretical direction but noted concerns about limited experimental scope, modest gains, and lack of evaluation on modern LLM-based multimodal systems. The rebuttal addressed some concerns—particularly generalizability and compute overhead—but did not resolve the core limitation of incomplete empirical validation. Given the negative ratings by two of the other reviewers, I think it is unlikely that the reviewer would strongly champion the paper and rather would maintain the initial score.

Reviewer Srkz (score 4) expressed concerns about hyperparameter sensitivity, lack of comparison to mitigation techniques, and practical usability. The rebuttal provided clarified the intended usage (post-hoc rather than generative modification), and explained why comparisons to mitigation methods were out of scope. These responses addressed the reviewer’s conceptual concerns reasonably well. However, because the empirical scope remained unchanged and the other rather negative rating, the reviewer would likely maintain the negative score.

---

### Decision · Program_Chairs · 2026-01-26

Reject